


**Changes in the future summer Mediterranean climate: contribution of teleconnections**
**and local factors.**
Monika J. Barcikowska[1], Sarah B. Kapnick[2], Lakshmi Krishnamurty[3], Simone Russo[4],
Annalisa Cherchi[5], Chris K. Folland[6,7,8]
[1]Environmental Defense Fund, New York City
[2]Geophysical Fluid Dynamics Laboratory, National Oceanic and Atmospheric Administration, 201 Forrestal
Road, Princeton, NJ 08540, USA
[3]Princeton University, GFDL Princeton University Forrestal Campus, 201, Forrestal Road, Princeton, NJ
08542, USA
[4]European Commission, Joint Research Centre, Via Enrico Fermi, Ispra, Italy
[5]Fondazione Centro Euro-Mediterraneo sui Cambiamenti Climatici, and Istituto Nazionale di Geofisica e
Vulcanologi a, Bologna, Italy
[6]School of Environmental Sciences, University of East Anglia, Norwich, UK
[7]Department of Earth Sciences, University of Gothenburg, Sweden
[8]International Centre for Applied Climate Sciences, University of Southern Queensland, Australia
**Abstract**
The realistic simulation of the summer Mediterranean climate requires not only refined spatial scales, but also
an adequate representation of land-atmosphere interactions and teleconnections. Addressing all of these issues
remains a challenge for most of the CMIP3/CMIP5 generation models. In this study we analyze high-
resolution (~0.5° lat x lon) RCP8.5 future projections of the Geophysical Fluid Dynamics Laboratory CM2.5
model with a new incorporated land model (LM3).
The simulated regional future changes suggest pronounced warming and drying over most parts of the
Mediterranean. However the changes are distinctively less radical when compared with the CMIP5
multimodel ensemble. Moreover, changes over the Southeast (off the coast area of the Balkans) and Central
Europe indicate not only a very modest warming, compared to the CMIP5 projections, but also wetting
tendencies.
The difference of CM2.5 projections of future changes over previous-generation models highlights the
importance of a) a correctly projected magnitude of changes of the North Atlantic Oscillation and its regional
impacts, which have the capacity to partly offset the anthropogenic warming and drying over the western and
central Mediterranean; b) a refined representation of land surface-atmospheric interactions, which are a
governing factor for thermal- and hydro-climate over Central and Southeastern Europe.
The CM2.5 projections also indicate a maximum of warming (Levant) and drying (Asia Minor) over the
eastern Mediterranean. The changes derived in this region indicate a decreasing influence of atmospheric
dynamics in maintaining the regional temperature and precipitation balance and instead an increasing
influence of local surface temperature on the local surface atmospheric circulation.



**1. Introduction**
The climate of the Mediterranean region is primarily characterized by mild, wet winters and hot, dry summers.
This climate zone is located between 30°-45°N, hence it is affected by the variability of the atmospheric
circulation both in mid-latitudes and in the tropics. Moreover, the geomorphological characteristics of the
Mediterranean Sea region, including gulfs, peninsulas, islands, as well as the mountain ridges surrounding the
Mediterranean Sea, make the climate of this region distinctively complex.
To a large extent, mid-latitudes influence winter precipitation in the northern parts of the region, particularly
through the North Atlantic Oscillation and the East Atlantic pattern (e.g. Hurrell, 1995, Krichak et al. 2002,
Barcikowska et al. 2017b). The summer expression of the NAO (Folland et al. 2009, Linderholm et al., 2009,
Blade et al. 2012), due to its northeastward-shifted location, affects the Mediterranean region, as well.
Nonetheless, this issue has been investigated to a lesser extent (Hurrell et al. 2003, 2009, Folland et al. 2009,
Blade et al. 2012). The northward shift of the Hadley cell in the summer reveals a connection between the hot
and arid eastern part of the region and the Asian and African monsoon (Rodwell and Hoskins 1996, Ziv et al.
2004). The regional summer climate features a seasonal minimum of the rainfall, persistent mid-level and
upper–level subsidence and low-level northerlies (Raicich et al. 2003, Mariotti et al. 2002) centered over the
central-eastern part of the Mediterranean. The latter are called the Etesian winds (HMSO 1962; Metaxas 1977;
Maheras 1980; Prezerakos 1984; Reddaway and Bigg 1996; Zecchetto and de Biasio 2007; Chronis et al.
2011), which result from the zonal pressure gradient forged by the Atlantic subtropical anticyclone and the
western flanks of the Asian monsoon heat low (Bitan and Saaroni 1992; Saaroni and Ziv 2000; Alpert et al.
2004; Saaroni et al. 2010). Ziv et al. 2004 demonstrated that the cool air advection of the Etesian winds
counterbalances the adiabatic warming of the mid- and upper level subsidence, maintaining in this way a
thermal balance over the eastern Mediterranean.
The geographic location and socio-economic state of the Mediterranean makes the population in this region
particularly vulnerable to climate change. The southern part of the Mediterranean, which is dominated by
agricultural activities, is especially sensitive to prolonged water shortages and their consequences, such as
drought and wildfires. Giorgi and Lionello (2008) found this region to be particularly responsive to the
projected climate change, calling it a climate hot spot. Most future climate projections indicate a very strong
warming and reductions in precipitation during the summer season for this region (Meehl et al. 2007,
Alessandri et al. 2014, Mariotti et al. 2015). These changes can severely impact water and food security.
At the same time, observational studies have yet to find unambiguous evidence of decreasing precipitation
(Blade et al. 2012, Blade et al. 2012b, Giorgi and Lionello 2008). Additionally, Blade et al. (2012) questioned
the credibility of future summer climate projections in this region. The authors argued that impacts of the
summer NAO teleconnection (hereafter SNAO) on the European hydroclimate are underrepresented in
CMIP3 models. The SNAO, which yields a bipolar precipitation pattern, with wetting over southern Europe
and drying over the northern parts in its positive phase, may play an important role in offsetting the
anthropogenic forcing response. In future climate projections, these areas have shown opposite sign
tendencies with respect to rainfall (the dry Mediterranean gets dryer and wet northern Europe gets wetter).
Hence the underrepresentation of the simulated impacts of SNAO could have spuriously amplified projected
Mediterranean warming and drying. Kelley et al. (2012) indicated that CMIP5 models show a rather modest
improvement in the simulated regional hydroclimate, most likely due to increased horizontal resolution.
Historical simulations of CMIP5 models still differ from the observations, for example by showing a strong
wetting over the northwestern parts of Europe and drying over the southwestern parts of the Mediterranean



(e.g. Kelley et al. 2012), though some earlier lower resolution models do show strong drying over many,
though not all, parts of north west Europe as well (e.g. Rowell and Jones, 2006). These inconsistencies do not
add to the credibility of the current future projections for the Mediterranean and suggest that the severe
simulated regional warming and drying requires further investigation, particularly in higher resolution models.
A further caution is evidenced by the fact that particularly over the last half century the SNAO and the
summer European atmospheric circulation as a whole have shown strong multi-decadal variability
(Linderholm and Folland, 2017), just as the winter NAO (Scaife et al. 2008). There is also paleoclimate data
evidence of related multi-decadal to century timescale variations in the SNAO and Mediterranean climate
since 1441 (Linderholm et al, 2009).
The interpretation of the strong projected future warming over the eastern Mediterranean region remains
debatable. Cherchi et al. (2016) attempted to interpret the projected for this region summer changes from the
perspective of the monsoon –desert mechanism (Hoskins 1996, Rodwell and Hoskins 1996), which links the
hot and arid climate of the eastern Mediterranean to the South Asian monsoon rainfall. This mechanism, as
shown in the idealized simulations of Rodwell and Hoskins (1996), can be enforced by the diabatic heating of
the monsoon convection, which triggers westward propagating Rossby waves and a subsidence and warming
localized over the eastern Mediterranean (hereafter EMED). Given the fact that most CMIP5 models project
an increase in the future Asian Monsoon precipitation, one would also expect a strengthening of a subsidence
in the EMED region. However projected changes indicate rather opposite tendencies of the subsidence in this
region, despite a strong severe drying and warming. Cherchi et al. (2016) could not unambiguously resolve
this inconsistency, suggesting an explanation associated with either changing representation/impacts of the
"desert-monsoon" mechanisms or the emerging presence of non-linear processes in the regional surface
circulation.
On the other hand, Seneviratne et al. (2006) identified soil moisture-temperature feedbacks as a dominant
factor controlling summer temperature variability in the Mediterranean and Central Europe in a changing
climate. Soil moisture-climate feedbacks were also linked (Diffenbaugh et al. 2007) to a non-linear warming
of hot extremes in climate change projections for the Mediterranean. Hirschi et al. (2011) confirmed the effect
of soil moisture availability for hot extremes in observations in Southeastern Europe and found also that the
soil moisture-temperature feedbacks in RCMs are often overestimated over Central Europe. Several studies
(Christensen and Boberg, 2012, Mueller and Seneviratne, 2014), argued that the deficiencies in the
atmosphere - land surfaces feedbacks are a primary cause of a strong summertime warm bias in most of the
CMIP3 and CMIP5 models, which spuriously amplifies the projected future temperatures.
In the following study, we aim to: a) analyze future summer climate changes over the Mediterranean region
and b) investigate a possibility of emerging new regional climate regimes, describing these regimes, and
quantifying their contribution to the projected future changes over Europe. Identifying a regional signal
attributed to a particular mechanism or climate regime is not an easy task, given the complexity of this climate.
This complexity stems not only from the complex morphology of the region, but also from the teleconnections
acting on yearly-to-multi-decadal timescales (e.g. SNAO), and multi-decadal changes caused by
anthropogenic gases and aerosols. Addressing these issues requires not only integrations at high spatial
resolution, but also long control runs with fixed levels of radiative forcing. Thus, here we will use high spatial
resolution (~50km) GFDL CM2.5 model simulations, including a control simulation consisting of about 1000
years with radiative forcing fixed at the preindustrial levels, a five-member ensemble of historical simulations
and also future projections as defined in the Intergovernmental Panel on Climate Change (IPCC) Relative



Concentration Pathway 8.5 (RCP8.5, Meinshausen et al. 2011, Riahi et al. 2011). The CM2.5 model
(Delworth et al., 2012) is a descendant of the GFDL CM2.1 model (Delworth et al., 2006) that incorporates
higher spatial resolution and improved land model (LM3), improving this way the simulated hydroclimate
over many continental regions including Europe.
In order to attain the objectives of the study, we will create an analysis of the Mediterranean summer climate
and interpret the results through the prism of large-scale circulations, teleconnections and the regional factors.
Section 2 describes the data and the methodology. Section 3 focuses on the summer time-mean climatology of
the region, as well as its teleconnections, using the long control run. It evaluates the performance of the model
in terms of the simulated regional precipitation, as well as large-scale circulation features, which shape the
summer regime of the Mediterranean climate. It also examines the main components of internal variability
dominating atmospheric circulation over the Euro-Atlantic region, such as the summer NAO, and their impact
on Mediterranean climate. The last part of this section focuses on a representation of key dynamical features
shaping the climate regime over the eastern Mediterranean, i.e. including the linkage between the mid- and
upper-level subsidence and the low-level pressure gradient (and the associated Etesian winds) together with its
coupling with the Indian Monsoon. Section 4 investigates future climate changes over the Mediterranean
derived from the CM2.5 projections. It examines the regional changes from the perspective of large-scale
circulation over the Euro-Atlantic and associated summer NAO teleconnections. The derived regional changes
are also interpreted in the context of the changing relationships between the dynamical factors governing the
eastern Mediterranean. Again, the long control run is used to verify whether the derived changes in the eastern
Mediterranean climate regime can be attributed to the local effects of the warming Mediterranean and how
these effects will impact the whole Mediterranean and European climate. Section 5 discusses and summarizes
the main results.
**2. Data and Methods**
This study analyzes experiments performed using the Geophysical Fluid Dynamics Laboratory (GFDL)
CM2.5 coupled model. CM2.5 has an atmospheric and land surface horizontal grid scale of approximately 50
km with 32 levels in the vertical. The horizontal grid scale of the ocean increases from 28 km in the tropics to
8-11 km in high latitudes.  Details of the CM2.5 model features can be found in Delworth et al. (2012). The
representation of the summer precipitation climatology in CM2.5 is also compared using a 4000-years control
run of GFDL CM2.1, that is the CM2.5's predecessor. The latter incorporates a grid scale of 2° latitude x 2.5°
longitude for the atmosphere. The ocean resolution is variable being approximately 1° latitude x 1° longitude,
with a finer meridional resolution in the tropics. The CM2.1 model has 24 vertical levels (Delworth et al.
37 2006).
Our experiments (see Table 1) consist of a control simulation (hereafter CTRL), an ensemble of historical
simulations (hereafter HIST), and an ensemble of future projections (hereafter PROJ). The CTRL simulation
consists of a 1000-year integration, where greenhouse gas and aerosol compositions are held fixed at the
levels of the year 1860. This data is used to verify the performance of the model in simulating features of the
mean summer Mediterranean climate (section 3.1). The time-mean large-scale circulation features are
analyzed based on the monthly means of hydro-meteorological variables for the summer (June, July and
August, hereafter JJA) season. The simulated features of large scale circulation are compared with reanalysis
data of monthly pressure at mean sea-level (hereafter SLP), wind vectors at the 850hPa and 200hPa levels,



and vertical velocity at 200hPa for the period 1979-2017. Reanalysis data is provided by the NCEP-DOE AMIP-II Reanalysis 2 with 2.5° x 2.5° horizontal resolution and 17 vertical levels (Kanamitsu et al., 2002; https://www.esrl.noaa.gov/psd/data/gridded/data.ncep.reanalysis2.html).

The simulated precipitation is compared with the seasonal time-averaged precipitation provided by the University of Delaware (V4.01), Legates and Willmott 1990; http://climate.geog.udel.edu/ ~climate/html_pages/README.ghcn_ts2.html (last access: July 2018). This is a global gridded land data set with 0.5° x 0.5° horizontal resolution.

The observational analysis of the summer North Atlantic Oscillation (section 3.2) is carried out using July-August mean sea level pressure (SLP), provided by NOAA/ESRL PSD 20th Century Reanalysis version 2c (Compo et al. 2006, https://www.esrl.noaa.gov/psd/data/20thC_Rean/). The spatial patterns of the dominant component of the SLP variations are computed with Empirical Orthogonal Function (EOF) analysis, over the domain [25°-70°N, 70°W-50°E], following Folland et al. 2009. The robustness of the pattern is tested against the length and chosen periods. The relationship between the SNAO and the North Atlantic storm track (Fig 7) is derived by computing correlations between the SNAO index and the proxy of storm track in July-August. The storm tracks are represented with the standard deviation of the 300 hPa geopotential height, calculated form the daily and bandpass-filtered data, following Folland et al. 2009. In this study we have used a Butterworth filter with a bandwidth of 2-8 days.

The teleconnection of the Mediterranean climate with SNAO is analyzed using the full (1000 year) CTRL run (section 3.2). Dominant components of the atmospheric variability are derived from an EOF analysis applied to the sea level pressure over the Euro-Atlantic region [25°N–75°N, 70°W–50°E] in "core summer" (July-August), following Folland et al. 2009. The impact of this teleconnection on Mediterranean climate is estimated using correlations with the regional temperature and precipitation.

The summer climate regime of the eastern Mediterranean is examined from the perspective of the regional mid- and upper-tropospheric subsidence and its physical linkage with the surface circulation (section 3.3). The seasonal variability of the subsidence over the eastern Mediterranean is derived from EOF analysis applied to vertical velocity (omega) fields at 500 hPa, and also at 200 hPa over the region covering the Mediterranean, North Africa and Middle East [18°N–46°N, 10°W–50°E] in June-August season (Fig 8a). The physical linkage between the subsidence and surface circulation is estimated using correlations between the time series of the first EOF component (PC1) and the regional sea level pressure, geopotential height and wind vectors at 850 hPa (Fig 8b,c,d). The relationship between the EMED region dynamics and the Indian Summer Monsoon (Fig 9) is estimated by computing additional correlations with precipitation, outgoing long wave radiation, and the vertically integrated water column.

An additional analysis investigates the potential influence of the local temperature (i.e. in the EMED region) on the derived local dynamical relationships (section 4.3 and Supplemenary Material: Fig SI1, SI2, SI3). Therefore, the derived correlations were differentiated between samples with the 300 warmest and the 300 coldest summers (July) over the Mediterranean, chosen from the control run time series. Their selection was based on surface temperature in the EMED region [30°-36°N, 36°-42°E]. Additionally, a diagnosis of temperature impacts on the regional atmospheric circulation was performed using from composite differences between the two temperature samples and the associated relative humidity, sea level pressure, wind components, geopotential height, vertical velocity and precipitation. The results were corroborated by testing their sensitivity to the precise choice of the region (Figure SI3).



Climate change signals are investigated using a five-member ensemble of CM2.5 simulations with changing
radiative forcing over the period 1861-2100, which include historical simulations and future projections. The
forcing follows the protocols of the Coupled Model Intercomparison Project Phase 5 (http://cmip-
pcmdi.llnl.gov/cmip5/forcing.html). For 1861-2005, the radiative forcings are based on observational
estimates of concentrations of well-mixed greenhouse gases (GHG), ozone, volcanoes, aerosols, solar
irradiance changes and land-use distribution. Radiative forcing for the period 2006-2100 follows an estimate
of projected changes defined in the IPCC RCP8.5 scenario (Meinshausen et al. 2011, Riahi et al. 2011). This
scenario assumes high population growth, slow technological change and energy intensity improvements, and
a lack of developed climate change policies, resulting in large energy demand and GHG emissions.
Future summer climate changes are analyzed by comparing climatological surface temperature, sea level
pressure, wind components, total precipitation rate and vertical velocity in June-August (JJA) between 1961-
1999 (HIST runs) and 2061-2099 (PROJ runs). Future changes in SNAO teleconnections were analyzed by
comparing longer periods, e.g. 1960-2010 and 2040-2100, of the HIST and PROJ runs. The analysis was also
tested for shorter periods (i.e. 50 and 30 years) which did not change the results in qualitative terms.
The evolution of the SNAO fingerprint in the 20[th] and 21[st] century is analyzed by projecting the vector time
series of HIST and PROJ runs (240 yrs, 1861-2100) on the SNAO eigenvector, derived from the CTRL run.
The derived signal time series are averaged over the five-members ensemble. Future changes in the spatial
fingerprint of SNAO, as well as its impact on the Mediterranean hydroclimate, are analyzed by applying the
EOF analysis to detrended periods long six decades of the SLP in historical and future runs. The analysis took
into account the fact that the sign of each derived EOF is arbitrary, unifying them before deriving the principal
component (PC) time series. Each of the five SNAO time series for the 1950-2010 and 2040-2100 periods
were correlated with the respective detrended precipitation fields and an average of the correlation fields
computed over each ensemble.
Estimation of the SNAO contribution to future changes in Mediterranean precipitation is based on
differences between 1961-1999 and 2061-2099, and a linear relationship derived between these two variables.
This regression relationship was computed primarily using the CTRL experiment and associated time series of
the SNAO and precipitation anomalies. Future changes in SNAO were estimated by subtracting the ensemble
average of the SNAO component time series in 2061-2099 from that of 1961-1999. The SNAO component
was reconstructed using the CTRL-based EOF. However, to take into account potential future changes in the
pattern of SNAO, we repeated the analysis using the period from the HIST and PROJ experiments, i.e. 1961-
35 2100.

Future changes in the dynamical linkages governing the summer climate regime over the eastern
Mediterranean were analyzed by comparing the five-decade long samples for July, i.e. 1960-2010 and 2050-
2100. The linkage was calculated in a similar manner to that of the control run using correlations between the
time series of the EOF over the EMED region subsidence and the atmospheric surface circulation fields. All
EOF time series were computed by projecting the respective run on the eigenvector derived from the control
run. The correlations were derived for each run (historical and future, respectively), using a priori detrended
timeseries. The final result shows the ensemble mean for the five-member historical and future correlations.





## 3 Results

### 3.1 Simulated climatology of the summer Mediterranean climate

This section analyzes the capacity of the model to reproduce the summer (JJA) climatology of the Mediterranean climate, as well as the large-scale atmospheric circulation which to a large extent shapes the regional climate.

Figure 1 demonstrates that the model captures the subtropical low-tropospheric circulation with high fidelity when compared with the reanalysis (NCEP-DOE). It reproduces accurately the zonal pressure gradient over the Mediterranean, both in terms of pattern and magnitude, forged by the difference between the subtropical anticyclone over the North Atlantic and the massive Asian monsoon heat low. The latter extends westward, through the Arabian Peninsula towards the Levant region and southern Asia Minor. Concomitant to the zonal pressure gradient and adjustments to the regional orography is a persistent west-northerly flow over the central and eastern Mediterranean (i.e. the Etesian winds). The model reliably captures its local-scale features, created by adjustments to the regional topography. This includes a local wind maximum centered over the Aegean Sea and its southern extension reaching to the Sahel region. These northerlies are also channeled through the Red Sea Straits and the Persian Gulf, reaching the Indian Ocean.

Figure 2 shows that the model reproduces the location and magnitude of the summer subtropical mid-troposphere anticyclone, which spreads from the eastern Mediterranean across the South Asia. The simulated mid-troposphere also captures the location and a realistic magnitude of the persistent mid-troposphere (500 hPa) subsidence (positive omega) which creates the exceptionally hot and arid climate of the eastern Mediterranean. This subsidence gradually decreases towards the Iranian Plateau, which together with ascending motion over the South Asian monsoon region, creates a large-scale time-mean zonal gradient. The simulated zonal gradient is well shown (Fig 3a) by a vertical cross-section of vertical velocity (omega) averaged over 20°-34°N between the EMED region (positive omega means strong subsidence) and the South Asia (negative omega means ascending air). This characteristic gradient agrees well with its observational counterpart (Fig 3b) both in terms of magnitude and pattern. Importantly, the model captures the observed local maximum of the EMED subsidence located at middle-tropospheric levels (300-700 hPa), the region most sensitive to the impact of the Indian monsoon.

Figure 4 shows climatologies of the Mediterranean precipitation provided by the observations, the CM2.5 control run, and also its low-resolution (CMIP3) predecessor, i.e. CM2.1 at their original horizontal resolutions. Although both CM2.1 and CM2.5 depict the general spatial features of the climatology (i.e. large values in the northern Mediterranean, particularly over the Alps and the Balkans), the former introduces large biases (up to 50%) in regions with sharp spatial gradients. CM2.5 resolves much better the spatial features of precipitation, clearly indicating the advantages of higher horizontal model resolution for regions with complex orography. However, precipitation magnitude in most mountainous areas, e.g. the northern Iberian Peninsula, the Alps and over Asia Minor, is apparently larger than in observations. Nevertheless, due to a relatively large observational uncertainty in many mountainous areas, it is difficult to validate the model rainfall climatology in these regions.

Kelley et al. (2012) underlined the importance of higher horizontal resolution in simulating precipitation over the Mediterranean. They compared the climatologies of CMIP3 and CMIP5, suggesting a modest improvement in the latter, most likely associated with increased horizontal resolution. However, neither of the CMIP precipitation climatologies could realistically capture the complexity of regional features (e.g. local



maxima over the northern Asia Minor and the Balkan coast) to such a high degree as in CM2.5.   Overall, our analysis indicates that the high-resolution CM2.5 control run provides a realistic mean representation of the surface- and upper-tropospheric circulation over the Mediterranean. The model skillfully simulates sharp gradients of precipitation over the morphologically complex terrain and its climatology is considerably better than its low-resolution predecessor, the CM2.1 model.

## 3.2 The impact of the summer North Atlantic teleconnections

The North Atlantic Oscillation (NAO) is the most prominent pattern of winter atmospheric variability in this region and numerous studies have demonstrated its impact on European hydroclimate.  Similar influences of its summer counterpart (the SNAO) have also been discussed  but with  considerable differences of detail which stem from a distinct northward shift of the SNAO pattern compared to the winter NAO. Using the SNAO definition in Folland et al. 2009, the observations-based SNAO shows a northern lobe centered over southern Greenland and a southern lobe located close to the British Isles (Folland et al. 2009 and Blade et al. 2012). The positive SNAO phase yields a stronger meridional SLP gradient over the North Atlantic, an enhanced anticyclonic southern lobe with dry conditions over northwest Europe and a rather wet conditions over the central Mediterranean. Therefore, the projected changes in SNAO under future climate warming, i.e. a pronounced strengthening towards its positive phase, may play an important role in offsetting the future warming and drying over the Mediterranean. On the other hand, the capability of CMIP3 models at simulating SNAO teleconnections with Mediterranean climate is rather ambiguous (Blade et al. 2012). This confusion arises partly because current literature has not yet reached a full consensus on the spatial definition (fingerprint), origin and impacts of the SNAO. The results of observational analysis vary, depending on the observational dataset, the chosen period, the chosen summer months, and the analysis method (Barnston and Livezey, Hurrell and van Loon 1997, Hurrell and Folland 2002; Hurrell et al. 2003, Cassou et al. 2005, Folland 2009, Blade 2012).

Folland et al. 2009 used for example the leading eigenvector of the sea level pressure in July-August, to define the SNAO pattern that explains 28% of the 2-month mean variance. Nevertheless, this study clearly stated that the derived component may stem from the combined effects of interannual to multi-decadal components, as further highlighted by Linderholm and Folland, 2017. For example, the observed overall positive trend and multi-decadal variations likely originate from both, climate variations like the Atlantic Multidecadal Oscillation in sea surface temperature, itself at least partly related to the Atlantic Meridional Overturning Circulation, anthropogenic warming and particularly the influence of changing anthropogenic aerosols (Delworth and Mann 2000; Enfield et al. 2001, Sutton and Hodson 2005, Knight et al. 2005, Rotstayn and Lohman 2002, Mann and Emanuel 2006, Baines and Folland, 2007, Booth et al, 2012, Barcikowska et al. 2017).

Blade et al. 2012 found an SNAO like pattern similar to the one in Folland et al. 2009, using the NCEP data set in the 1950-2010 period. However that study revealed a sensitivity of their SNAO to the chosen period (i.e., for shorter periods of about 30 years-length) with marked variations in the percentage of SLP variance explained over the chosen domain. For example, the anti-correlation  of SLP between the two SLP centers is -0.52 in the longer period 1950-2010, which they recommend for calculating the SNAO. However over some 30 year periods both before 1950 and especially afterwards, when SLP data is of better quality, the SNAO is not distinguishable from other EOF patterns or not a leading pattern. Thus the SNAO seems temporally less robust than the winter NAO.



This inconsistency supports the idea that summer atmospheric circulation over the SNAO region may be influenced by rather different key factors at different times, giving rise to time-varying dominant modes of apparent internal variability. Thus the character of apparent SNAO temporal variability, as defined by Folland et al 2009, between 1850 and 2017 is much more dominated by multi-decadal variability after about 1965 than before (Allan and Folland 2018, Fig 2.35). Allowing for this, and taking into account that each HIST run represents a different, non-deterministic state of internal climate variations, one should not expect to obtain from each run a replica of the observed SNAO component. On the other hand an increasing impact of the anthropogenic forcing in the HIST and PROJ runs, may intensify the SNAO contribution. However, the imperative of the following section is to test the capability of the model to simulate the SNAO as an independent, internally generated climate component, which would prove the physical validity of the statistically-derived component.

In the following two sub-sections, we investigate the dominant components of the atmospheric variability over the North Atlantic and their teleconnection with the Mediterranean climate in CM2.5. For this purpose we use the monthly output (July-August) of the full period (1000 years) of the CTRL experiment (Table 1). The spatio-temporal signature of the atmospheric circulation variability over the North Atlantic is analyzed using the EOF approach. The EOF is computed over the domain [25°-70°N, 70°W-50N°], following Folland et al. 2009. The analysis is also repeated for SLP observations taken from the 20CR dataset for the period 1870-2010, as well as from the HIST and PROJ runs. Teleconnections between the EOF time series and Mediterranean climate are diagnosed by correlating them with regional precipitation and temperature.

### 3.2.1 Spatial pattern of SNAO

In the CTRL run two EOFs dominate the variability of large-scale summer circulation over the North Atlantic. EOF1 contributes most to summer SLP variations, explaining twice as much total variance as EOF2. The percentages of explained variance (34% and 15%) resemble those of the observations-based EOF1 and EOF2 in Folland et al. 2009, i.e. ~28% and ~14% respectively. In the following we focus on the leading pattern. CTRL EOF1 (Figure 5a) resembles the observed SNAO pattern with a dipole SLP signature which is northward shifted compared to the winter NAO (shown in Barcikowska et al. 2017). The dipole pattern has a northern lobe over the south-western flank of Greenland and a southern lobe centered north of the Azores in the vicinity of ~45°N, 30°E. At its positive phase the SNAO dipole manifests negative SLP anomalies over the northern lobe and positive anomalies over the southern lobe, strengthening in this way the meridional SLP gradient over the North Atlantic. The pattern is similar when analyzed in the single months July and August. The simulated SNAO pattern is almost identical to the one derived from the HadCM3 model control run, as shown in Folland et al. 2009.

The location of the southern lobe in the SNAO simulated with the CM2.5 and HadCM3 models, both in CRTL mode, is distinctly south west of its observational counterpart, as shown for the most recent six decades in Blade 2012 and Syed et al. 2012. In Syed et al. 2012 the southern node of the leading SLP component, derived from the NCEP dataset, appears in the vicinity of the British Isles much as in the CRTL mode of HadGEM1 (Folland et al 2009). Blade et al. 2012 corroborated these results, using CPC, NCEP and the dataset in Trenberth and Paolino 1980 for the period 1950-2010. Nevertheless, Blade et al. 2012 pointed that the observed SNAO pattern depends on analysis period as described above. For example, SNAO dominates the variability in the 1940-1975 period, while since 1970s the dominant variability is confined only to the southern lobe, i.e. centered over the British Isles. These findings are consistent with the variations in EOF1 of



the 20CR reanalysis when comparing different periods through the twentieth century.  Figure 6 depicts  EOF1
derived from four 50-yr periods of the 20CR reanalysis (i.e. a. 1870-1920, b. 1900-1950, c. 1940-1990, d.
1960-2010). It shows clearly that the pattern of this leading component changes in time; similar results were
obtained for the independent 40-yr periods (1851-1890, 1891-1930, 1931-1970, 1971-2010 in Figure SI8).
The pattern derived for periods before 1950s (shown here for 1870-1920 and 1900-1950) closely resembles
the SNAO dipole simulated in the CM2.5 control run, i.e. including northern centers of action at southern
Greenland and with the southern lobe situated north of the Azores (~45°N, 35°E). In contrast, the observed
pattern from the last five decades (1960-2010) represents much closer a  monopole centered over the British
Isles, though there is still a weak opposite lobe over Southern Greenland. Overall, there is a change from near
two equal EOF1 lobes in early decades of the last 140 years to a considerably more dominant and stronger
southern lobe on recent decades. At the same time the southern lobe has moved north east to be centered over
or very close to the British Isles. The position of the northern lobe might also have changed but poorer data in
early decades in this region precludes a definitive conclusion. The patterns derived for the latter half of the
century (Fig 6c,d) are in best agreement with the one derived from 1950-2010 in Blade et al. 2012, or Folland
et al. 2009.
Analysis of EOF1 for the HIST runs shows an evolution similar to the observations-based SNAO pattern
(Figure 6). For the early observational period (1870-1920) SNAO patterns in all HIST runs (Figure 6b, Figure
SI6 left column) and observations resemble very closely the pattern in the CTRL run (Figure 5) such that the
northern lobe is over Greenland and the southern lobe is north of the Azores around 40-45ºN. However, for
the most recent period (1960-2010, Figure 6g, h) observations and HIST runs both show a distinct north-
eastward shift of the southern lobe compared to the CTRL-based SNAO pattern. This shift manifests as a
strong SLP anomaly extending from the British Isles towards Scandinavia and a weak anomaly close to the
southern tip of Greenland. In fact in four of the HIST runs there is a clear northeastward movement and
intensification of the southern lobe and weakening of the northern lobe between 1870-1920 and 1960-2010
(Figure SI6). This tendency becomes even more evident when the periods from PROJ runs are used (e.g.
1970-2030, 1970-2060 Fig SI7). For example, in the 1970-2030 period, all five derived SNAO components
replicate their observational counterpart in 1960-2010 with high fidelity. Note that the HIST runs are free-
running global coupled-model simulations with historical climate forcing data, and are therefore not
constrained with the boundary conditions of observed historical sea surface temperatures or circulation
patterns. This suggests an important contribution of anthropogenic forcing in shaping the SNAO, providing a
plausible explanation for differences in the SNAO pattern prior to anthropogenic forcing (CTRL run) or
including a smaller impact (observations and HIST runs in 1870-1920).
**3.2.2 Impact of SNAO on the regional hydroclimate**
The teleconnection between the SNAO simulated in CM2.5 and European precipitation resembles the
characteristic observed bipolar pattern of anomalies over the northern Europe and opposite-signed anomalies
over southern Europe and most of the Mediterranean region (e.g. Folland et al. 2009). Figure 5c shows
correlations between simulated precipitation and simulated time series of the SNAO component for the CRTL
run. Thus, variations of the SNAO phase are negatively correlated with the precipitation across the British
Isles, southern Scandinavia and eastern Europe, while being positively correlated with precipitation over the
Mediterranean.



The positive (negative) SNAO phase, manifested in the positive (negative) SLP anomalies over the southern
node, are consistent with an anomalous drying (wetting) over northwestern Europe. The simulated location of
the southern SLP node is located southwest of the British Isles, and so its largest impact is in this region with
negative correlations reaching  a value of about -0.7. The maximum impact of the observed teleconnection, as
shown in Folland et al. 2009, is of similar magnitude and exceeds ~0.6 though its location is centered over the
British Isles.  This stems from the difference in the position of the observed southern node of SNAO from that
in CRTL runs. For Mediterranean hydroclimate, the consequences of a positive SNAO phase are the inverse
of those in northwest Europe. Positive correlations (Fig 5c) indicate wetting over most of the Mediterranean
region and exceed 0.2.  These are centered over Italy, the central and southeastern Iberian Peninsula, and Asia
Minor.
This CM2.5 CRTL teleconnection and correlation values corresponds well with that derived from a historical
run of the GFDL CM2.1 model, the low resolution predecessor of CM2.5 (Blade et al. 2012).  The CM2.5
correlation coefficients over the Mediterranean are also quite similar to those of the observations-based SNAO
in Folland et al. 2009, exceeding locally 0.3 (1900-1998). In contrast, Blade et al. 2012, using a shorter data
set (EOBS, 1950-2010), found much larger correlations, locally exceeding 0.6. Nevertheless, the SNAO
variations observed in Blade et al. 2012 are largely shaped by a multi-decadal signal of natural and
anthropogenic origin. By contrast, anthropogenic climate components are not included in the CM2.5 control
run and the simulated SNAO teleconnection stems largely from interannual variations. Accordingly the
findings in Blade et al. 2012 are used here to validate this simulated teleconnection.
The SNAO teleconnection with hydroclimate in  northwestern Europe can be straightforwardly explained with
variations in large-scale circulation over the North Atlantic and its associated variations in the North Atlantic
storm tracks as shown in Folland et al. 2009. Using NCEP data, they showed that a positive (negative) SNAO
index relates to the North Atlantic storm tracks being shifted northward (southward) towards the northwestern
(southwestern) Europe. Here we confirm that relationship using the SNAO index and storm track proxy
derived from 20CR, as well as the proxy derived from NCEP. It is also worth noting that the strength of the
derived relationship (Figure 7) depends on the chosen period. For example, the derived correlations are
distinctively stronger for the period starting in mid-century (i.e. 1950-1990), which is dominated with the
dipole SNAO pattern (Figure 6e), than for the latter period (1970-2011), which features more of a monopole
over the British Isles (Figure 6g).
However, the impact of the SNAO on Mediterranean hydroclimate (shown by in observations by Linderholm
et al 2009 to include soil moisture) can be better explained through associated changes in a mid- and upper-
tropospheric geopotential height, consistent with Blade et al 2012. Correlations derived between the SNAO
time series and 500hPa geopotential height (Fig 5b), show the strongest influence over the western and central
Mediterranean, i.e. the Iberian Peninsula, Italy and to a smaller extent the Balkan coast. Well collocated with
this pattern are correlations with precipitation and surface temperature, although for the latter the impact of
the SNAO seems much smaller. Hence, anomalous precipitation in these regions during the positive SNAO
phase  likely stem from  an anomalous mid- and upper tropospheric trough,  associated cooling, and
intensified potential instability over the central Mediterranean. This reasoning is  consistent with that
discussed in Blade et al. 2012.



Overall, the results presented in this section demonstrate that CM2.5 is capable of simulating SNAO
teleconnections and their contribution to Mediterranean hydroclimate. The associated physical mechanisms
are consistent with observational findings and other modeling studies employing older generations of models.
**3.3 Summer climate regime over the eastern Mediterranean**
The summer climate regime over the eastern and central Mediterranean is dominated by two dynamical
factors, i.e. the mid- and upper - tropospheric subsidence, and the low-level northerly Etesian winds (HMSO
1962; Metaxas 1977; Maheras 1980; Prezerakos 1984; Reddaway and Bigg 1996; Zecchetto and de Biasio
2007; Chronis et al. 2011). Both factors are centered over the eastern Mediterranean near Crete. They have
opposite influences on surface temperature and the regional climate (Ziv et al. 2004, Tyrlis et al. 2013).
Tropospheric subsidence tends to make eastern Mediterranean summer climate extremely hot and dry. The
Etesian winds, resulting from the zonal pressure gradient between the subtropical North Atlantic anticyclone
and the heat low expanding over the South-West Asia and Arabian Peninsula, give rise to the advection of
cool air from southeastern Europe.
Moreover, Ziv et al. 2004 have identified a significant correlation between these factors. This likely stems
from the Asian Summer Monsoon teleconnection exerting an influence on both the surface and mid- and
upper-troposphere dynamics. The impact of the Asian Monsoon on the mid- and upper tropospheric
subsidence over the EMED was demonstrated by Rodwell and Hoskins (1996). The study showed that
diabatic heating, excited by the Asian summer monsoon deep convection and precipitation, (centered north of
Bay of Bengal (90°E, 25°N)) triggers a Rossby wave pattern response. The waves propagate westward, as the
warm thermal structure intrinsic to the wave pattern starts interacting with the mid-level westerlies, and
induces subsidence and hot and dry conditions over the eastern Mediterranean and the Sahara. This
mechanism has been corroborated in several observational and modeling studies (Tyrlis et al. 2013, Rizou et
al. 2015, Cherchi et al. 2014, Cherchi et al. 2016). On the other hand, the impact of the monsoon on the
Etesian winds has been shown (Rodwell and Hoskins 2001) to be a direct result of changes in the subsidence
over EMED, which via Sverdrup's equation trigger changes in the low-level northerly flow (i.e. the Etesians).
This agrees with Ziv et al. 2004 who pointed that changes in the low-level monsoon heat low, which expands
westward across Arabian Peninsula, could modulate the zonal pressure gradient and the concomitant Etesians.
In this section we investigate the ability of the CM2.5 control experiment to reproduce the summer climate
regime over the eastern Mediterranean, including a) the relationship between the surface dynamics and the
mid- and upper-tropospheric circulation over EMED; b) its teleconnection with the Indian Summer Monsoon.
The availability of 1000 years of integrations in CTRL provides a much better representation of climatic
internal variability compared to the relatively short observational record, allowing us to infer relationships
with a high statistical level of certainty. The analysis is based on simulated monthly variations of hydro-
meteorological variables, conducted for June, July and August separately. Here we will focus attention on
results for July. Results for June and August are shown in the Supplementary Information (Figure SI5). This
choice is driven by the fact that the magnitude of subsidence and the Etesians is at its maximum in July while
the response of the Rossby waves to monsoon rainfall is also strongest (Tyrlis et al. 2012, Lin et al. 2007, Lin
et al. 2009).
Figure 8d, (see also Ziv et al. 2004, Tyrlis 2012) shows that the model captures skillfully the observed
features of the regional relationship between the mid- and upper-tropospheric subsidence and surface
circulation. The relationship is depicted in Figure 8 as correlations computed between time series of EOF1



(Figure 8a) of the mid and upper- tropospheric vertical velocity (omega) over the Mediterranean and those of
geopotential height at 850hPa, wind vector, outgoing longwave radiation and precipitation.  EOF1 derived
either for omega at 500 hPa or 300 hPa, explains the dominant amount of variance (~33% and 35%,
respectively). Its spatial pattern closely resembles the simulated and the observed (Tyrlis et al 2012, Ziv et al.
2004) key feature of the mid- and upper omega summer climatology, i.e.  with a maximum located near Crete.
The correlations (Figure 8b,c), derived between the EOF1 500hPa and the regional surface circulation, depict
a physically consistent linkage (Figure 10d, Ziv et al. 2004, Tyrlis 2012) between  strengthening of the
subsidence over the EMED and strengthening of the zonal pressure gradient over the Mediterranean and thus
the Etesians over the EMED.
The Etesians spread southward from the Aegean Sea, through Egypt and towards the Sahel which is consistent
with observational findings underlining the role of the Etesians in regulating the moisture transport over
Africa and hence for regulating the African monsoon (Raicich et al. 2003, Ziv et al 2004, Lelieveld et al.
2002; Rowell 2003). The northerlies intensify also over the Arabian Peninsula as north westerlies, with wind
anomalies spreading towards the Gulf of Persia and the Indian Ocean. The simulated pattern agrees well with
its observational counterpart, derived by correlating the regional anomalies of omega 500hPa and meridional
wind using the detrended NCEP –DOE data set.  Note that due to a relatively short data set (1979-2015), the
correlations are not significant at the 10% level for some regions and thus not shown, even if they agree well
with those simulated, for example the positive correlations in northwest Africa.
This relationship between mid-level subsidence and the low-level circulation is manifested to some extent in
the regional summer precipitation and outgoing longwave radiation. Figure 9 shows that correlations derived
between the same EOF of omega at 500hPa and precipitation are negative and centered over southern Italy
and the Balkans. This implies that the stronger subsidence over EMED may contribute to precipitation
reductions in these regions, consistent with the intensified zonal pressure gradient and the associated
intensification of the anticyclonic circulation over the western and central parts of the Mediterranean.
However, the summer precipitation in the Mediterranean is in general very low and correlations do not exceed
0.3, implying a rather small effect. On the other hand, the EMED subsidence variations show much stronger
correlations (up to 0.8) with the outgoing long-wave radiation (OLR) over the EMED (Figure 9). This is
consistent with the impact of the adiabatic descent and associated radiative cooling in dry regions under clear
sky conditions.
The modelled connection (Figure 9) between the interannual timescale variations in EMED region and the
northwestern parts of India is consistent with the features of the observed teleconnection with the Indian
summer Monsoon (ISM) (Hoskins et al. 1996, Hoskins et al. 2001, Tyrlis et al.  2012, Ziv 2004, Cherchi et al.
2014).  The variations of the EOF of omega at 500hPa show significant and relatively high correlations with
the three monsoon indices, i.e. subsidence over EMED is negatively correlated with OLR (up to -0.45), and
positively correlated with vertically integrated water vapor (up to 0.6, not shown) and precipitation over the
northwestern parts of India (up to 0.3). This implies a connection between anomalously stronger subsidence
over the EMED region and the intensified Indian summer monsoon. Consistent with this linkage are
intensified southerly winds over the Arabian Sea, which feed the ISM with moisture (Figure 6b, c), and a
deeper heat low spreading west from  India towards the Arabian Peninsula and the Sahel.  The latter
intensifies the zonal pressure gradient over the Mediterranean, which illustrates how the ISM impacts the low-
level circulation in Mediterranean as suggested in Ziv et al. 2004.



These results suggest that CM2.5 is capable of capturing the most prominent features of the summer climate
regime over the EMED: the mid- and upper tropospheric subsidence over the EMED and the low-level
Etesian winds. Moreover, these results also show that the model reproduces their links with the ISM. Changes
in these interactions over the EMED and their teleconnections may give rise to pronounced effects on local
Mediterranean summer temperature regime. Accordingly, the next section investigates the projected future
Mediterranean climate, interpreting this through changes in large-scale circulation as well as through local
relationships and teleconnections.

## 10    4. Climate changes in the 21$^{st}$ century

In this section, we analyze changes in the Mediterranean summer climate as projected by the RCP8.5 scenario.
Five realizations of CM2.5 future projections allow a relatively robust extraction of the climate change signal.
In section 4.1 future changes are presented in the context of the large-scale circulation over the Euro-Atlantic
and South-Asia regions. In section 4.2 we investigate potential changes in SNAO teleconnections, sub-
regional linkages over the EMED region and their relationship with the ISM. We concentrate on differences
between the two periods 2060-2099 and 1960-1999, referred to as the future and the present climate,
respectively. The analysis is performed for the seasonal average of June, July, August (JJA) unless otherwise
specified.

### 21    4.1 Comparison of future and present summer climate

In CM2.5, the projected changes in large-scale circulation features over the Euro-Atlantic region and west
Asia correspond to a large extent to those seen in the CMIP3 and CMIP5 projections. Figure 10 depicts
changes in the summer surface atmospheric circulation, as shown by SLP in JJA and wind vectors estimated
at the original model horizontal resolution (~0.5°). The most prominent feature over the Euro-Atlantic region,
a northward shift and strengthening of the meridional SLP gradient, and hence the meridional circulation cells,
is a typical fingerprint of anthropogenic climate change (Collins et al. 2013). The associated anomalous
bipolar SLP pattern shows negative (cyclonic) SLP anomalies centered over Greenland and positive
(anticyclonic) SLP anomalies centered southwest of the British Isles. This pattern closely resembles the
fingerprint of the CTRL-based SNAO. The projected positive SLP anomaly is however shifted slightly north
eat of the CRTL SNAO, so it is likely that the anthropogenic fingerprint will project on the positive SNAO
phase, quite like as found by Folland et al. 2009 for HadCM3 and HadGEM1.
CM2.5 projections also indicate negative SLP anomalies over North Africa, the eastern Mediterranean and the
Arabian Peninsula, which are accompanied by a very strong warming exceeding 6°C. These changes, together
with an anomalous convergent flow and pronounced anomalies of ascending air at low-and mid-tropospheric
levels (Fig 3c, Figure SI4) over the Arabian Peninsula, contribute to the intensification of the Persian trough.
Although the Persian trough constitutes an extension of the southwest Asian Monsoon heat low, the future
SLP over the monsoon region shows positive anomalies. This indicates a weakening of the monsoon heat low
and, correspondingly, of the northerly flow over the Persian Gulf. It is also worth noting that the anomalies of
ascending motion over the Levant region and the Arabian Peninsula, together with an anomalous subsidence
over the center, north-eastern Mediterranean, and Asia Minor, create a distinct gradient in the vertical velocity
changes. This gradient is particularly evident in July (Figure SI4), due to a magnified intensification of the
heat low and anomalous ascending motion over the Levant region.



The projections also show clear differences, both quantitatively and qualitatively, when compared with the
CMIP5 multi-model ensemble of RCP8.5 projections (Collins et al. 2013) and the CMIP3 ensemble of the
A1B scenario (Giorgi and Lionello 2008). The intensity of the changes over the Euro-Atlantic region shown
by CM2.5 is much larger (Fig 10c) than in the CMIP5 ensemble average (Collins et al. 2013, Fig 12.18), even
when the analysis is conducted at the CMIP resolution. For example, one of the most robust features of the
anthropogenic fingerprint, i.e. anomalous positive SLP anomaly in the vicinity of British Isles, is almost twice
as intense (~up to 3hPa), when analyzed at 2° horizontal resolution. Moreover, in contrast to the CMIP5
ensemble, the projected CM2.5 positive SLP anomalies over the British Isles expands through northwestern
Europe to the southeastern regions: this intensifies the anticyclonic circulation over the western and central
Mediterranean. The CMIP3 ensemble (Giorgi and Lionello et al. 2008) show similar tendencies but with a
much smaller magnitude confined just to northwestern Europe. On the other hand, the CMIP5 ensemble
shows negative SLP anomalies spread over the whole Europe, North Africa and Arabian Peninsula. These
differences between the CM2.5 and the CMIP ensemble yield opposite effects on the Mediterranean surface
circulation. In CM2.5, positive SLP tendencies to the west of the Mediterranean and the strengthening of the
heat low in the eastern Mediterranean, Arabian Peninsula and North Africa foster the strengthening of the
climatological zonal pressure gradient, and the associated northerly winds over the Aegean Sea. In contrast,
the strengthening of the heat low, which expands over the whole Mediterranean, gives rise to a weakening
effect on the regional zonal pressure gradient and the northerly flow in the central and eastern Mediterranean
(Collins et al. 2013, Fig 12.18).
Analysis of future changes in mean surface temperature and precipitation indicates (Fig 10b,c) a less radical
warming and a weaker and less spatially extensive drying over central and southern Europe in CM2.5 when
compared to the CMIP5 ensemble and a high resolution EURO-CORDEX GCM-RCM RCP8.5 multi-model
ensemble (Fussel et al. 2017, Jacob et al. 2014). Furthermore, the general warming gradient between the
northeastern and southwestern parts of Europe in CM2.5 is much sharper and shifted southward when
compared with these ensembles. For example, the minimum of warming (1.5-2°C) in the EURO-CORDEX
ensemble is located over the southern Baltic countries, while in the CM2.5 the minimum expands southward
to the Black Sea coast, creating a strong contrast in the warming between the coastal (4-6°) and inland
Balkans (1°C). It is also worth noting that the temperature gradient depicted in Fig 10b, which expands from
the southeastern Balkans, through the central Europe up to the North Sea coast, marks a transition zone
between the drying of southwestern Europe and the wetting of northeastern Europe (Fig 10c). This transition
zone, analogously to the warming gradient, is much sharper and shifted southward in the CM2.5 projections
when compared with both ensemble averages.
Overall, the CM2.5 indicates a pronounced warming and drying over most of the Mediterranean, although
with a smaller magnitude, compared to the CMIP3 (Dubrowski et al. 2014), CMIP5 (Collins et al. 2013) and
EURO-CORDEX (Jacob et al. 2014, Fussel et al. 2017) ensemble averages. For regions like the Iberian
Peninsula and Asia Minor warming is on average ~2°C lower. The largest difference (more than 3°C) occurs
over the northern parts of Balkans, which feature a local minimum of warming ~1°C and also weak wetting
tendencies. The latter has not been captured in the both CMIP3 and CMIP5 ensemble averages. Both CMIP
ensemble means show a strong warming and robust drying over the whole region and the pronounced
magnitude of these changes has been often linked to the many models' warm summertime bias caused by
deficiencies in representing moisture-temperature feedbacks. Although CMIP3 and CMIP5 models
incorporate much lower horizontal resolutions, compared to CM2.1, some models with relatively high
resolutions (CNRM-CM5 and EC-Earth-DMI, ~1-1.5°, Christensen and Boberg, 2012) featured both
relatively low temperature bias and also relatively mild anthropogenic changes.



Owing to its relatively high resolution, CM2.5 also provides more spatially refined information, which
includes for example sharper gradients along the coasts or in the mountainous regions. All coastal regions
experience reductions in precipitation, expected from the strengthening temperature contrast between the fast
warming land and slower warming sea. These reductions are especially pronounced along the northwestern
coasts of the Iberian Peninsula, where rainfall is typically larger due to incoming North Atlantic storms. The
advantages of the increased horizontal resolution also become apparent for the Levant and the Arabian
Peninsula. This is because the low-resolution CMIP models have difficulties with realistically resolving the
projected zonal gradient between oppositely signed rainfall tendencies (i.e. drying over the eastern
Mediterranean and wetting on the western flanks of the monsoon). This results in a much smaller changes and
statistical insignificance.
Finally, the projected magnitude of the strong warming and drying in the Mediterranean shown by CMIP3 and
CMIP5 has often been questioned, due to the deficiencies in representing land-atmosphere feedbacks and a
weak sensitivity to atmospheric teleconnections such as the SNAO. For example, Blade et al. 2012 argued that
the drying projected in most CMIP3 and CMIP5 models is overestimated due to a deficiency in capturing the
SNAO teleconnection and its cooling and wetting effects over the Mediterranean associated with a future
more positive SNAO. Here we have shown that CM2.5 is capable of capturing the SNAO teleconnection
(section 3) even though the regional impacts are almost the same as those indicated by the low-resolution
CM2.1 model. However projected CM2.5 changes in large scale circulation including the more positive
SNAO are more intense compared with most CMIP5 models.
**4.2 Future changes in SNAO- Mediterranean teleconnections**
In this section we investigate the future evolution of the SNAO and its impact on the Mediterranean
hydroclimate. For this purpose we analyze time series of the SNAO principal component for 1860-2100. The
time series were derived by projecting the historical and future ensemble-runs (240 seasons) onto the CTRL-
based SNAO eigenvector. However to account for possible changes in the spatial pattern of the future SNAO
pattern and its impacts, we utilize eigenvectors derived separately for a six decades long period in the
historical runs (1950-2010) and in the future projections (2040-2100) (see Data and Methods). The last part
of this section shows that a contribution of future SNAO changes to the projected changes in the
Mediterranean precipitation described in section 4.1.
Figure 11c depicts the temporal evolution of the SNAO across all five HIST and PROJ realizations. In
agreement with the previous modeling studies (Collins et al. 2013, Folland 2009), CM2.5 reveals weak
positive SNAO tendencies in the latter half of the 20[th] century and a strong positive trend during the 21[st]
century. In the presence of strong modelled internal SNAO variations, the long-term tendencies during the
former period are discernible only in the ensemble average. This is consistent with recent observations of
SNAO featuring a rich interannual to multi-decadal variability and a weak positive trend described in section
3.2.1. However, Figure 11c indicates also that for 2040-2100 the trend becomes strong enough to be
discernible in every realization.
A possible evolution of the future SNAO pattern is shown by carrying out the associated EOF analysis
separately for each ensemble-member for 1960-2010 and 2050-2100. This suggests subtle changes in the



future spatial pattern of the SNAO with a northeastward movement of the SLP dipole consistent with the
projected intensification and northeastward shift of the meridional SLP gradient over the North Atlantic
(Figure 10a). These changes in the SNAO pattern are reflected in teleconnections with European hydroclimate.
Figure 11a, b shows ensemble average correlations between the time series of the SNAO principal component
and precipitation anomalies for 1960-2010 and 2050-2100. This indicates an increase and northeastward
expansion of SNAO impacts. Correlations associated with the center of the southern SNAO lobe are shifted
closer to the British Isles indicating a stronger influence on Euro-Atlantic precipitation with stronger drying
during the enhanced positive SNAO phase in 2050-2100. Negative correlations also show a slight increase
over southern Europe, particularly for the Iberian Peninsula, southern Balkans and Asia Minor, indicating
stronger moistening of the Mediterranean in 2050-2100.
A key implication of these results is that the long-term tendency towards a positive SNAO may offset the
projected drying in the Mediterranean. We explore this further focusing on the contribution of the SNAO to
the projected changes in Mediterranean precipitation between 1961-1999 and 2061-2099. The impact of
SNAO on the regional precipitation is quantified using a linear regression method (see Data and Methods)
estimated from the 1000-years CTRL run. Figure 11e shows that the largest effect of future SNAO changes is
projected to be over the Iberian Peninsula, Italy, Balkan coast, and Asia Minor, in agreement with correlation
maps in Figure 5c and Figure 11a, b. Fig 11e clearly indicates future wetting over these regions due to the
changed SNAO, reaching locally up to 0.3 mm/day over 2061-2099. These are significant numbers; for some
regions (i.e. Iberian Peninsula) their magnitude is comparable with the magnitude of projected drying.
Therefore, a comparison (Figure 11d, e) of projected total future precipitation change and the change with the
SNAO contribution removed shows for the latter a much stronger drying for most of the Mediterranean land
areas. For example, average drying over the southeast and central Iberian Peninsula over 2061-2099
intensifies from ~ 0.4 to 0.65 mm/day, and for the Balkan coast from ~0.3 to 0.55mm/day. This confirms a
projected important role of future changes in the SNAO for offsetting the projected Mediterranean drying.
Moreover, given that projected correlations between the SNAO and the Mediterranean precipitation shows a
slight increase in the future (compare Figure 5c with 11a, b), the offsetting impact of a strengthening positive
SNAO might be slightly higher than that shown here.
This analysis confirms the results of Blade et al. 2012 in showing an important offsetting effect of the SNAO
teleconnection on European hydroclimate. However, the authors also argued that the low capability of CMIP5
models (e.g. CM2.1) in simulating the SNAO teleconnection is the major reason for the projected southern
Europe strong drying. This statement is not fully consistent with our findings. The projected CM2.5 changes
in precipitation over Europe are very different from those of CM2.1 (Blade et al. 2012), CMIP3 and CMIP5
ensembles (Collins et al. 2013), despite the fact that the impact of SNAO (in terms of pattern and magnitude)
is for CM2.5 and CM2.1 almost the same. For example, CM2.5 shows generally weaker drying over the
Mediterranean than the CM2.1 and CMIP3 ensembles. Also, CM2.5 shows wetting over the northern Balkans
in contrast to drying in CM2.1 and CMIP3/CMIP5 ensembles, despite the fact that both GFDL models show
very small or no SNAO influence in this region. Moreover, CM2.5 indicates future wetting over most of the
central Europe, in contrast to the drying in both CMIP ensembles, suggesting an impact of other factors.
Associated with these results is a much smaller surface warming in CM2.5 in parts of eastern and central
Europe (green area in Fig 11b), due to a stronger influx of cool and moist air from the North Sea (Figure 10a,
b, c). The warming is also less over southern and Europe and the Mediterranean than in CMIP5 (Collins et al.
44   2013).






**4.3 Future changes in other aspects of the summer climate in the EMED region**

This section investigates other projected CM2.5 climate changes in the EMED region, including low-tropospheric circulation and mid- and upper tropospheric subsidence, the main factors that maintain the regional temperature balance. The future large –scale circulation changes over the Mediterranean have been shown in section 4.1 (Figure 10a,b). The projected future intensification of the Etesians as well as the weakening subsidence should yield a cooling and wetting effect on that region. Nevertheless, future projections indicate a very strong warming and drying in this region. This suggests a decreasing influence of the atmospheric dynamics on the temperature balance of this region and an increasing impact of surface warming on surface circulation. The latter is manifested in the intensification of the heat low over Sahara, EMED and Arabian Peninsula, accompanied by anomalous surface convergence and ascending air centered over the maximum warming—the Levant and the Arabian Peninsula.

In the following, we investigate future changes in these factors governing the summer EMED regime by analyzing the stationarity of the local linkages and teleconnections by comparing their representation in the HIST and PROJ experiments. We also compare coupling of the EMED regime with the Indian summer Monsoon and explore the impact of the local surface warming on the regional surface circulation.

**4.3.1 Changes in the local and remote relationships with the EMED climate regime**

We first compare the relationship between the mid- and upper-tropospheric subsidence and the low-level circulation over EMED in July derived for two 50-year periods i.e. 1960-2010 and 2050-2100. We use correlations between time series of the first EOF of omega at 200hPa over the EMED and time series of SLP, meridional wind, precipitation and column integrated water vapor fields as in section 3. The correlations are derived separately for each member of the HIST ensemble (1960-2010) and of the PROJ ensemble (2050-2100) followed by averages of the ensemble members. As the sign of the EOFs is arbitrary, we ensured that all EOF time series represent the same EOF and have the same sign.

Figure 12 compares the HIST and PROJ five-member ensemble mean correlations, showing substantial qualitative and quantitative differences between the historical and the future periods. The linkage between the EMED subsidence and surface circulation, in terms of pressure, northerly winds over the Aegean Sea (Etesians), as well as their extension towards North Africa and Persian Gulf, is much weaker for 2050-2100. For example, correlations between the EMED subsidence and the northerlies decrease locally (e.g. Persian Gulf) by more than a factor of two (from ~0.7 to ~0.3) between 1960-2010 and 2050-2100. This is consistent with radically decreased correlations with water vapor and precipitation (from ~0.4-0.5 to ~0) over the African monsoon region, which is largely fed by moisture transported by the Etesians.

At the same time, correlations between the EMED subsidence and ISM indices (July), i.e. precipitation and column integrated water vapor (Fig 12e, f), does not change much. In both cases correlations reach 0.5 for precipitation, and 0.6 for the water vapor. For the latter, the maximum correlations only decrease from 0.7 to 0.6. Both patterns are slightly shifted toward the southern parts of India in the future period, consistent with a southward shift of the southerlies over the Arabian Sea that act as a moisture supply for the ISM monsoon.

These results imply generally insignificant changes in the future mid- or upper-tropospheric teleconnection between the EMED and ISM regions. However they do suggest a pronounced weakening of the local linkage between the mid- and upper-tropospheric subsidence and surface circulation over the EMED. Moreover, given



that the local linkage serves as a "medium path" for the teleconnection between the ISM and surface
circulation over EMED, future weakening of the local linkage will most likely diminish the impact of this
teleconnection on the EMED surface circulation. On the other hand, the projected intensification of the heat
low over North Africa, EMED and Middle East suggests a growing contribution of surface temperature to the
summer climate regime in those regions. Thus in the following section we explore apparent nonlinearities in
the summer climate regime of the eastern Mediterranean associated with the local surface temperature.
**4.3.2 Nonlinear dependency of the local interrelations between the low-tropospheric and the mid-**
**tropospheric dynamics and their contributions to the thermal balance over EMED.**
The nonlinear dynamical influences over EMED can be explored using the CTRL in order to differentiate
impacts of the local (EMED) temperature on the local linkage between the mid- and upper tropospheric
subsidence and surface circulation. Removing the time-varying climate forcing impacts from the HIST runs
allows us to focus on the natural variability of the system and nonlinear interactions that may be difficult to
statistically calculate in shorter HIST runs. We use two samples from the 1000yr CTRL run, each of which
represent the Mediterranean summer climate (July) for 300 CTRL months with the lowest temperature and
300 with the highest temperature, based on the mean surface temperature for the EMED region (30°-36°N,
36°-42°E). We carry out a correlation analysis for these two periods, much as done in the previous section
comparing recent historical and future periods. Figure SI1 shows for the sample with warmer seasons over
EMED a radical drop in derived correlations between the mid-level subsidence and the Mediterranean
pressure, Etesian winds and their extension over the North Africa and Persian Gulf, and water vapor over the
Sahel. These results are similar to those comparing present and future climate for these variables (Fig 12).
To extract a direct response of the summer Mediterranean climate to the surface warming over EMED, Figure
13 shows composite differences between the 300-season warm and cold samples for July for the EMED
region for temperature, relative humidity, pressure and wind vector, geopotential height at 500hPa and 800hPa,
omega at 500hPa and precipitation. Interestingly, Fig 13a shows maximum of warming over Asia Minor and
not the Levant, suggesting a potential feedback between these regions. Fig 13c shows bipolar SLP anomalies,
with low pressure coincident with intensified heat low anomalies over North Africa, EMED and Middle East,
and an anomalous anticyclonic circulation between northeast and northwest of this region. The latter is
centered over the Black Sea and spreads towards the central Mediterranean, creating a stronger zonal pressure
gradient over the Mediterranean and intensified Etesian winds. The intensified heat low over the EMED and
Arabian Peninsula (Fig 13c) agree well with the enhanced local convergence and reduced subsidence at the
low- and mid-tropospheric levels at 500hPa (Fig 13e) and 700hPa (not shown). At the same time, the positive
SLP anomalies (Fig 13c) and increased subsidence over Asia Minor and Black Sea are physically consistent
with increased adiabatic warming and stability, manifested in the local maximum warming, reduced relative
humidity and precipitation. An analysis for the JJA season yields similar results, although with a reduced
magnitude (Figure SI2) due to a smaller contribution of June and August months.
Analysis of the effects of local warming over larger domains, i.e. including southern parts of the center and
western Mediterranean (Figure SI3) yields similar results, i.e. opposite SLP anomalies for the southern
Mediterranean (North Africa and the EMED) and northern central and eastern Mediterranean. However, the
magnitude of the response over southwestern regions becomes equally strong to that over the EMED.
Interestingly, analyses for regions which include the northeastern parts of the Mediterranean (30°-50°E, 30°-
45°N), yields a reduced response (a weaker intensifying anticyclone) over these northern regions. Thus the
derived anomalous anticyclone over Balkans and Black Sea is much stronger when the warming is confined to





the southeastern parts (i.e. Levant and Arabian Peninsula). This emphasizes the role of the warming surface temperature over arid regions of North Africa and Arabian Peninsula in shaping the climate of the surrounding regions.

The composite response to anomalous warming over EMED can be readily associated with changes in the local circulation, i.e. an intensified zonal pressure gradient and Etesians over Mediterranean, as well as an intensified heat low, anomalous convergence and weakening low- and mid-tropospheric subsidence over the EMED and the Arabian Peninsula. All these changes are also seen in the future projections obtained for JJA (Figure 10a,b,c, Figure SI4a,c,e), and particularly in the month of July (Figure SI4b,d,f). Moreover, the response of positive SLP, increased subsidence and drying over central Mediterranean (i.e. Italy), the Balkan coast and Asia Minor are also consistent with the future changes (Figure 10, Figure SI2), suggesting a potential contribution of the warming over Levant and the Arabian Peninsula to projected warming and drying over Asia Minor.

This analysis indicates that the dynamical regime over the EMED has a nonlinear influence on local temperature. During relatively cool years the dynamical relationship between the low-level circulation and mid-level subsidence, which balances the temperature over EMED, seems to be much stronger. By contrast, warming over EMED region can trigger a local response in the surface atmospheric circulation, which weakens the local dynamical linkages and hence their contribution in maintaining the local temperature balance. Hence it is possible that surface temperature-driven atmospheric responses will become a more prominent factor shaping future Mediterranean climate. This idea is supported by the consistency of this response, i.e. strong warming, intensifying heat low, anomalous convergence and very pronounced ascending motion at the low and mid-levels of the EMED and Arabian Peninsula, intensified zonal pressure gradient and Etesians, drying over Asia Minor and southern Balkans; with projected anthropogenic changes over the EMED region.

The analysis however does not explain the processes involved in the dipole-like response in the circulation, which includes SLP, winds and omega anomalies north from the EMED region (particularly Asia Minor and Black Sea). One might suspect that, in response to warming over the EMED, the anomalous convergence and ascending motion over the EMED triggers a seesaw connection with northward-located regions. This link could stem from the interactions of the anomalous warming and upward velocity anomalies with the seasonally varying descending branch of the Hadley Cell over EMED, in result expanding it towards Asia Minor. Similar results are obtained for June and August, although with different intensity and location of the SLP and precipitation anomalies. However testing this hypothesis should be done in the future using more elaborate analysis.

**5. Summary and Discussion**

Based on the state of the art future projections (CMIP3 and CMIP5-generation) the Mediterranean has been identified as a climate change hot spot (Giorgi and Lionello 2008), not only due to the sensitivity of its climate to the anthropogenic forcing, but also due to the socio-economic vulnerability of the local societies. Yet the projected changes are not fully reflected in the observations for the second half of the 20[th] century. While the derived anthropogenic fingerprint clearly suggests a strong warming and drying during the summer, the observations indicate opposite wetting tendencies for some regions—in the vicinity of Black Sea and off the Balkan coast. This discrepancy may stem from the fact that the Mediterranean climate features abundant cross-scale variations, which at present dominate the anthropogenic signal. But there can be other reasons for



this inconsistency, i.e. the deficiencies of models in capturing interactions between the atmosphere and the land-surface or the deficiencies in simulating the SNAO teleconnection. The former has been shown to cause an overestimation of the projected future summer warming and drying in most of CMIP3 and CMIP5 models (Christensen and Boberg, 2012, Christensen and Boberg, 2012, Mueller and Seneviratne, 2014), particularly in the Mediterranean, Central and Southeast Europe (Diffenbaugh et al. 2007, Hirschi et al. 2011, Seneviratne et al., 2006). The latter has been suggested to incapacitate CMIP3/CMIP5 models in offsetting projected future regional drying, and hence spuriously exaggerate the regional warming and drying (Blade et al. 2012).

Overall, simulating the climate of the Mediterranean remains challenging, taking into account the complex orography of that region as well as the influence of different teleconnections originating either from the mid-latitudes or the subtropics. Thus realistic future projections for this region require not only refined spatial scales, but also a realistic balance between the contributing impacts of local land-atmosphere interactions, large-scale circulation, and teleconnections.

This study utilizes the high-resolution CM2.5 climate model integrations to analyze the projected regional future changes and interpret them through the prism of simulated SNAO teleconnections, the local dynamical regime, and the impact of warming land surface.

In the first part of the analysis we use the 1000-year control run, with atmospheric forcing fixed at preindustrial times, to examine its capability to realistically simulate summer climatology of the Mediterranean. This analysis shows that the model very accurately reproduces key features of the regional regime, i.e. the location and magnitude of the atmospheric circulation, including the subtropical mid-tropospheric anticyclone between the Levant and South Asia, and the low-tropospheric zonal pressure gradient between the subtropical North Atlantic anticyclone and the massive Asian monsoon heat low. The high spatial resolution of the integrations allows to faithfully capture the adjustments of the low-level atmospheric flow to the local orography, manifested in two branches of Etesian winds: over the Aegan Sea and its southward extension toward the Sahel region, as well as over the Persian Gulf. Also the mean precipitation, which features an exceptional spatial complexity in the Mediterranean, is represented with a much higher degree of realism when compared with the low-resolution CM2.1.

The analysis has confirmed the capability of CM2.5 to simulate the regional teleconnection. The model faithfully reproduces the fingerprint of the summer atmospheric variability dominating the North Atlantic (SNAO), the associated physical mechanism, and its influence on the Mediterranean, when compared with 20[th] century observations (Folland et al. 2009). Remarkably, the simulated SNAO pattern is in better agreement with the observations before the 1970s/1980s. This may stem from to the fact that in the most recent observations the strong multi-decadal signal of the SNAO (as highlighted by Linderholm and Folland 2017) may have a contribution from anthropogenic forcing, which is not included in the CM2.5 control run. The simulated SNAO teleconnection yields (Folland et al. 2009) an influence on the Mediterranean precipitation comparable with observations and CMIP3/CMIP5 simulations, e.g. as manifested in increased precipitation over the western, central and northeastern parts of the region during the positive SNAO phase.

The study indicates also that the model skillfully reproduces the most prominent features of the eastern Mediterranean summer climate regime: the low-level northerly flow and the mid- and upper -tropospheric subsidence. The strong linkage between these two factors demonstrates their counterbalancing effects on the regional temperature. Additionally, the significant correlations between the mid- and upper tropospheric subsidence over the Mediterranean and the indices of the Indian monsoon are consistent with the Indian summer monsoon teleconnection described in Rodwell and Hoskins, 1996, and Tyrlis et al., 2013.



Overall, the long control run with fixed forcing allows us to show the important role of the regional SNAO
teleconnections (i.e. precipitation over northern Mediterranean), and of the local linkage between the surface
and upper-level dynamics (i.e. temperature balance over the eastern Mediterranean) in the Mediterranean
summer regime. The future evolution of these two factors is of primary importance for projections of the
regional summer climate.
The analysis of the CM2.5 five-member projections using the RCP8.5 scenario indicates pronounced large-
scale circulation changes, which are in general agreement with the CMIP5 counterpart. These changes, an
expansion of the Hadley cell, an intensification and a northward shift of the atmospheric meridional cells over
the North Atlantic, manifested by an anomalous anticyclone southwest of British Isles, constitute a typical
fingerprint of the future summer anthropogenic change.
On the other hand, the CM2.5 future changes projected over most of Europe and western Asia are remarkably
different from the CMIP5 ensemble, which yields fundamental differences in projected Mediterranean climate.
CM2.5 projections show higher sea level pressure expanding from an anomalous anticyclone over the North
Atlantic across the northwestern and southeastern parts of Europe and an intensification of the heat low over
Sahara, the Persian trough and Asia Minor. In contrast, the intensified heat low in the CMIP5 ensemble covers
not only arid subtropical regions but spreads over most of Eurasia (except the southern part of the monsoon
region), suggestive of the influences of surface warming. As a consequence, CM2.5 projections indicate a
strengthening of the zonal pressure gradient over the Mediterranean and the associated northerly flow/Etesian
winds, contrary to a rather small weakening of the flow projected in CMIP5.
The general drying of the subtropics projected in CM2.5 and CMIP3/CMIP5 is consistent with the
thermodynamic response of the warming atmosphere, i.e. wet-get-wetter and dry-get-drier (Held and Soden,
2006; Seager et al., 2007). However, the CM2.5 projections indicate a much smaller magnitude of changes
and also a more complex pattern for Mediterranean precipitation, including wetting anomalies in the Alps, the
northern parts of the Balkans, Central and Southeast Europe. The weaker magnitude of warming and drying
(or even wetting) projected in CM2.5, particularly over the Iberian Peninsula, coastal regions, and Asia Minor,
could be to some extent associated with the future changes of SNAO. The projected northeastward shift of
SNAO and strengthening towards its positive phase manifests in the positive anomalies of precipitation over
the northwestern, central and northeastern parts of the Mediterranean, and hence counterbalances the
thermodynamic effects of subtropical drying. For example, without the impact of SNAO, the projected drying
over some regions in the Iberian Peninsula, Italy, and the Balkan coast would be much stronger (locally up to
~30-40%). Nevertheless, the representation of the regional impact of the SNAO teleconnection, as well as its
future changes projected in CM2.5, agrees well with some older generations models such as CM2.1, both in
terms of pattern and magnitude. Thus SNAO does not seem to be a strong candidate explaining the stark
contrast between the future projections for the summer European climate between CM2.5 and CMIP3/CMIP5
projections.
Additionally, the SNAO teleconnection has been shown to be of minor importance (in models and
observations) for Central and Southeastern Europe and thus it cannot explain the wetting anomalies projected
in these regions (particularly the northern Balkans). Many studies have suggested rather the soil moisture-
climate feedbacks to be of dominant importance for summer climate in these regions (Seneviratne et al. 2006,
Diffenbaugh 2007, Hirschi et al 2011). At the same time, most of CMIP3 and CMIP5 models are deficient in
capturing soil moisture-atmosphere feedbacks (Christensen and Boberg, 2012; Mueller and Seneviratne, 2014,



Boberg and Christensen, 2012). This causes a temperature dependent bias and spuriously amplifies the projected regional warming and drying, manifested ultimately in the CMIP5 ensemble mean as a very strong overall warming and drying over Europe. Hence the apparent stark contrast between the CMIP3/CMIP5 and CM2.5 (mild warming and wetting) regional projections could more likely originate from the enhancements in the land model incorporated to CM2.5 at high spatial resolution, rather than the impacts of the SNAO.

Additionally, the changes in climate regime projected in CM2.5 also suggest a weakening role of atmospheric dynamics in maintaining the regional hydroclimate and temperature balance over the eastern Mediterranean. For example, the projected changes over Asia Minor show very strong drying and warming, despite the increasing influence (i.e. wetting and cooling) of the SNAO teleconnection. Also, the warming over the Asia Minor and Levant regions constitutes a local maximum for the changes in the Mediterranean region, despite the cooling effect of strongly intensifying Etesians and weakening mid- and upper-tropospheric subsidence in EMED.

The derived weakening linkage between the low-level circulation (e.g. northerly flow) and the mid-and upper-level subsidence over the EMED, which balances the regional temperature, is indicative of the nonlinearity in the summer EMED regime introduced by the local surface temperature. Additionally, the comparison of warmer and cooler EMED seasons in the control run indicates an increasing influence of local surface temperature on the local low-level atmospheric circulation in a warmer climate. This response stems likely from a sensitivity of this desert-like land to radiative forcing. It is manifest in an anomalous intensification of the heat low over the EMED, Sahara and the Persian trough, but also in positive SLP anomalies, increasing subsidence, maximum warming, reduced relative humidity, and reduced precipitation in the regions located north from Levant, in particular Asia Minor. All of these features are consistent with future climate changes projected by CM2.5. This supports the overall notion of an increasing impact of the interactions between land surface and atmosphere, as well as a decreasing influence of atmospheric dynamics, and hence of teleconnections, on the summer regime in this region.

**"The authors declare that they have no conflict of interest."**

### Acknowledgements
The authors are grateful to Ileana Blade, Fred Kucharski and Eduardo Zorita and Alex Petrescu for helpful comments and discussions.





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



Table1 Abbreviation names for the CM2.5 runs

| NAME of the experiment | Ensemble size | Number of years total | Historical period [yrs] |
|---|---|---|---|
| CNTR | 1 | 1000 yrs | - |
| HIST | 5 | 145 | 1861-2005 |
| FUT | 5 | 95 | 2006-2100 |






**FIGURES**
**Figure 1. Seasonal (JJA) sea level pressure (hPa) and wind vector at 850hPa (m/s) in a) NCEP-DOE2, b) CM2.5.**
**Time-mean of seasonal data from years 101–1000 of the control simulations are used, and years 1979-2017 of the**
**observed data.**

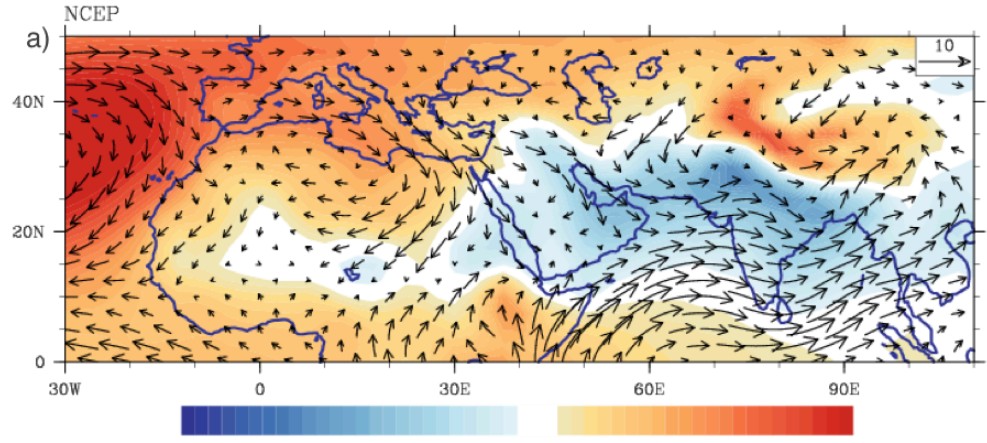

6 .

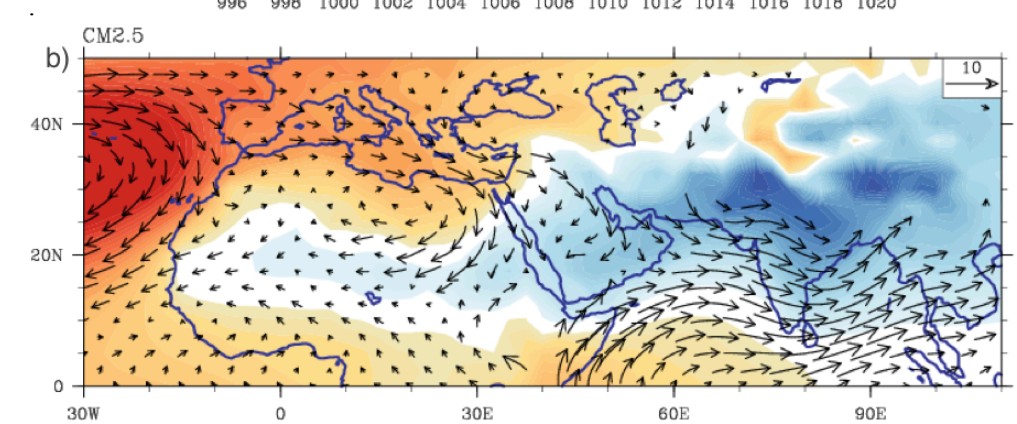



**Figure 2. Seasonal (July) time-mean vertical velocities at 500hPa (Pa/s) and wind vectors at 200hPa (Pa/s),**
**estimated for a) NCEP-DOE2 in the period 1979-2017, and b) CM2.5 CTRL run in years 101-1000. Both data**
**sets are interpolated to the 2.5° horizontal grid.**

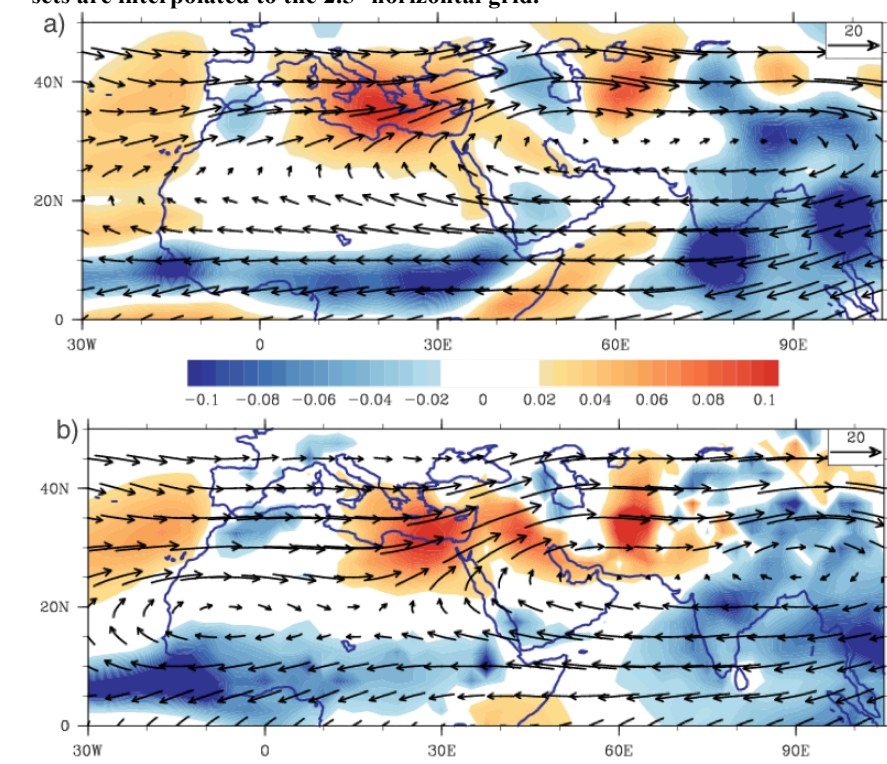






**Figure 3. Height (pressure)-longitude cross-section of vertical velocity (Pa/s, shaded contours, downward motion denoted with positive values) and vector of zonal wind (m/s) and vertical velocity (converted to m/s and scaled with a factor of 1000) in July. Figure shows time-mean values in July a) derived for the period 1979-2017 in NCEP-DOE2, b) derived from 101-1000 years of CNTL run in CM2.5; and c) projected future changes in the period 2061-2099 in PROJ ensemble mean, compared with the baseline period 1961-1999 in the HIST ensemble mean. All fields are shown on the 2.5°x2.5° horizontal grid and at the original vertical levels, common for CM2.5 and NCEP-DOE2.**

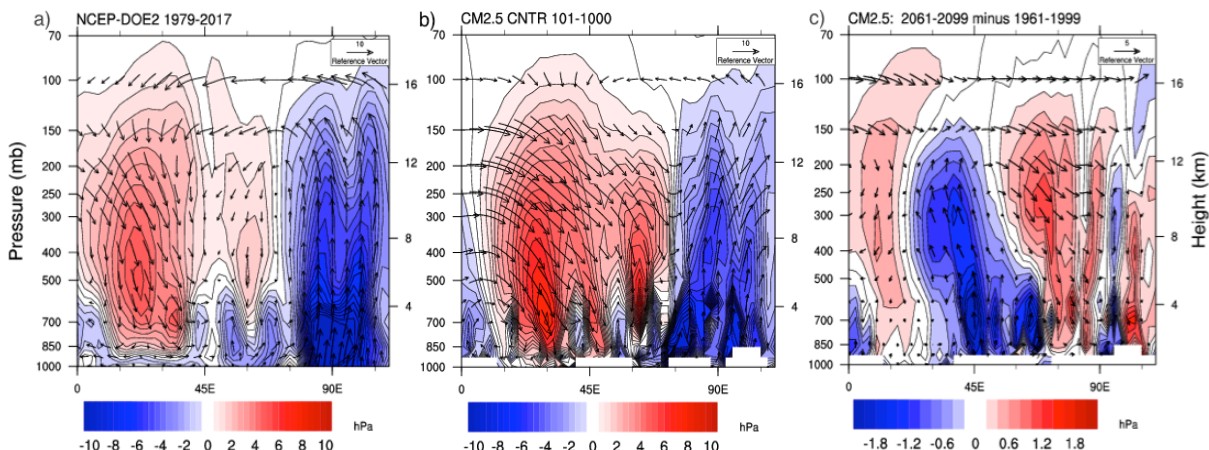

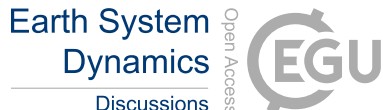

**Figure 4. Seasonal (JJA) mean precipitation (mm/day) for a) University of Delaware Climatology, b) CM2.1, c) CM2.5. The time-mean of seasonal data from years 101–1000 of the control simulations are used, and years 1900-2010 of the observed data.**

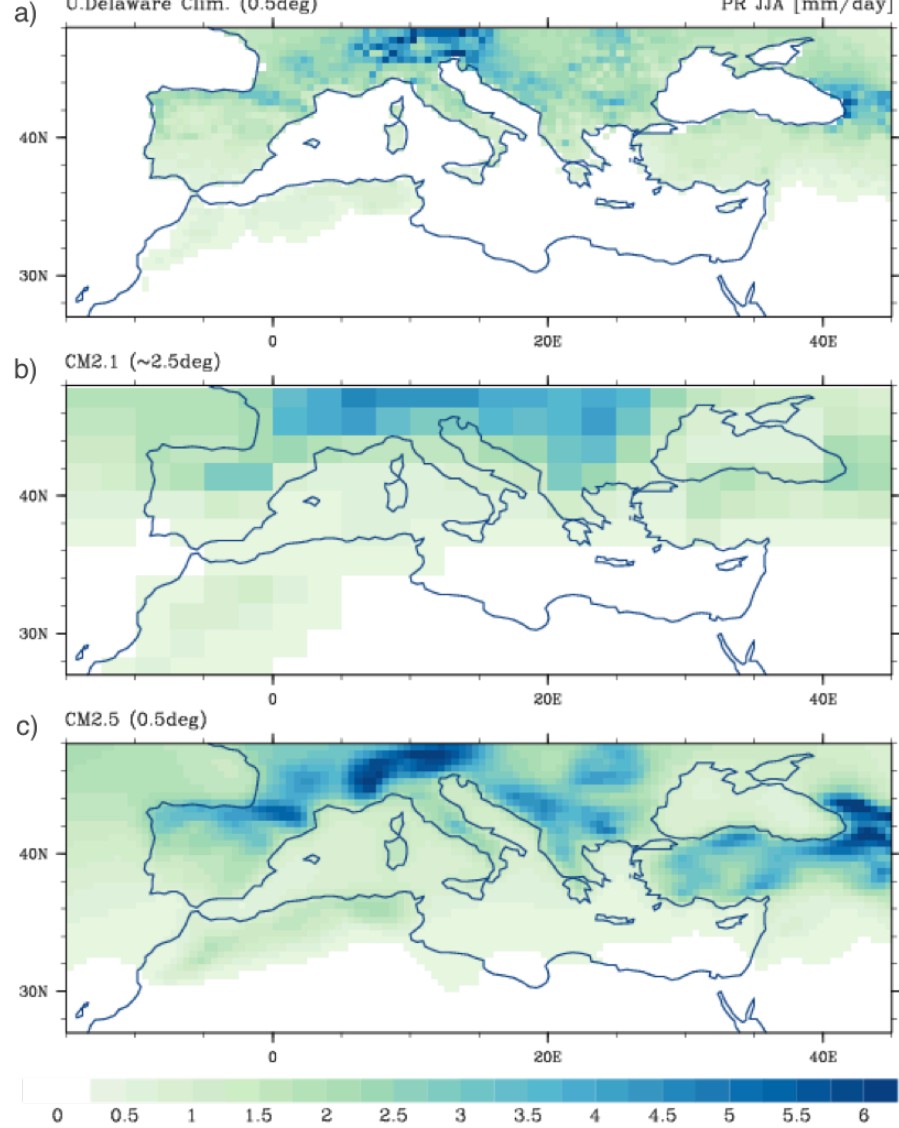





**Figure 5. Correlation between principal component time series of the SNAO SLP in JA and a) sea level pressure**
**b) temperature at 2m (shaded) and geopotential height at 850hPa (contours), c) precipitation. All derived from**
**the CTRL run. Contours in a) and c) are shown for 0.25 and 0.5 correlations. Correlations are shown only when**
**significant at 1% level.**

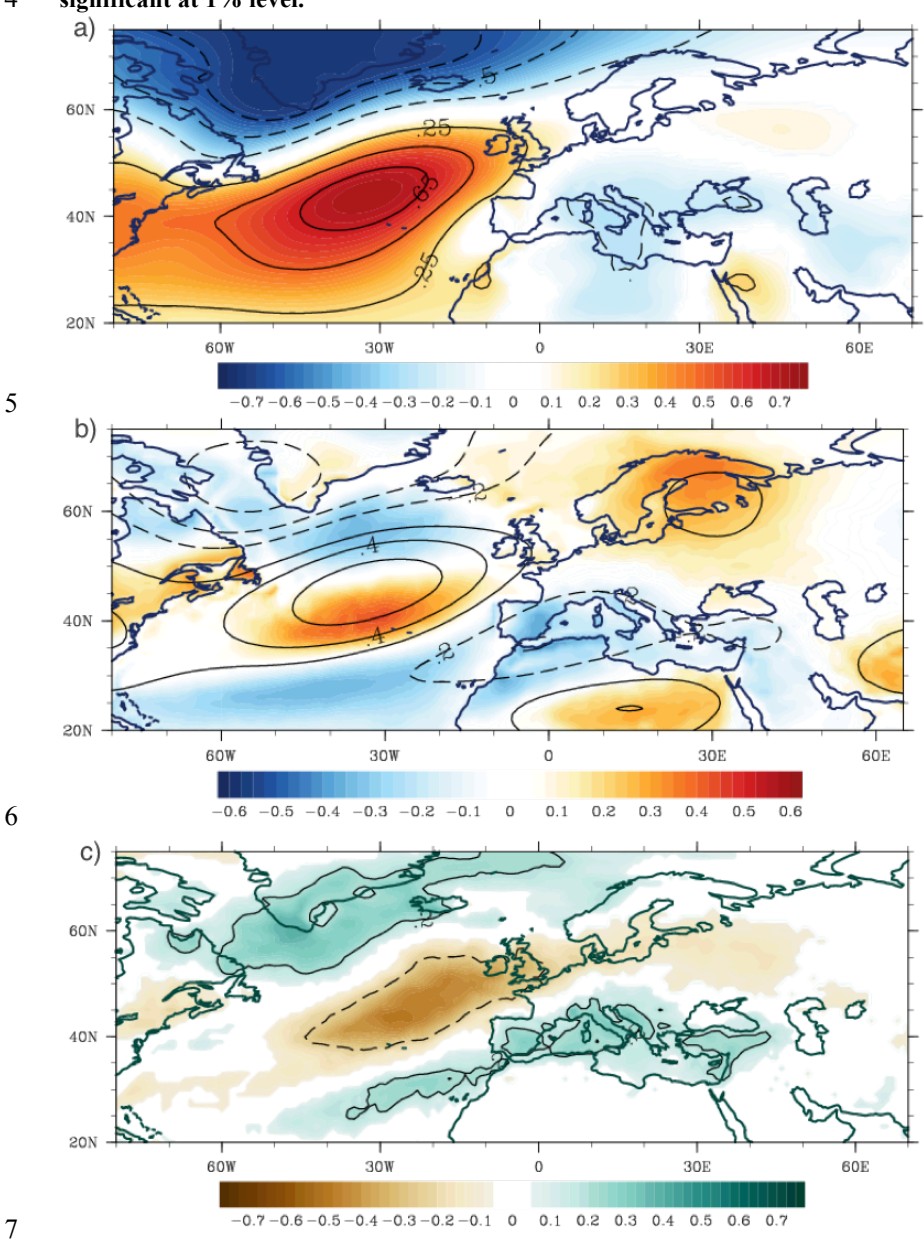





1  **Figure 6. Spatial pattern of the SNAO, derived from the 20CR reanalysis (left), and from the first CM2.5 HIST**
2  **run (right), shown as correlations between the first principal component time series and SLP in July-August.**
3  **The pattern is derived from periods a)-b) 1870-1920, c)-d) 1900-1950, e)-f) 1940-1990, g)-h) 1960-2010.**






**Figure 7. (left) Mean storm track patterns for 1950–1990 (top) and 1970-2011 (bottom), represented by the**
**standard deviation of daily 300 hPa geopotential height (m) in July–August. The data was bandpass-filtered on**
**time scales of 2–8 days. (right) Correlations derived between the standard deviation of the filtered seasonal (July-**
**August) values of 300-hPa height (derived from NCEP/NCAR 1 data) and the SNAO index (derived from 20CR**
**data).**

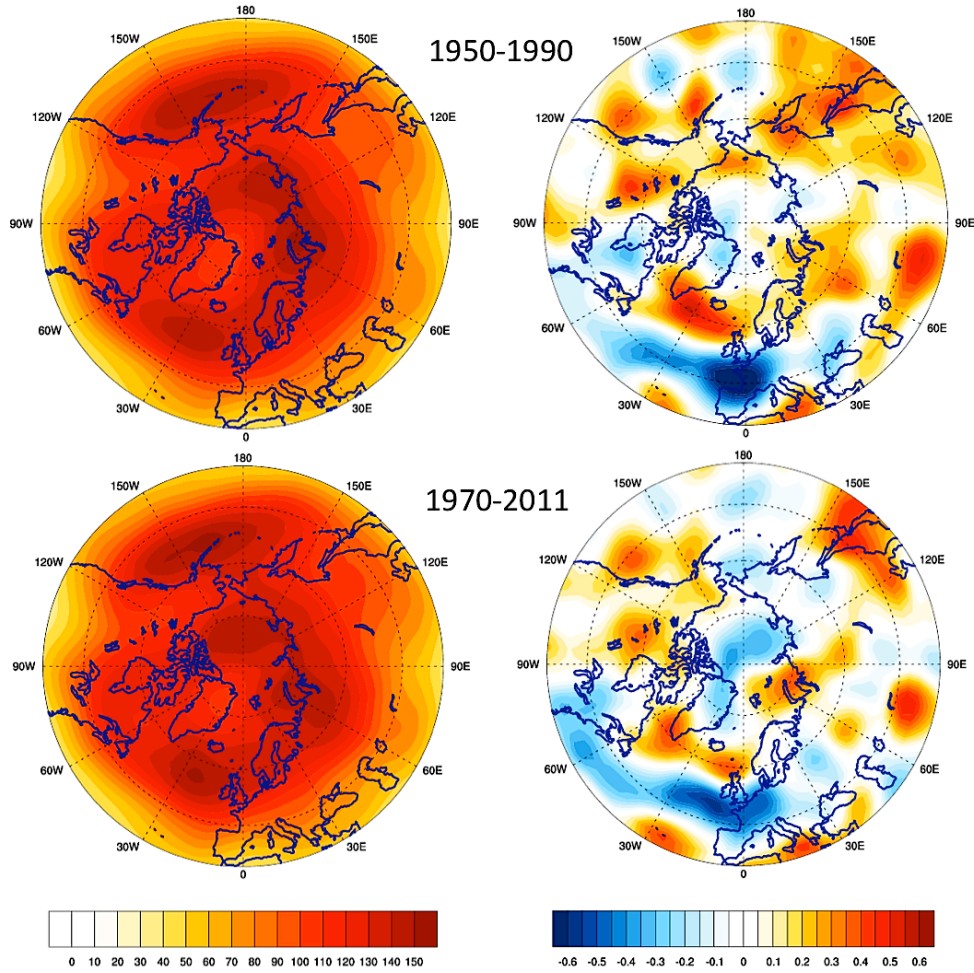






**Figure 8. a) First EOF of CM2.5 vertical velocities at 500 hPa (EOF1 omega, shaded) and 300hPa (contours), derived from the monthly mean of July in the CTRL run. The time series of EOF1 omega are correlated with b) geopotential height (shaded), u, v components (shown as vector) at 850hPa and c) meridional wind at 850hPa. d) Correlations derived between the observed (NCEP) omega 500hPa over the eastern Mediterranean region (32°-34°N, 25°-30°E) and the meridional wind at 850hPa. Correlations shown are at b)-c) the 1%, and d) the 10% significance level.**

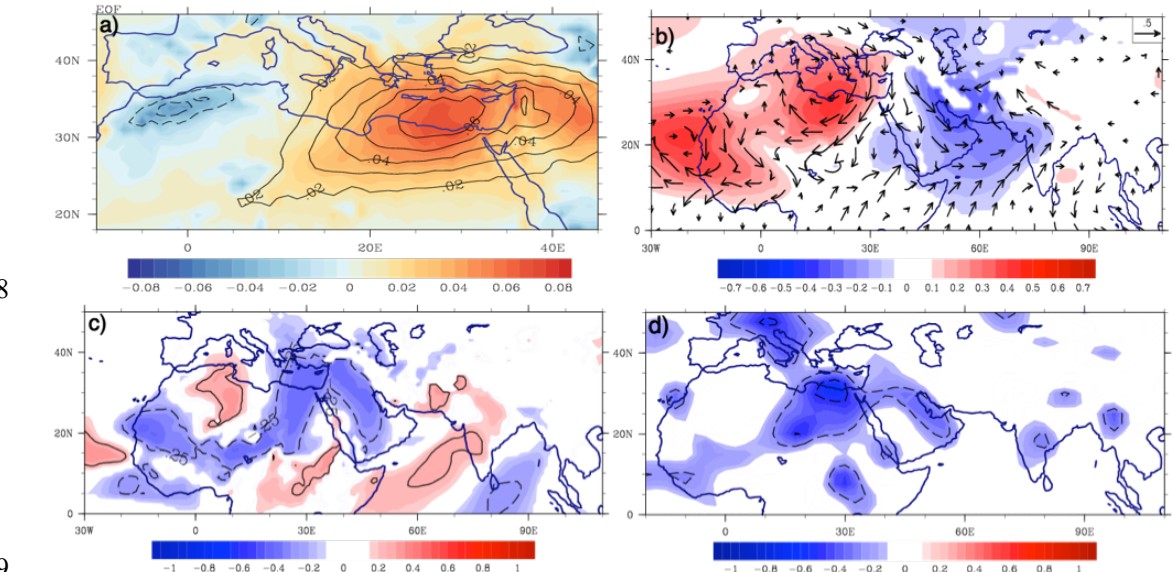



**Figure 9. Correlations between the time series of the EOF1 omega (derived for omega 500hPa, see Figure 7a) and a) outgoing long wave radiation (shaded), omega at 500 hPa (contours: -0.2, 0.2, 0.4), b) precipitation; derived from the monthly means of July in CTRL run. Correlations shown are at the 1% significance level.**

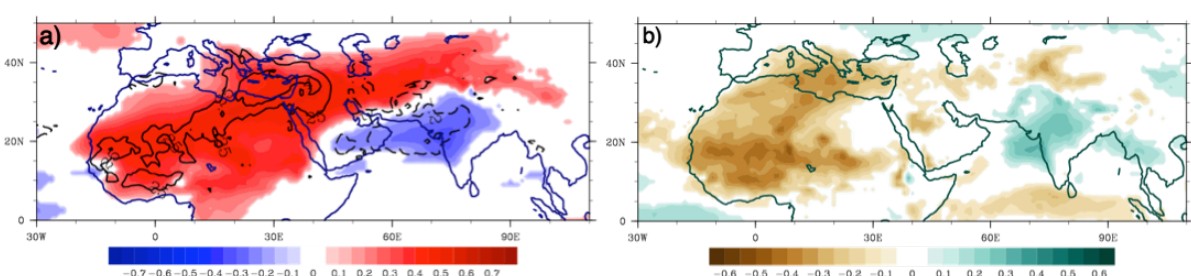





1 **Figure 10. Projected future changes for the summer (JJA) a) sea level pressure (hPa, shaded) and u,v wind**
2 **components (m/s, vector), b) surface temperature (°C), c) total precipitation rate [mm/day], over the period**
3 **2061-2099 compared with the baseline period 1961-1999. Changes are derived at the original horizontal**
4 **resolution (~0.25°).**






**Figure 11. Correlations between the principal component time series of SNAO SLP and precipitation in the**
**summer period a) 1900-1950 (HIST runs), b) and 2050-2100 (PROJ runs). Correlations are shaded with contours**
**denoting 0.25 and 0.5. Figure c) shows combined time series of the principal component time series of SNAO SLP**
**for individual HIST and PROJ runs (blue) and the ensemble mean (red). Figure d) shows projected changes in**
**the summer precipitation (mm/day) (as in Fig 9c, except that estimated at 1° horizontal resolution). Figure e)**
**shows future changes in precipitation as in d), but with the impact of the SNAO removed (shaded).**

13



**Figure 12. Correlations between the PC1 time series of omega at 500hPa in July and surface atmospheric**
**circulation in the periods (a,c,e) 1960-2010 and (b,d,f) 2050-2100.  Correlation values are estimated for a)-b) SLP**
**(shaded and contours), c)-d) meridional wind (shaded and contours), e)-f) precipitation (shaded and contours)**
**and vertically integrated water vapor (contours for the values -0.5, 0.3, 0.5). For a)-d) contours are shown for**
**0.25 and 0.5 correlation values.**

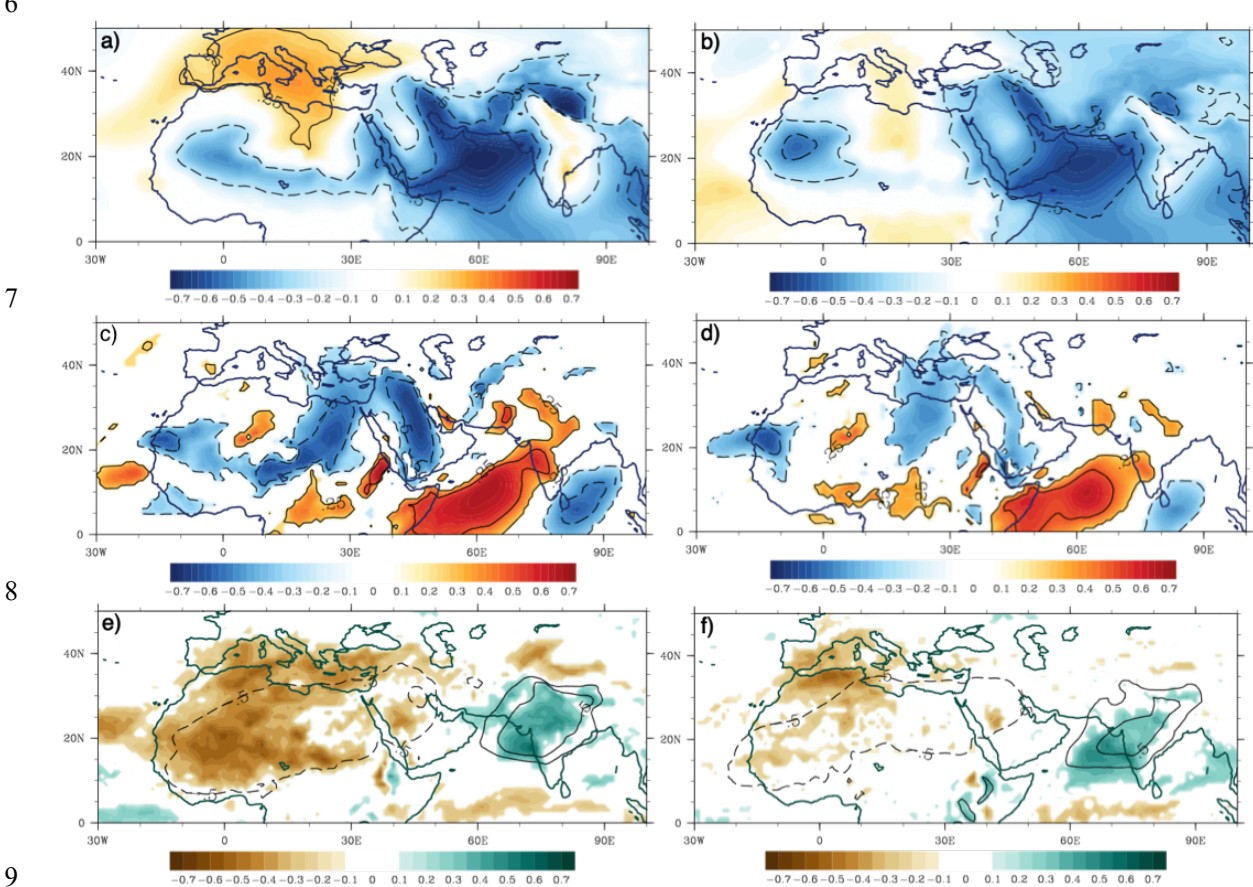




**Figure 13. a) Composite differences between the sample with the 300 warmest and 300 coolest seasons over the**
**eastern Mediterranean (30°-36°N, 36°-42°E), for July in the CTRL run, derived from a) surface temperature**
**(°C), and associated differences in b) relative humidity, c) SLP (hPa) and vector wind at 850hpa (m/s), d) height**
**at 850hPa (shaded) and 500hPa (contours), e) omega at 500 hPa, f) precipitation (mm/day).**
