# Peer review of "Changes in the future summer Mediterranean climate: contribution of teleconnections"

_Earth System Dynamics, 2018_

## Referee Comment (RC1) · Anonymous Referee #1 · 4 Mar 2019

The paper tackles the problem of climate change in the Mediterranean, exploring the role of teleconnections and local feedbacks in modulating future projections. To this aim, climate simulations are designed and run using a state-of-the-art climate model. The authors find climate change in agreement with previous literature (warming and drying), however their findings suggest reduced amplitudes in change. They explain these differences with the improved ability of the model in simulating SNAO teleconnections and land surface feedbacks.

Major comments

1. I acknowledge the huge work the authors did in carefully revising and discussing the

literature, and in performing and discussing many analysis, but this makes the paper very long. My first general recommendation is to somehow shorten the manuscript, to facilitate the reader to be focused on the key messages delivered by the paper. For instance, discussions of previous findings is sometimes too detailed and redundant: accepted knowledge on summer climate in the Mediterranean region should be described in detail in the Introduction (or in a dedicated Background section) and briefly recalled when necessary in the text.

2. The main finding of the paper is the different amplitude of future projection of the Mediterranean climate simulated by the CM2.5 model in comparison with CMIP3 and CMIP5 simulations. To illustrate this crucial aspect, the authors refer to the existing literature on the topic. However, when quantitative differences are discussed, comparison with a Figure would be helpful. I suggest the authors to add some figures (in the supplement) showing projections of the future Mediterranean climate (precipitation and temperature) in the CMIP5 ensemble or, if downloading CMIP5 data is too time consuming, at least the CM2.1 model output, to show differences with the same model at lower resolution and not including the improved land model LM3.

3. Data: why NCEP reanalysis are selected for comparison? On the same period, ERAI data are available at higher resolution. ECMWF datasets are also available for the 20th century. Same question about precipitation data: why University of Delaware? Testing other temperature/precipitation datasets (CRU, EOBS) would change your results? In general, comparing your results with different datasets would improve the robustness of your conclusions.

4. One main issue of the paper is the choice of the time window to be analysed. The aim of the paper is to study summer Mediterranean climate, so that you select JJA. However, through the paper, different periods are selected for different analysis: JA or just July. I recommend to homogenise the period to be analysed (preferably JJA), for better comparison of the results. If there are specific reasons to analyse different period, these reasons should be highlighted.

5. Methods section is rather long and sometimes confused: EOF analysis is described twice, the description of correlation/regression analysis is not really necessary here, as well as the reference to figures discussed later in the paper. I recommend to focus the section on the description of more sophisticated methods, such as EOF and stormtrack definition, and leave the description of correlation/regression analysis to the Results section. The section should be then shortened and optimised.

6. Model validation: in Figures 1-4 you compare the CTRL simulation to the NCEP data, and I see some important biases in terms of intensity (SLP in monsoonal regions) and location of some features (axes of the anticyclonic circulations at 500 hPa). This is due to the fact that in NCEP reanalysis there is GHG forcing, which is not included in the CTRL simulation (as you also highlight in the text, P21, L34-37). It would not be more consistent to compare the CTRL simulation to a different period, i.e. a period of 20C reanalysis/precipitation less affected by GHG forcing?

7. SNAO simulation: the analysis of SNAO impact in Figure 5 is not compared with any reanalysis product. The SNAO impact on climate in Europe is explained with variations in the stormtrack (Figure 7): why reanalysis data are shown and not model simulation?

8. Figure 11: the caption of the figure and discussion at P17 should be improved. You first state that you compute EOFs for the CNTR simulation, than you project the HIST-PROJ fields onto the CTRL EOF to get the 1860-2100 time series. Than you state that you also compute EOFs separately for HIST and PROJ. Then you discuss the 1860-2100 time series in Fig. 11c, then you go to HIST and PROJ EOFs in Fig. 11ab, and finally you discuss the contribution of the 1860-2100 SNAO to the end-of-century projection of precipitation (Fig. 11de). I find this discussion confusing. This is a crucial point of the paper and should be presented clearly. I recommend the authors to improve the readability of this section. Moreover, the discussion of the SNAO impact on temperature projection should be significantly expanded.

9. Results: the differences between CM2.5 and CMIP5 models in projecting the

Mediterranean climate are explained with a) better representation of the SNAO tele-connection and b) improved representation of the land-atmosphere interaction by the LM3 model (see also the Abstract). However, the improvements of the LM3 model are not presented in the paper, nor how these new features actually improve the representation of the land-atmosphere interactions (e.g. representation of soil moisture, evapotranspiration, albedo). A brief presentation of the LM3 model as well as a discussion of how it improves climate simulation in the Mediterranean is needed.

10. Conclusions: most of the paper is devoted to the analysis of the SNAO teleconnection and its impact on future climate change in the Mediterranean, which show a significant (P17, Figure 11) impact on precipitation in southern Europe. And in the abstract you indicate this as one of the main results of the paper. Conversely, in the Conclusions you somehow reduce the importance of the SNAO impact (P22, L38-40), explaining the differences with the CMIP5 simulations as a consequence of the improved land model. This point needs to be clarified.

Minor comments

P2, L15: the connection between the Mediterranean and the African monsoon has been robustly described as Mediterranean → Africa (see papers by Raicich et al. 2003 and Rowell 2003 [https://doi.org/10.1175/1520-0442(2003)016<0849:TIOMSO>2.0.CO;2]). The influence of the African monsoon on the Mediterranean is less clear: Ziv et al. 2004, but also Fontaine et al. 2011 [https://doi.org/10.1002/joc.2108], actually find a link between convection in Africa and subsidence in the Mediterranean, however the mechanism is still not clear (see Gaetani et al. 2011 [https://doi.org/10.1029/2011GL047150]). Indeed, the Asian monsoon could be dominant in modulating the Mediterranean-Africa connection. Please modify the sentence to account for this aspect.

P2, L26-29: please add a reference.

P3, L18-20: please add a reference.

P3, L43: "fixed levels of radiative forcing", do you mean 'radiative forcing from fixed levels of emission/concentration'?

P4, Methods: is the model fully coupled? How many vertical levels are in the ocean model?

P6, L15-16: this sentence should be moved to the Results section.

P6, L18-19: what do you mean with "vector time series"? The time series of the vector containing spatial data?

P7, L39: from Figure 4, precipitation magnitude is actually, not "apparently", larger than observations.

P7, L45: "none of the CMIP5..."

P8, L1: do you mean that the CM2.5 runs in the CMIP5 archive are better than other models in the archive? Or do you refer to the runs you analyse in this paper? If this is the case, you should provide a figure to support this statement.

P8, L10-13: when discussing the impact of NAO and SNAO on European climate, add references.

P8, L17: "and rather wet conditions".

P8, L18-19: add references on future projections of SNAO.

Section 3.2: the objective is to test the capability of the model in simulating the SNAO as an independent internally-generated mode of climate variability. However, the long introduction at P8-9 does not actually help in understanding why this is necessary. Is the internal variability modulated at multidecadal time scales? Is this modulation externally forced? Please try to clarify motivations and objectives of the section.

P8, L20-26: this paragraph is confusing: on the one hand, it is true that different approaches/datasets may lead to uncertainty in the observed SNAO-Mediterranean tele-

connection; on the other hand, uncertainties in model simulations originate from model shortcomings. Therefore uncertainty in the real and model worlds could originate from both intrinsic non-linear nature of the phenomenon and inadequate statistical/modelling tools. Please rephrase.

P9, L8-9: I cannot understand why and how anthropogenic forcing should intensify SNAO contribution (to the summer atmospheric circulation over North Atlantic). Please explain.

P9, L34: add the figures for July and August to the Supplement.

P9, L35: what is the interest of comparing with the HadCM3 model?

P9, L42: is it HadGEM1 or HadCM3?

P13, L6-7: why an East Mediterranean index is used to compute correlation in Figure 8d, instead of the first EOF for NCEP omega?

P14, L25-26: what do you mean with "estimated at the original model resolution"? Do you mean "computed"?

P14, L32: east.

P17, L1-3: I don't understand why you refer to Fig. 10a (showing end-of-century projections) to discuss changes in SNAO. You could maybe use this figure to support your analysis of future SNAO.

P17, L14-16: it would be preferable to present the regression method to estimate the SNAO impact here rather than in the Method section.

P17, L43: "warming is lower over . . . than . . ."

P18, L7-12: it is not clear to me whether you are discussing you results (in Figure 10) or previous findings. If you discuss your results, please add more references to Figure 10, otherwise add a reference to a paper.
P20, L39-41 and 42-45: please add references.

P21, L20: "preindustrial value".

P22, L9-10: please add a citation to CMIP5 results.

Figures: for better comparison, figures presenting climate change in the Mediterranean should share the same geographical boundaries. Same recommendation for figures presenting SNAO and Asian monsoon teleconnections, respectively.

Figure 6: what do contours represent? The sign looks reversed with respect to the standard SNAO pattern. Could you please fix this, not to mislead the reader?

Figure 7: does it make sense to project the SNAO index derived from 20CR onto NCEP data? Why not just analyse one dataset?

Figure 8: Do you perform EOF on omega 500 and 300 together? Or is EOF analysis performed separately on omega 500 and 300? If this is the case, which time series do you use for correlations?

Figure 10: wind is displayed at which level? Is not model resolution 0.5?

Figure 11: in panels d and e you show regressions, while in Fig. 9c you show correlations.

Figure 12: is omega at 200 or 500? See P18, L23.

Supplement: please follow the logical order of the paper to number the figures. Also please write complete captions, avoiding to refer to captions in the main text.

---

## Referee Comment (RC2) · Anonymous Referee #2 · 8 Mar 2019

Dear editor,

I have read this paper and think it fits the scope of ESD. Moreover, I think the material presented in the paper is an welcome addition to the knowledge on climatic changes in the Mediterranean. My main issues with the current manuscript are in the presentation. I think it is quite long and should be more focused, possibly with a reduction in the number of figures.

Kind regards,
* * *
[Figure]

Specific comments: 1. The paper is quite long and elaborate, maybe wise to focus a bit more and reduce the number of figures?

2. There are many long sentences throughout the paper, which make it hard to follow the reasoning sometimes. I suggest to have a good look at opportunities to shorten them.

3. In many places in the manuscript, geographic names (Levant, Asia Minor, Balkans, Sahara, North Africa, EMED, and many more) are used to describe the model results. This assumes that the reader knows the location of all these places, which is probably not true. I suggest to indicate the relevant places in a figure and to be more specific in the geography when mentioning other places.

4. You compare the high resolution model output to NCEP/DOE reanalysis of 2.5 degree resolution. Why not compare it to higher resolution products, such as ERA5?

5. Is there a significant difference between the top and bottom panels in Fig 7? It is not clear to me, maybe use a different color scale?

6. Indicate different data sources (HIST, PROJ runs) in Fig 11c? Maybe by a vertical line in the plot?

7. P21L35, "This may stem from...". I agree that the anthropogenic forcing is not included in the control run and may contribute to this discrepancy. But what about other explanations?

---

## Author Comment (AC1) · 5 Apr 2019

The paper tackles the problem of climate change in the Mediterranean, exploring the role of teleconnections and local feedbacks in modulating future projections. To this aim, climate simulations are designed and run using a state-of-the-art climate model. The authors find climate change in agreement with previous literature (warming and drying), however their findings suggest reduced amplitudes in change. They explain these differences with the improved ability of the model in simulating SNAO teleconnections and land surface feedbacks.

AU: We thank the reviewer for a very through revision comments and useful feedback

regarding our manuscript. Please find below our responses to the reviewer's concerns. We have specified how we would address all of the major comments when revising the manuscript. We have also made an effort to address the minor comments. We will focus our attention also to improve the readability of the text and make it more concise.

Major comments 1. I acknowledge the huge work the authors did in carefully revising and discussing the literature, and in performing and discussing many analysis, but this makes the paper very long. My first general recommendation is to somehow shorten the manuscript, to facilitate the reader to be focused on the key messages delivered by the paper. For instance, discussions of previous findings is sometimes too detailed and redundant: accepted knowledge on summer climate in the Mediterranean region should be described in detail in the Introduction (or in a dedicated Background section) and briefly recalled when necessary in the text.

AU: Thank you for the comment. Following the reviewer's suggestion we will work on the manuscript to make it more comprehensible. Particularly, we will summarize the discussion of the previous studies in a more concise way. The relevant information on the Mediterranean climate, located in sections 3.2 and 3.3, will be moved to the Introduction or to the dedicated background section.

2. The main finding of the paper is the different amplitude of future projection of the Mediterranean climate simulated by the CM2.5 model in comparison with CMIP3 and CMIP5 simulations. To illustrate this crucial aspect, the authors refer to the existing literature on the topic. However, when quantitative differences are discussed, comparison with a Figure would be helpful. I suggest the authors to add some figures (in the supplement) showing projections of the future Mediterranean climate (precipitation and temperature) in the CMIP5 ensemble or, if downloading CMIP5 data is too time consuming, at least the CM2.1 model output, to show differences with the same model at lower resolution and not including the improved land model LM3.

AU: Thank you for the advice. We will be happy to include a supplementary figure

showing future projections using the CM2.1 model output, analyzed for the summer season. We agree that this will be a good way to illustrate the impacts of the improved land model and increasing spatial resolution. Also, we would like to reference specific figures from Delworth et al. 2012, comparing the annual values in the CM2.1 and CM2.5 future projections.

3. Data: why NCEP reanalysis are selected for comparison? On the same period, ERAI data are available at higher resolution. ECMWF datasets are also available for the 20th century. Same question about precipitation data: why University of Delaware? Testing other temperature/precipitation datasets (CRU, EOBS) would change your results? In general, comparing your results with different datasets would improve the robustness of your conclusions.

AU: We took the reviewer's comment into consideration and will make an effort to address it during the revision process. Yes, we agree that a verification of the results with different datasets would improve the robustness of the conclusions. However, we would refrain from using the 20th century data sets such as ECMWF's ERA-20C or NOAA-CIRES 20th Century Reanalysis. Both of these data sets are based on the assimilation of the surface pressure values, which makes them a less plausible reference in the analysis of the upper level atmospheric dynamics. On the other hand, input data in the CRU TS time series is not fully homogenized, which constitutes a major limitation of the data sets. We will include a comparison using alternative datasets, where it is possible, and extend the discussion appropriately. We will also include the justification of the data choices in the response to the reviewer's and the revised version of the manuscript.

4. One main issue of the paper is the choice of the time window to be analysed. The aim of the paper is to study summer Mediterranean climate, so that you select JJA. However, through the paper, different periods are selected for different analysis: JA or just July. I recommend to homogenise the period to be analysed (preferably JJA), for better comparison of the results. If there are specific reasons to analyse different

period, these reasons should be highlighted.

AU: The choice of the time window is dictated by the method being most adequate to the analyzed climate component.. The teleconnection of SNAO with the climate over the Mediterranean region is manifest mostly during the peak summer, i.e. July-August. On the other hand the analysis in section 3.3. "Summer climate regime over the eastern Mediterranean" focuses on the month of July. "The choice is driven by the fact that the magnitude of subsidence and the Etesians is at its maximum in July while the response of the Rossby waves to monsoon rainfall is also the strongest (Tyrlis et al. 2012, Lin et al. 2007, Lin et al. 2009). The justifications of all the time window choices are included in the manuscript, but we will make our best to highlight this information.

5. Methods section is rather long and sometimes confused: EOF analysis is described twice, the description of correlation/regression analysis is not really necessary here, as well as the reference to figures discussed later in the paper. I recommend to focus the section on the description of more sophisticated methods, such as EOF and stormtrack definition, and leave the description of correlation/regression analysis to the Results section. The section should be then shortened and optimised.

AU: Thank you. Following the reviewer's suggestion, we will make an effort to clarify the methods section, and rewrite it in a more concise way.

6. Model validation: in Figures 1-4 you compare the CTRL simulation to the NCEP data, and I see some important biases in terms of intensity (SLP in monsoonal regions) and location of some features (axes of the anticyclonic circulations at 500 hPa). This is due to the fact that in NCEP reanalysis there is GHG forcing, which is not included in the CTRL simulation (as you also highlight in the text, P21, L34-37). It would not be more consistent to compare the CTRL simulation to a different period, i.e. a period of 20C reanalysis/precipitation less affected by GHG forcing?

AU: Comparing CTRL simulation with a different or longer period of observations than the recent three decades would be a more appropriate solution, from the perspective

of contributing forcing components. Therefore for precipitation (University of Delaware climatology) we have used a longer period of data, i.e. 1900-2010. On the other hand, for the analysis of atmospheric circulation we prefer to use a shorter period of NCEP/NCAR2 (DOE) data set rather than the longer Twentieth Century Reanalysis. We justify this choice with the findings in Krueger et al. 2013. This study has shown that the early part of the SLP record (first half of the 20th century) in the Twentieth Century Reanalysis suffers substantial inhomogeneities, most likely associated with the increasing number of observations and improved measurement techniques. Moreover, taking into account that the Twentieth Century Reanalysis assimilates only surface pressure reports, sea ice, and sea surface temperature distributions, the expression of the high level atmospheric variables could be highly uncertain in this dataset. To address the reviewer's concern on consistency in the GHG forcing between the compared data sets, we offer a solution and compare the observations with the historical runs of the model as an alternative to the control run.

7. SNAO simulation: the analysis of SNAO impact in Figure 5 is not compared with any reanalysis product. The SNAO impact on climate in Europe is explained with variations in the stormtrack (Figure 7): why reanalysis data are shown and not model simulation?

AU: This manuscript focuses on the summer Mediterranean climate, analyzed from the perspective of the contributing factors, simulated in the GFDL CM2.5 model. In this regard, we devote a part of our attention on the capabilities of the model to simulate the regional (Mediterranean region) impact of the SNAO. The representation of the observed features of SNAO, based on the SLP in different time periods, is shown in Figure 6. However, the observed impacts of SNAO (including precipitation and temperature) are the main objective and has been shown in detail in Folland et al. 2009 and Blade et al. 2012. We would prefer to make a reference to the existing literature rather than to repeat an existing analysis, especially to maintain brevity given major comment 1 requesting that we shorten the manuscript length. Yes, Figure 7 shows that the North Atlantic stormtrack is a good proxy to explain the SNAO impact on northwestern Europe (as shown also in Folland et al. 2009). Thus a comparison of the simulated and observed stormtracks would be a valuable input for a study focusing on northwestern Europe. However, the main objective of this study is different. By showing Figure 7 we intend to emphasize the sensitivity of the relationship between the SNAO and the storm tracks to the chosen period. For example, the relationship is stronger for the period starting in mid-century (i.e. 1950-1990, dominated with the dipole SNAO pattern (Figure 6e)), than for the later period (1970-2011, dominated with the monopole over the British Isles (Figure 6g), suggesting that the representation of SNAO derived from the recent decades can be obscured by other climate components. We will clarify this point in the text of the manuscript.

8. Figure 11: the caption of the figure and discussion at P17 should be improved. You first state that you compute EOFs for the CNTR simulation, than you project the HIST-PROJ fields onto the CTRL EOF to get the 1860-2100 time series. Than you state that you also compute EOFs separately for HIST and PROJ. Then you discuss the 1860-2100 time series in Fig. 11c, then you go to HIST and PROJ EOFs in Fig. 11ab, and finally you discuss the contribution of the 1860-2100 SNAO to the end-ofcentury projection of precipitation (Fig. 11de). I find this discussion confusing. This is a crucial point of the paper and should be presented clearly. I recommend the authors to improve the readability of this section. Moreover, the discussion of the SNAO impact on temperature projection should be significantly expanded.

AU: Following the reviewer's suggestion, we will clarify this part of the discussion, as well as the caption of Figure 11 and the corresponding methods.

9. Results: the differences between CM2.5 and CMIP5 models in projecting the paper Mediterranean climate are explained with a) better representation of the SNAO teleconnection and b) improved representation of the land-atmosphere interaction by the LM3 model (see also the Abstract). However, the improvements of the LM3 model are not presented in the paper, nor how these new features actually improve the representation of the land-atmosphere interactions (e.g. representation of soil moisture, evapotranspiration, albedo). A brief presentation of the LM3 model as well as a discussion of how it improves climate simulation in the Mediterranean is needed.

AU: Yes, we agree that including additional information, referring to the improvements in the land model would certainly refine the manuscript. We will make an effort to add a very brief description of the LM3 model as well as include appropriate references. However, an attribution of the differences in the future projections between CM2.1 and CM2.5 to the particular component of the land model is beyond the scope of this manuscript. Addressing that issue would require a new and differently designed analysis.

10. Conclusions: most of the paper is devoted to the analysis of the SNAO teleconnection and its impact on future climate change in the Mediterranean, which show a significant (P17, Figure 11) impact on precipitation in southern Europe. And in the abstract you indicate this as one of the main results of the paper. Conversely, in the Conclusions you somehow reduce the importance of the SNAO impact (P22, L38-40), explaining the differences with the CMIP5 simulations as a consequence of the improved land model. This point needs to be clarified.

AU: Following the reviewer's suggestion, we will clarify the main message of the manuscript. We agree that the analysis of the impacts of the SNAO teleconnection consumes a significant part of the manuscript. This part shows a significant contribution of the SNAO to precipitation over southern Europe. However, the comparison of the SNAO impacts between CM2.5 model and CMIP3/CMIP5 models suggests that the SNAO can not explain the difference in the future projections between these models. The apparent stark contrast between the CMIP3/CMIP5 and CM2.5 regional projections could more likely originate from the enhancements in the LM3 land model incorporated to CM2.5 at high spatial resolution, rather than the impacts of the SNAO. Future work should focus on understanding differences in land surface responses in this region to the SNAO and projected climate change.

[Figure]

Minor comments: P2, L15: the connection between the Mediterranean and the African monsoon has been robustly described as Mediterranean → Africa (see papers by Raicich et al. 2003 and Rowell 2003 [https://doi.org/10.1175/1520-0442(2003)0162.0.CO;2]). The influence of the African monsoon on the Mediterranean is less clear: Ziv et al. 2004, but also Fontaine et al. 2011 [https://doi.org/10.1002/joc.2108], actually find a link between convection in Africa and subsidence in the Mediterranean, however the mechanism is still not clear (see Gaetani et al. 2011 [https://doi.org/10.1029/2011GL047150]). Indeed, the Asian monsoon could be dominant in modulating the Mediterranean-Africa connection. Please modify the sentence to account for this aspect. AU: Thank you for the comment. We will apply the correction.

P2, L26-29: please add a reference. P3, L18-20: please add a reference. AU: Thank you, we will add the references.

P3, L43: "fixed levels of radiative forcing", do you mean 'radiative forcing from fixed levels of emission/concentration'? AU: Thank you for the correction.

P4, Methods: is the model fully coupled? How many vertical levels are in the ocean model? AU: We will add the information.

P6, L15-16: this sentence should be moved to the Results section. P6, L18-19: what do you mean with "vector time series"? The time series of the vector containing spatial data? P7, L39: from Figure 4, precipitation magnitude is actually, not "apparently", larger than observations. P7, L45: "none of the CMIP5. . ." AU: Yes. Thank you for the correction. We will apply all of the above suggestions.

P8, L1: do you mean that the CM2.5 runs in the CMIP5 archive are better than other models in the archive? Or do you refer to the runs you analyse in this paper? If this is the case, you should provide a figure to support this statement. AU: Thank you. We referred to the CMIP5 analysis shown in Kelley et al. (2012). We will clarify this and add the reference where appropriate.

P8, L10-13: when discussing the impact of NAO and SNAO on European climate, add references. AU: We will add the references. P8, L17: "and rather wet conditions". AU: Thank you, we will apply the correction. P8, L18-19: add references on future projections of SNAO. AU: We will add the references.

Section 3.2: the objective is to test the capability of the model in simulating the SNAO as an independent internally-generated mode of climate variability. However, the long introduction at P8-9 does not actually help in understanding why this is necessary. Is the internal variability modulated at multidecadal time scales? Is this modulation externally forced? Please try to clarify motivations and objectives of the section.

AU: As stated in the introduction of section 3.2, the purpose is to analyze the capability of the CM2.5 model to simulate the SNAO as an independent, internally generated climate component, which would prove the physical validity of the statistically-derived component. Yes, we agree that the introduction of section 3.2 is too long. Following one of the previous comments, we will move part of the information to section 1 and clarify the main purpose of section 3.2.

P8, L20-26: this paragraph is confusing: on the one hand, it is true that different approaches/datasets may lead to uncertainty in the observed SNAO-Mediterranean teleconnection; on the other hand, uncertainties in model simulations originate from model shortcomings. Therefore uncertainty in the real and model worlds could originate from both intrinsic non-linear nature of the phenomenon and inadequate statistical/modelling tools. Please rephrase. AU: Thank you, we will clarify the paragraph.

P9, L8-9: I cannot understand why and how anthropogenic forcing should intensify SNAO contribution (to the summer atmospheric circulation over North Atlantic). Please explain. AU: We will make an effort to elaborate more on this issue.

P9, L34: add the figures for July and August to the Supplement. AU: Thank you, we will take this comment into consideration.
P9, L35: what is the interest of comparing with the HadCM3 model? AU: This section analyzes the capability of CM2.5 in simulating SNAO and compares it with the available results of other models, in this case HadCM3 and HadGEM1 in Folland et al. 2009.

P9, L42: is it HadGEM1 or HadCM3? AU: The statement is correct.

P13, L6-7: why an East Mediterranean index is used to compute correlation in Figure 8d, instead of the first EOF for NCEP omega? AU: Figure 8d shows the correlations computed based on three-decade time series of NCEP omega at the mid-atmospheric level. Taking into account the relatively short length of the data set (compared to 1000 years CTRL run) and a relatively smaller plausibility of the data at the middle and higher atmospheric levels, we refrain from applying an EOF analysis to the NCEP dataset. In our consideration, applying an EOF analysis to such a short time series could lead to degeneracy of the derived eigenvalues. In other words, the EOF mode derived from the NCEP dataset could be easily a spurious combination of several modes, rater than a realistic representation of the SNAO mode. Therefore for computing correlations using such a short data set (Figure 8d), we prefer to use full time series rather than time series of computed EOF.

P14, L25-26: what do you mean with "estimated at the original model resolution"? Do you mean "computed"? AU: Yes, Thank you for the correction. P14, L32: east. P17, L1-3: I don't understand why you refer to Fig. 10a (showing end-of-century projections) to discuss changes in SNAO. You could maybe use this figure to support your analysis of future SNAO. AU: We are using this figure to support the analysis of future SNAO, but also the interpretation of the analysis of the already observed SNAO changes. The fingerprint of the future changes, derived from the sea level pressure projections, is consistent with the observed evolution of SNAO and thus it may constitute a possible contribution of the anthropogenic component already observed in the 20th century. We will try to clarify this issue in the manuscript.

P17, L14-16: it would be preferable to present the regression method to estimate the

SNAO impact here rather than in the Method section. P17, L43: "warming is lower over . . . than . . ." AU: Thank you, we will take both of the comments above into consideration. P18, L7-12: it is not clear to me whether you are discussing you results (in Figure 10) or previous findings. If you discuss your results, please add more references to Figure 10, otherwise add a reference to a paper. AU: Yes, we will follow the advice. P20, L39-41 and 42-45: please add references. AU: Thank you, we will add the references. P21, L20: "preindustrial value". P22, L9-10: please add a citation to CMIP5 results. AU: Thank you, we will follow the advice.

Figures: for better comparison, figures presenting climate change in the Mediterranean should share the same geographical boundaries. Same recommendation for figures presenting SNAO and Asian monsoon teleconnections, respectively. Figure 6: what do contours represent? The sign looks reversed with respect to the standard SNAO pattern. Could you please fix this, not to mislead the reader? Figure 7: does it make sense to project the SNAO index derived from 20CR onto NCEP data? Why not just analyse one dataset? AU: We agree with the reviewer that it is usually easier to use just one data set. However for the sake of consistency with an earlier part of the analysis, we used the 20CR dataset instead of the NCEP dataset. In the earlier part of the analysis we used the 20CR dataset, because the alternative ones, such as NCEP-NCAR1 or NCEP –DOE, would be too short for the analysis of the evolution of SNAO during the 20th century.

Figure 8: Do you perform EOF on omega 500 and 300 together? Or is EOF analysis performed separately on omega 500 and 300? If this is the case, which time series do you use for correlations? AU: Yes, the EOF analysis is performed separately for each level. The method is described in the manuscript but we will clarify and highlight this information.

Figure 10: wind is displayed at which level? Is not model resolution 0.5? AU: Thank you for the correction. We will also add the necessary information regarding the level.

Figure 11: in panels d and e you show regressions, while in Fig. 9c you show correlations. Figure 12: is omega at 200 or 500? See P18, L23. Supplement: please follow the logical order of the paper to number the figures. Also please write complete captions, avoiding to refer to captions in the main text. AU: Thank you for all the comments and advice regarding the figures. We will make an effort to improve the quality of the figures and we will adjust and correct the respective captions.

Please also note the supplement to this comment:
https://www.earth-syst-dynam-discuss.net/esd-2018-85/esd-2018-85-AC1-supplement.pdf

---

## Author Comment (AC2) · 5 Apr 2019

Dear editor, I have read this paper and think it fits the scope of ESD. Moreover, I think the material presented in the paper is an welcome addition to the knowledge on climatic changes in the Mediterranean. My main issues with the current manuscript are in the presentation. I think it is quite long and should be more focused, possibly with a reduction in the number of figures. Kind regards,

AU: We thank the reviewer for a very thorough revision comments and useful feedback regarding our manuscript. Please find below our responses to the reviewer's concerns. We have specified how we would address all of the major comments when revising the

manuscript. We have also made an effort to address the minor comments. We will focus our attention also to improve the readability of the text and make it more concise.

Specific comments: 1. The paper is quite long and elaborate, maybe wise to focus a bit more and reduce the number of figures?

AU: Thank you for the comment. Following the reviewer's suggestion, we will work on the manuscript to make it more comprehensible. Particularly, we will summarize the discussion of the previous studies in a more concise way. We will also move the relevant information on the Mediterranean climate, located in sections 3.2 and 3.3, to the Introduction or to the dedicated background section.

2. There are many long sentences throughout the paper, which make it hard to follow the reasoning sometimes. I suggest to have a good look at opportunities to shorten them.

AU: Thank you. We will follow the suggestion and clarify the key messages.

3. In many places in the manuscript, geographic names (Levant, Asia Minor, Balkans, Sahara, North Africa, EMED, and many more) are used to describe the model results. This assumes that the reader knows the location of all these places, which is probably not true. I suggest to indicate the relevant places in a figure and to be more specific in the geography when mentioning other places.

AU: Thank you. We will make an effort to be more specific when using geographic names.

4. You compare the high resolution model output to NCEP/DOE reanalysis of 2.5 degree resolution. Why not compare it to higher resolution products, such as ERA5?

AU: Thank you for the comment. We will take it into consideration. We will reconsider the choice of the reference observations used in the manuscript.

5. Is there a significant difference between the top and bottom panels in Fig 7? It is not

clear to me, maybe use a different color scale?

AU: Yes there is a significant difference between the top and bottom correlations, computed between the stormtrack proxy and the SNAO index. For example, negative correlations over northwest Europe in the 1970-2011 (bottom) are weaker than -0.5, while the correlations in 1950-1990 (top) are stronger than -0.65. We will consider a way to improve the figure, so the difference is more discernible for the reader.

6. Indicate different data sources (HIST, PROJ runs) in Fig 11c? Maybe by a vertical line in the plot?

AU: Thank you, we will follow the advice.

7. P21L35, "This may stem from...". I agree that the anthropogenic forcing is not included in the control run and may contribute to this discrepancy. But what about other explanations?

AU: Here we would like to provide a more thorough explanation for the apparent better agreement between the observed and simulated SNAO before the 1980s than for the later period:

"This may stem from the fact that the SNAO component derived for the simulated and observed recent decades (1980s-2010s) could be to a large extent conditioned by a coincidence of the multidecadal scale internal (unforced) variability and multidecadal anthropogenic forcing. Taking into account the random nature of the unforced climate variations, their temporal evolution in the simulated 1980's-2010's period may look very different from the observed one. In a presence of an additional long-term component, such as anthropogenic forcing, the EOF analysis (which is set to extract a signal explaining a maximum variance) is likely to choose a combination of a random representation of the unforced variability and the anthropogenic forcing. This combination (regarded as the SNAO component) may have a very different form in the observations and simulations."

We will clarify the relevant paragraph of the manuscript.

Please also note the supplement to this comment:
https://www.earth-syst-dynam-discuss.net/esd-2018-85/esd-2018-85-AC2-supplement.pdf

---

## Author Response (AR1)

The paper tackles the problem of climate change in the Mediterranean, exploring the role of teleconnections and local feedbacks in modulating future projections. To this aim, climate simulations are designed and run using a state-of-the-art climate model. The authors find climate change in agreement with previous literature (warming and drying), however their findings suggest reduced amplitudes in change. They explain these differences with the improved ability of the model in simulating SNAO teleconnections and land surface feedbacks.

**We thank the reviewer for a very through revision comments and feedback regarding our manuscript. We have addressed all of the major comments when revising the manuscript. This includes either applying the reviewer's suggestion or a thorough justification for not doing that. We have also made an effort to address the minor comments. Overall, we've reworked the text of the manuscript to improve its conciseness and clarity.**

Major comments 1. I acknowledge the huge work the authors did in carefully revising and discussing the literature, and in performing and discussing many analysis, but this makes the paper very long. My first general recommendation is to somehow shorten the manuscript, to facilitate the reader to be focused on the key messages delivered by the paper. For instance, discussions of previous findings is sometimes too detailed and redundant: accepted knowledge on summer climate in the Mediterranean region should be described in detail in the Introduction (or in a dedicated Background section) and briefly recalled when necessary in the text.

**AU: Thank you for the comment. Following the reviewer's suggestion we've reworked the manuscript to make it more comprehensible. Particularly, we have summarized the discussion of the previous studies in a more concise way (section 3.2.1 and 3.2.2, pp 9: lines 1-46, pp 10: lines 7-21). The relevant information on the Mediterranean climate moved from sections 3.2 and 3.3 (pp 8: lines 15-27) to the Introduction (pp2: lines 10-22).**

2. The main finding of the paper is the different amplitude of future projection of the Mediterranean climate simulated by the CM2.5 model in comparison with CMIP3 and CMIP5 simulations. To illustrate this crucial aspect, the authors refer to the existing literature on the topic. However, when quantitative differences are discussed, comparison with a Figure would be helpful. I suggest the authors to add some figures (in the supplement) showing projections of the future Mediterranean climate (precipitation and temperature) in the CMIP5 ensemble or, if downloading CMIP5 data is too time consuming, at least the CM2.1 model output, to show differences with the same model at lower resolution and not including the improved land model LM3.

**AU: Thank you for the suggestion. In order to facilitate the comparison between the CM2.5 future projections and other models we have updated references and also included a supplementary figure. Figure R1 depicts changes estimated based on the RCP8.5 CSIRO-Mk3-6-0 model 10-member projections ensemble. Unfortunately, there are no RCP scenarios for GFDL CM2.1. Instead, we use the CSIRO-Mk3-6-0 model, which includes the ocean component based on the GFDL ocean model. We refer also to the very comprehensive analysis in Jacob et al. 2014 and Fussel et al. 2017 (Map 3.4, pp. 76), employing many GCM-RCM combined simulations from the EURO-CORDEX project. Discussion of these results is included in the text (section 4.1, pp 15: lines 10-27).**

[Figure]

Figure R1. Projected future changes for the summer (JJA) surface temperature (left,°C), and precipitation (mm/day, right) based on the 10-member ensemble simulations of the CSIRO-Mk3-6-0 model, for the forcing scenario RCP8.5.

3. Data: why NCEP reanalysis are selected for comparison? On the same period, ERAI data are available at higher resolution. ECMWF datasets are also available for the 20th century. Same question about precipitation data: why University of Delaware? Testing other temperature/precipitation datasets (CRU, EOBS) would change your results? In general, comparing your results with different datasets would improve the robustness of your conclusions.

**AU: Yes, we agree that a verification of the results with different datasets would improve the robustness of the conclusions. Following reviewer's suggestion we have incorporated EOBS dataset in the analysis of the regional precipitation (Figure 4). We agree that EOBS provides high quality data and serves as a good reference for the regional precipitation. Given the same spatial resolution as CM2.5, EOBS allows for comparing very fine features over the complex orography of the Mediterranean.**
**However, incorporating datasets with higher resolution wouldn't necessarily serve the purpose of the analysis of the large-scale circulation features. We have included below an additional figure (Figure R2), depicting sea level pressure and wind vector at 850hPa both in ERA-I and ERA 20CR, showing very good agreement between the suggested observational datasets.**

[Figure]

Figure R2. Seasonal (JJA) sea level pressure (hPa) and wind vector at 850hPa (m/s) in (top) ERA-20C, (bottom) ERA-I; for the 1979-2017 period.

**In the analysis of the mid-level and upper level atmospheric dynamics, we would refrain from using the 20th century data sets such as ECMWF's ERA-20C or NOAA-CIRES 20th Century Reanalysis. Both of these data sets are based on the assimilation of the surface pressure values, which makes them a less plausible reference in the analysis of the upper level atmospheric dynamics. For example, the comparison of the vertical velocity at 500 hPa in Figure R3 reveals a strong positive bias in ERA-I, compared to the rest of the analyzed datasets (note the difference in the scalebar).**

[Figure]

Figure R3. Seasonal (July) time-mean vertical velocities at 500hPa (Pa/s), estimated for (top) NOAA-CIRES 20th Century Reanalysis, (middle) as ECMWF's ERA-20C, (bottom) ERA-Interim for 1979-2017.

4. One main issue of the paper is the choice of the time window to be analysed. The aim of the paper is to study summer Mediterranean climate, so that you select JJA. However, through the paper, different periods are selected for different analysis: JA or just July. I recommend to homogenize the period to be analyzed (preferably JJA), for better comparison of the results. If there are specific reasons to analyze different period, these reasons should be highlighted.

**AU: The choice of the time window is dictated by the method being most adequate to the analyzed climate component.. The teleconnection of SNAO with the climate over the Mediterranean region is manifest mostly during the peak summer, i.e. July-August. On the other hand the analysis in section 3.3. "Summer climate regime over the eastern Mediterranean" focuses on the month of July. "The choice is driven by the fact that the magnitude of subsidence and the Etesians is at its maximum in July while the response of the Rossby waves to monsoon rainfall is also the strongest (Tyrlis et al. 2012, Lin et al. 2007, Lin et al. 2009). The justifications of all the time window choices are included in the manuscript, but we have highlighted this information both in Methods (pp 5, lines: 22-23; pp 6, lines: 25-27; pp 7, lines: 5-8) and in the text (pp 12, lines: 10-14).**

5. Methods section is rather long and sometimes confused: EOF analysis is described twice, the description of correlation/regression analysis is not really necessary here, as well as the reference to figures discussed later in the paper. I recommend to focus the section on the description of more sophisticated methods, such as EOF and stormtrack definition, and leave the description of correlation/regression analysis to the Results section. The section should be then shortened and optimised.

**AU: We have clarified and rewritten the Method section in a more concise way (pp 6, lines: 12-27, 29-38; pp 7, lines: 1-21). Additionally, we have updated the description of storm track definition (p5, 40-46).**

6. Model validation: in Figures 1-4 you compare the CTRL simulation to the NCEP data, and I see some important biases in terms of intensity (SLP in monsoonal regions) and location of some features (axes of the anticyclonic circulations at 500 hPa). This is due to the fact that in NCEP reanalysis there is GHG forcing, which is not included in the CTRL simulation (as you also highlight in the text, P21, L34-37). It would not be more consistent to compare the CTRL simulation to a different period, i.e. a period of 20C reanalysis/precipitation less affected by GHG forcing?

**AU: We would refrain from the suggestion offered by the reviewer. Comparing CTRL simulation with a different or longer period of observations than the recent three decades would be a more appropriate solution, from the perspective of contributing forcing components. However, the quality of the observations-based precipitation datasets is much lower before the satellite era (~1979). For the analysis of atmospheric circulation we prefer to use a shorter period of NCEP/NCAR2 (DOE) data set rather than the longer Twentieth Century Reanalysis. We justify this choice with the findings in Krueger et al. 2013. This study has shown that the early part of the SLP record (first half of the 20th century) in the Twentieth Century Reanalysis suffers substantial inhomogeneities, most likely associated with the increasing number of observations and improved measurement techniques. Moreover, taking into account that both the NOAA's 20CR Reanalysis as well as ECMWF's ERA-20C datasets assimilate only surface pressure reports, sea ice, and sea surface temperature distributions, the expression of the high level atmospheric variables is highly uncertain in these datasets, as shown for example in the Figure 2 (omega at 500hPa) in the response to the question #3.**

7. SNAO simulation: the analysis of SNAO impact in Figure 5 is not compared with any reanalysis product. The SNAO impact on climate in Europe is explained with variations in the stormtrack (Figure 7): why reanalysis data are shown and not model simulation?

**AU: The representation of the SNAO in observations, using the SLP in different time periods, is shown in Figure 6. However the observed impact of SNAO has been shown in detail in Folland et al. 2009 and Blade et al. 2012. We prefer to make a reference to the existing literature rather than to repeat an existing analysis, especially to maintain brevity given major comment 1 requesting that we shorten the manuscript length.**

**Yes, the SNAO teleconnection with hydroclimate in northwestern Europe can be directly explained with variations in large-scale circulation over the North Atlantic and associated variations in storm tracks (as shown in observations and simulations in Folland et al. 2009). In this manuscript we intend to emphasize the sensitivity of the relationship between the SNAO and the storm tracks to the chosen period. For example, the relationship is stronger for the period starting in mid-century (i.e. 1950-1990, dominated with the dipole SNAO pattern (Figure 6e) than for the later period (1970-2011, dominated with the monopole over the British Isles (Figure 6g), suggesting that the representation of SNAO derived from the recent decades is likely being obscured by other climate components. We don't intend to investigate the simulated in CM2.5 impacts of SNAO on the North-Atlantic storm tracks. This is of course an interesting topic, but beyond the scope of this manuscript. We will clarify this point in the manuscript (pp 10: lines 33- 45; pp 11: 1-3). To maintain brevity, given major comment 1 requesting that we shorten the manuscript length, we will also move Figure 7 to the supplementary material.**

8. Figure 11: the caption of the figure and discussion at P17 should be improved. You first state that you compute EOFs for the CNTR simulation, than you project the HIST-PROJ fields onto the CTRL EOF to get the 1860-2100 time series. Than you state that you also compute EOFs separately for HIST and PROJ. Then you discuss the 1860-2100 time series in Fig. 11c, then you go to HIST and PROJ EOFs in Fig. 11ab, and finally you discuss the contribution of the 1860-2100 SNAO to the end-of century projection of precipitation (Fig. 11de). I find this discussion confusing. This is a crucial point of the paper and should be presented clearly. I recommend the authors to improve the readability of this section. Moreover, the discussion of the SNAO impact on temperature projection should be significantly expanded.

**AU: Following the reviewer's suggestion, we will clarify this part of the discussion, as well as the caption of Figure 11 and the corresponding methods. Yes, we agree. In this section we address three problems, which requires three different analysis approaches. We consider important to justify each of them. We have made a substantial effort to optimize the length of the discussion, clarified the discussion and improved the relevant caption (pp 15, 16).**

**We also would like to remind, that the manuscript focuses on the impact of SNAO on the Mediterranean hydroclimate, and its future changes (section 3.2.2 and 4.2). There is a very limited knowledge on the impacts of SNAO on temperature (basically two papers: Blade et al. 2012, Folland et al. 2009), and these results are far from unequivocal. The results shown in these two articles differ depending on chosen length of dataset and method of analysis (e.g. whether the correlations are based on interannual variations or including multidecadal signal). Our analysis seems more consistent with Folland et al. 2009, who applied rigorous statistics to avoid effects of autocorrelation caused by low-frequency components in a relatively short data set. We have now included the relevant information and highlighted the existing ambiguity to the section 3.2.1 (pp 11). However, we would prefer to abstain from further discussion, to avoid speculation. If reviewer has certain opinion on this issue, it might be helpful to receive a more elaborate suggestion.**

9. Results: the differences between CM2.5 and CMIP5 models in projecting the paper Mediterranean climate are explained with a) better representation of the SNAO teleconnection and b) improved representation of the land-atmosphere interaction by the LM3 model (see also the Abstract). However, the improvements of the LM3 model are not presented in the paper, nor how these new features actually improve the representation of the land-atmosphere interactions (*e.g. representation of soil moisture, evapotranspiration, albedo*). A brief presentation of the LM3 model as well as a discussion of how it improves climate simulation in the Mediterranean is needed.

**AU: Thank you for bringing up this gap in our writing. We have clarified the Abstract, added references to the data section (pp 5, lines 1-4), and added text to the results and summary section (pp 15, lines 4-13; pp 21, lines 10-24) relating to land model improvements and some research highlighting LM3 performance. We agree that more analysis is required to further support this claim, which would add significant text and figures to the paper. We have included references of studies beginning to look into this problem, but have left the exploration of land-atmosphere interactions of the SNAO using LM3 for future research.**

Added references:

Berg, A., B.R. Lintner, K. Findell, S.I. Seneviratne, B. van den Hurk, A. Ducharne, F. Chéruy, S. Hagemann, D.M. Lawrence, S. Malyshev, A. Meier, and P. Gentine, 2015: Interannual Coupling between Summertime Surface Temperature and Precipitation over Land: Processes and Implications for Climate Change. J. Climate, 28, 1308–1328, https://doi.org/10.1175/JCLI-D-14-00324.1

Berg, A., Findell, K., Lintner, B., Giannini, A., Seneviratne, S. I., Van den Hurk, B., Ruth Lorenz, Andy Pitman, Stefan Hagemann, Arndt Meier, Frédérique Cheruy, Agnès Ducharne, Sergey Malyshev & P. C. D.

Milly, 2016: Land–atmosphere feedbacks amplify aridity increase over land under global warming. Nature Climate Change, 6(9), 869.

Milly, P.C., S.L. Malyshev, E. Shevliakova, K.A. Dunne, K.L. Findell, T. Gleeson, Z. Liang, P. Phillipps, R.J. Stouffer, and S. Swenson, 2014: An Enhanced Model of Land Water and Energy for Global Hydrologic and Earth-System Studies. J. Hydrometeor., 15, 1739–1761, https://doi.org/10.1175/JHM-D-13-0162.1

10. Conclusions: most of the paper is devoted to the analysis of the SNAO teleconnection and its impact on future climate change in the Mediterranean, which show a significant (P17, Figure 11) impact on precipitation in southern Europe. And in the abstract you indicate this as one of the main results of the paper. Conversely, in the Conclusions you somehow reduce the importance of the SNAO impact (P22, L38-40), explaining the differences with the CMIP5 simulations as a consequence of the improved land model. This point needs to be clarified.

**AU: Following the reviewer's suggestion, we have clarified the main message of the manuscript. We agree that the analysis of the impacts of the SNAO teleconnection consumes a significant part of the manuscript. This part shows a significant contribution of the SNAO to precipitation over southern Europe. However, the comparison of the SNAO impacts between CM2.5 model and CMIP3/CMIP5 models suggests that the SNAO can not explain the difference in the future projections between these models. The apparent stark contrast between the CMIP3/CMIP5 and CM2.5 regional projections could more likely originate from the enhancements in the LM3 land model incorporated to CM2.5 at high spatial resolution, rather than the impacts of the SNAO. Future work should focus on understanding differences in land surface responses in this region to the SNAO and projected climate change.**

Minor comments:
P2, L15: the connection between the Mediterranean and the African monsoon has been robustly described as Mediterranean → Africa (see papers by Raicich et al. 2003 and Rowell 2003 [https://doi.org/10.1175/1520- 0442(2003)0162.0.CO;2]).

The influence of the African monsoon on the Mediterranean is less clear: Ziv et al. 2004, but also Fontaine et al. 2011 [https://doi.org/10.1002/joc.2108], actually find a link between convection in Africa and subsidence in the Mediterranean, however the mechanism is still not clear (see Gaetani et al. 2011 [https://doi.org/10.1029/2011GL047150]). Indeed, the Asian monsoon could be dominant in modulating the Mediterranean-Africa connection. Please modify the sentence to account for this aspect.

**Thank you for the comment. We found and included all the missing references, and modified the sentence (Raicich et al. 2003 and Rowell 2003, Fontaine et al. 2011) (pp 2, lines: 22-24).**

P2, L26-29: please add a reference.
P3, L18-20: please add a reference.
**Thank you, we add the references.**

P3, L43: "fixed levels of radiative forcing", do you mean 'radiative forcing from fixed levels of emission/concentration'?
**We have corrected the sentence (pp:4, line 14)**

P4, Methods: is the model fully coupled? How many vertical levels are in the ocean model?
**We have added the information (pp 4).**

P6, L15-16: this sentence should be moved to the Results section. **Thank you, we have applied the suggestion.**

P6, L18-19: what do you mean with "vector time series"? The time series of the vector containing spatial data? **Correct.**

P7, L39: from Figure 4, precipitation magnitude is actually, not "apparently", larger than observations. **We have corrected the sentence. (pp 8, line 19)**

P7, L45: "none of the CMIP5. . ." **Yes, we applied the correction (pp 8, line 24).**

P8, L1: do you mean that the CM2.5 runs in the CMIP5 archive are better than other models in the archive? Or do you refer to the runs you analyse in this paper? If this is the case, you should provide a figure to support this statement.
**Thank you. We referred to the CMIP5 analysis shown in Kelley et al. (2012). We will clarify this and add the reference where appropriate.**

P8, L10-13: when discussing the impact of NAO and SNAO on European climate, add references.
**We have added the references throughout the paper.**

P8, L17: "and rather wet conditions".
**Thank you, we applied the correction (pp 2, line 11).**

P8, L18-19: add references on future projections of SNAO.
**This paragraph is now rewritten and moved to Introduction (pp2)**

Section 3.2: the objective is to test the capability of the model in simulating the SNAO as an independent internally-generated mode of climate variability. However, the long introduction at P8-9 does not actually help in understanding why this is necessary. Is the internal variability modulated at multidecadal time scales? Is this modulation externally forced? Please try to clarify motivations and objectives of the section.

**As stated in the introduction of section 3.2, the purpose is to analyze the capability of the CM2.5 model to simulate the SNAO as an independent, internally generated climate component, which would prove the physical validity of the statistically-derived component. The "internally-generated" means that that the modulation is generated by the internal climate variations and it is not externally forced. However "the origin of the multi-decadal signal of SNAO has been linked (Knight et al. 2006, Folland et al 2009, Linderholm and Folland 2017) to the Atlantic Multidecadal Oscillation, which originates from internal variations in thermohaline circulation (Knight et al. 2005, 2006, Delworth and Mann 2000; Enfield et al. 2001), but for the recent decades also from anthropogenic sources (Rotstayn and Lohman 2002; Mann and Emanuel 2006)." Yes, we agree that the introduction of section 3.2 is too long. We have moved part of the information to section 1 and clarified the main purpose of section 3.2 (pp 8, lines: 30-44).**

P8, L20-26: this paragraph is confusing: on the one hand, it is true that different approaches/datasets may lead to uncertainty in the observed SNAO-Mediterranean teleconnection; on the other hand, uncertainties in model simulations originate from model shortcomings. Therefore uncertainty in the real and model worlds could originate from both intrinsic non-linear nature of the phenomenon and inadequate statistical/modelling tools. Please rephrase.
**We have substantially shortened and clarified the paragraph (pp 8, lines 30-44).**

P9, L8-9: I cannot understand why and how anthropogenic forcing should intensify SNAO contribution (to the summer atmospheric circulation over North Atlantic). Please explain.
**We have substantially rewritten this section and this sentence is not included.**

P9, L34: add the figures for July and August to the Supplement.
**We won't add this figure to the Supplementary Information, which already contains 10 supplementary figures, but have included the information that the figure is not attached (pp 9, line: 15),**

P9, L35: what is the interest of comparing with the HadCM3 model?
**This section analyzes the capability of CM2.5 in simulating SNAO and compares it with the available results of other models, in this case HadCM3 and HadGEM1 in Folland et al. 2009.**

P9, L42: is it HadGEM1 or HadCM3?

**The statement is correct.**
P13, L6-7: why an East Mediterranean index is used to compute correlation in Figure 8d, instead of the first EOF for NCEP omega? **Figure 8d shows the correlations computed based on three-decade time series of NCEP omega at the mid-atmospheric level. Taking into account the relatively short length of the data set (compared to 1000 years CTRL run) and a relatively smaller plausibility of the data at the middle and higher atmospheric levels, we refrain from applying an EOF analysis to the NCEP dataset. In our consideration, applying an EOF analysis to such a short time series could lead to degeneracy of the derived eigenvalues. In other words, the EOF mode derived from the NCEP dataset could be easily a spurious combination of several modes, rater than a realistic representation of the SNAO mode.**

P14, L25-26: what do you mean with "estimated at the original model resolution"? Do you mean "computed"? **Yes, Thank you for the correction (pp 13, line 8)**
P14, L32: east. **Correction applied (pp 13, line 15)**
P17, L1-3: I don't understand why you refer to Fig. 10a (showing end-of-century projections) to discuss changes in SNAO. You could maybe use this figure to support your analysis of future SNAO.
**The figure shows the projected future changes (please note that in the current version this is Figure 9) and indeed we are using this figure to support the analysis of future SNAO (although it supports also the interpretation of the analysis of the already observed SNAO changes). The fingerprint of the future changes, derived from the sea level pressure projections, is consistent with the observed evolution of SNAO and thus it may constitute a possible contribution of the anthropogenic component already observed in the 20th century. We have tried to clarify this issue in the manuscript (pp 15, lines: 11-30).**

P17, L14-16: it would be preferable to present the regression method to estimate the SNAO impact here rather than in the Method section. **Yes, we moved this line to the Methods.**
P17, L43: "warming is lower over . . . than . . ."
**Thank you, we have substantially rewritten and improved this section.**
P18, L7-12: it is not clear to me whether you are discussing you results (in Figure 10) or previous findings. If you discuss your results, please add more references to Figure 10, otherwise add a reference to a paper.
**Yes, we have clarified the paragraph (pp 16, 17-25).**
P20, L39-41 and 42-45: please add references.
**Thank you, we have updated references in the section.**
P21, L20: "preindustrial value". **We have substantially shortened and clarified whole section.**
**P22, L9-10: please add a citation to CMIP5 results.**
**Thank you, we have applied the correction.**

Figures: for better comparison, figures presenting climate change in the Mediterranean should share the same geographical boundaries. Same recommendation for figures presenting SNAO and Asian monsoon teleconnections, respectively.

**Unfortunately this idea is not feasible and doesn't serve the purpose of the analysis. Each figure is presented in different context. For example, projected future changes for SLP are discussed in the context of the Euro-Atlantic climate. In contrast, regarding precipitation and temperature we want to focus on the Mediterranean region, but also discuss it form the perspective of climate in Europe. We have thoroughly reconsidered the way each figure is presented.**

Figure 6: what do contours represent? The sign looks reversed with respect to the standard SNAO pattern. Could you please fix this, not to mislead the reader?
**We have improved the figure and presented the SNAO at its positive phase. We have included the clarifying information in the caption. However, we strongly disagree with the reviewer in saying that the "positive phase" is a "standard pattern".**

Figure 7: does it make sense to project the SNAO index derived from 20CR onto NCEP data? Why not just analyse one dataset?

**We agree with the reviewer that it is usually easier to use just one data set. However for the sake of consistency with an earlier part of the analysis, we used the 20CR dataset instead of the NCEP dataset. In the earlier part of the analysis we used the 20CR dataset, because the alternative ones, such as NCEP-NCAR1 or NCEP –DOE, would be too short for the analysis of the evolution of SNAO during the 20th century.**

Figure 8: Do you perform EOF on omega 500 and 300 together? Or is EOF analysis performed separately on omega 500 and 300? If this is the case, which time series do you use for correlations?
**Yes, the EOF analysis is performed separately for each level. The method is described in the manuscript but we will clarify and highlight this information also in the caption (pp7, lines 1-8).**

Figure 10: wind is displayed at which level? Is not model resolution 0.5?
**Thank you, we have included the missing information and applied the correction.**
 Figure 11: in panels d and e you show regressions, while in Fig. 9c you show correlations.
**Figure d and e doesn't show regressions, but the projected changes. The impact of the SNAO in figure e is removed using linear regression. We have clarified the figure the caption.**
 Figure 12: is omega at 200 or 500? **The information is included.**

See P18, L23. Supplement: please follow the logical order of the paper to number the figures. Also please write complete captions, avoiding to refer to captions in the main text.
**Thank you for all the comments and advice regarding the figures.**

Dear editor,

I have read this paper and think it fits the scope of ESD. Moreover, I think the material presented in the paper is an welcome addition to the knowledge on climatic changes in the Mediterranean. My main issues with the current manuscript are in the presentation. I think it is quite long and should be more focused, possibly with a reduction in the number of figures.

Kind regards,

**AU: We thank the reviewer for a very through revision comments and feedback regarding our manuscript. We have addressed all of the major comments when revising the manuscript. This includes either applying the reviewer's suggestion or a thorough justification for not doing that. We have also made an effort to address the minor comments. Overall, we've reworked the text of the manuscript to improve its conciseness and clarity.**

Specific comments:

1. The paper is quite long and elaborate, maybe wise to focus a bit more and reduce the number of figures?

**AU: Thank you for the comment. Following the reviewer's suggestion we will work on the manuscript to make it more comprehensible. We've reworked the manuscript to make it more comprehensible. Particularly, we have summarized the discussion of the previous studies in a more concise way (section 3.2.1 and 3.2.2, pp 9: lines 1-46, pp 10: lines 7-21). The relevant information on the Mediterranean climate moved from sections 3.2 and 3.3 (pp 8: lines 15-27) to the Introduction (pp2: lines 10-22). We also moved one figure with storm tracks to the Supplementary Material (Figure SI 2).**

2. There are many long sentences throughout the paper, which make it hard to follow the reasoning sometimes. I suggest to have a good look at opportunities to shorten them.

**AU: Thank you. We have improved the readability of the Introduction, Methods, Results and Summary.**

3. In many places in the manuscript, geographic names (Levant, Asia Minor, Balkans, Sahara, North Africa, EMED, and many more) are used to describe the model results. This assumes that the reader knows the location of all these places, which is probably not true. I suggest to indicate the relevant places in a figure and to be more specific in the geography when mentioning other places.

**AU: Thank you. We described the necessary regions in the Methods section, and provided the specific locations (particularly pp 6, lines: 12-17).**

4. You compare the high resolution model output to NCEP/DOE reanalysis of 2.5 degree resolution. Why not compare it to higher resolution products, such as ERA5?

**AU: Thank you for the comment. Incorporating datasets with higher resolution wouldn't necessarily serve the purpose of the analysis of the large-scale circulation features. We have included below an additional figure (Figure R2), depicting sea level pressure and wind vector at 850hPa both in ERA-I and ERA 20CR, showing very good agreement between the suggested observational datasets.**

[Figure]

[Figure]

Figure R2. Seasonal (JJA) sea level pressure (hPa) and wind vector at 850hPa (m/s) in (top) ERA-20C, (bottom) ERA-I; for the 1979-2017 period.

**However a verification of the precipitation results with higher resolution datasets would improve the robustness of the conclusions. Following reviewer's suggestion we have incorporated EOBS dataset in the analysis of the regional precipitation (Figure 4). EOBS provides high quality data and serves as a good reference for the regional precipitation. Given the same spatial resolution as CM2.5, EOBS allows for comparing very fine features over the complex orography of the Mediterranean.**

5. Is there a significant difference between the top and bottom panels in Fig 7? It is not clear to me, maybe use a different color scale?

**AU: Please note that the figure is now in the Supplementary Material. Yes there is a significant difference between the top and bottom correlations (but not the storm track patterns), computed between the stormtrack proxy and the SNAO index. For example, negative correlations over northwest Europe in the 1970-2011 (bottom) are weaker than -0.5, while the correlations in 1950-1990 (top) are stronger than -0.65. The difference in the computed correlations questions the robustness of the SNAO (not the storm tracks), derived from the recent three-to –four decades. Please find more discussion on this in pp 10, lines 30-40.**

6. Indicate different data sources (HIST, PROJ runs) in Fig 11c? Maybe by a vertical line in the plot?

**AU: Thank you, we have included the vertical line in the figure.**

7. P21L35, "This may stem from...". I agree that the anthropogenic forcing is not included in the control run and may contribute to this discrepancy. But what about other explanations?

**AU: Please note tht we substantially rewritten and clarified the Summary and Discussion section (18 and 19). We also included more information in Introduction (pp2, lines 11-16), and section 3.2.1 (pp 8, 32-36):**
**"The origin of the multi-decadal signal of SNAO has been linked (Knight et al. 2006, Folland et al 2009, Linderholm and Folland 2017) to the Atlantic Multidecadal Oscillation, which originates from internal variations in thermohaline circulation (Knight et al. 2005, 2006, Delworth and Mann 2000; Enfield et al. 2001), but for the recent decades also from anthropogenic sources (Rotstayn and Lohman 2002; Mann and Emanuel 2006)."**

**We would like to provide a more thorough explanation for the apparent better agreement between the observed and simulated SNAO before 1980s than for the later period:**

**This may stem from the fact that the SNAO component derived for the simulated and observed recent decades (1980s-2010s) could be to a large extent conditioned by a coincidence of the multidecadal scale internal (unforced) variability and multidecadal anthropogenic forcing. Taking into account the random nature of the unforced climate variations, their temporal evolution in the simulated 1980's-2010's period may look very different from the observed one. In a presence of an additional long-term component, such as anthropogenic forcing, the EOF analysis (which is set to extract a signal explaining a maximum variance) is likely to choose a combination of a random representation of the unforced variability and the anthropogenic forcing. This combination (regarded as the SNAO component) may have a very different form in the observations and simulations.**

[revised manuscript text omitted]

---

## Author Response (AR2)

The paper tackles the problem of climate change in the Mediterranean, exploring the role of teleconnections and local feedbacks in modulating future projections. To this aim, climate simulations are designed and run using a state-of-the-art climate model. The authors find climate change in agreement with previous literature (warming and drying), however their findings suggest reduced amplitudes in change. They explain these differences with the improved ability of the model in simulating SNAO teleconnections and land surface feedbacks.

**We thank the reviewer for their very through revision comments and feedback regarding our manuscript. We have addressed all of the major comments when revising the manuscript. This includes either applying the reviewer's suggestion or a thorough justification for not doing that. We have also made an effort to address the minor comments. Overall, we have substantially reworked the text of the manuscript to improve its conciseness and clarity. The manuscript is shortened by ~20%. The main focus of the study, described in the Introduction, is clarified. The key messages of the study, presented in the Discussion and Summary section, are clarified and enhanced. The number of figures is reduced from 12 to 10, by merging Figure 1 and 2 (in the previous version) into Figure 1, and Figure 8 and 9 (in the previous version) into Figure 6. We have also reworked the structure of the manuscript, so the text discussing the results is now included in Discussion and Summary. We have clarified the abstract.**

**We have also removed the discussion of the North Atlantic storm track, and the relevant figure (Figure 7 in the previous version of the manuscript). This part of the discussion will be very important in the further study that investigates the impact of the SNAO on northern Europe, but it is of minor importance for the Mediterranean hydroclimate investigated in the current manuscript.**

 Major comments 1. I acknowledge the huge work the authors did in carefully revising and discussing the literature, and in performing and discussing many analysis, but this makes the paper very long. My first general recommendation is to somehow shorten the manuscript, to facilitate the reader to be focused on the key messages delivered by the paper. For instance, discussions of previous findings is sometimes too detailed and redundant: accepted knowledge on summer climate in the Mediterranean region should be described in detail in the Introduction (or in a dedicated Background section) and briefly recalled when necessary in the text.

**AU: Thank you for the comment. Following the reviewer's suggestion we have reworked the manuscript to make it more comprehensible. The discussion of the previous studies is summarized in a more concise way and moved to the Introduction or to the Discussion and Summary section.  The accepted knowledge on the regional summer climate is moved from the section 3.2.1. and 3.2.2 (in the previous version: pp 9: lines 1-46, pp 10: lines 7-21) to the Introduction (in the current version: pp1, l 45 – pp2, l 29; pp2, l 40- pp3, l 18). Please note that the Introduction is partly rewritten and shortened. The results are discussed through the prism of previous findings, and those are recalled when necessary in Discussion and Summary (in the current version: pp 15, l1-15, pp 16 l 25-46).  The Discussion and Summary is substantially rewritten and shortened, the key message is highlighted and clarified.**

2. The main finding of the paper is the different amplitude of future projection of the Mediterranean climate simulated by the CM2.5 model in comparison with CMIP3 and CMIP5 simulations. To illustrate this crucial aspect, the authors refer to the existing literature on the topic. However, when quantitative differences are discussed, comparison with a Figure would be helpful. I suggest the authors to add some figures (in the supplement) showing projections of the future Mediterranean climate (precipitation and temperature) in the CMIP5 ensemble or, if downloading CMIP5 data is too time consuming, at least the CM2.1 model output, to show differences with the same model at lower resolution and not including the improved land model LM3.

**AU: Thank you for the suggestion. In order to facilitate the comparison between the CM2.5 future projections and other models we have updated references and also included a supplementary figure. Figure R1 depicts changes estimated based on the RCP8.5 CSIRO-Mk3-6-0 model 10-member projections ensemble. Unfortunately, there are no RCP scenarios for GFDL CM2.1. Instead, we use**

**the CSIRO-Mk3-6-0 model, which includes the ocean component based on the GFDL ocean model. We also refer to the very comprehensive analysis in Jacob et al. 2014 and Fussel et al. 2017 (Map 3.4, pp. 76), employing many GCM-RCM combined simulations from the EURO-CORDEX project. Discussion of these results is included in the text (pp10: lines 34-41, pp 16: lines 25-32).**

[Figure]

Figure R1. Projected future changes for the summer (JJA) surface temperature (left,°C), and precipitation (mm/day, right) based on the 10-member ensemble simulations of the CSIRO-Mk3-6-0 model, for the forcing scenario RCP8.5.

3. Data: why NCEP reanalysis are selected for comparison? On the same period, ERAI data are available at higher resolution. ECMWF datasets are also available for the 20th century. Same question about precipitation data: why University of Delaware? Testing other temperature/precipitation datasets (CRU, EOBS) would change your results? In general, comparing your results with different datasets would improve the robustness of your conclusions.

**AU: Yes, we agree that a verification of the results with different datasets would improve the robustness of the conclusions. Following reviewer's suggestion we have incorporated EOBS dataset in the analysis of the regional precipitation (Figure 3). We agree that EOBS provides high quality data and serves as a good reference for the regional precipitation. Given the same spatial resolution as CM2.5, EOBS allows for comparing very fine features over the complex orography of the Mediterranean.**
**However, incorporating datasets with higher resolution would not necessarily serve the purpose of the analysis of the large-scale circulation features. We have included below an additional figure (Figure R2), depicting sea level pressure and wind vector at 850hPa both in ERA-I and ERA 20CR, showing very good agreement between the suggested observational datasets.**

[Figure]

Figure R2. Seasonal (JJA) sea level pressure (hPa) and wind vector at 850hPa (m/s) in (top) ERA-20C, (bottom) ERA-I; for the 1979-2017 period.

**In the analysis of the mid-level and upper level atmospheric dynamics, we would refrain from using the 20th century data sets such as ECMWF's ERA-20C or NOAA-CIRES 20th Century Reanalysis. Both of these data sets are based on the assimilation of the surface pressure values, which makes them a less plausible reference in the analysis of the upper level atmospheric dynamics. For example, the comparison of the vertical velocity at 500 hPa in Figure R3 reveals a strong positive bias in ERA-I, compared to the rest of the analyzed datasets (note the difference in the scalebar).**

[Figure]

Figure R3. Seasonal (July) time-mean vertical velocities at 500hPa (Pa/s), estimated for (top) NOAA-CIRES 20th Century Reanalysis, (middle) as ECMWF's ERA-20C, (bottom) ERA-Interim for 1979-2017.

4. One main issue of the paper is the choice of the time window to be analysed. The aim of the paper is to study summer Mediterranean climate, so that you select JJA. However, through the paper, different periods are selected for different analysis: JA or just July. I recommend to homogenize the period to be analyzed (preferably JJA), for better comparison of the results. If there are specific reasons to analyze different period, these reasons should be highlighted.

**AU: The choice of the time window is dictated by the method being most adequate to the analyzed climate component.. The teleconnection of SNAO with the climate over the Mediterranean region is manifest mostly during the peak summer, i.e. July-August. On the other hand the analysis in section 3.3. "Summer climate regime over the eastern Mediterranean" focuses on the month of July. "The choice is driven by the fact that the magnitude of subsidence and the Etesians is at its maximum in July while the response of the Rossby waves to monsoon rainfall is also the strongest (Tyrlis et al. 2012, Lin et al. 2007, Lin et al. 2009). The justifications of all the time window choices are included in the manuscript, but we have highlighted this information both in Methods (pp 5, lines: 8-12, 14-17,41-44) and in the text (pp 9, lines: 6-9).**

5. Methods section is rather long and sometimes confused: EOF analysis is described twice, the description of correlation/regression analysis is not really necessary here, as well as the reference to figures discussed later in the paper. I recommend to focus the section on the description of more sophisticated methods, such as EOF and stormtrack definition, and leave the description of correlation/regression analysis to the Results section. The section should be then shortened and optimised.

**AU: We have clarified and rewritten the Method section in a more concise way (pp 6, lines: 12-27, 29-38; pp 7, lines: 1-21). Additionally, we have removed the description of storm track definition (p5, 40-46), which follows the decision of not including the discussion of the impacts of SNAO through the perspective of the North Atlantic storm track.**

6. Model validation: in Figures 1-4 you compare the CTRL simulation to the NCEP data, and I see some important biases in terms of intensity (SLP in monsoonal regions) and location of some features (axes of the anticyclonic circulations at 500 hPa). This is due to the fact that in NCEP reanalysis there is GHG forcing, which is not included in the CTRL simulation (as you also highlight in the text, P21, L34-37). It would not be more consistent to compare the CTRL simulation to a different period, i.e. a period of 20C reanalysis/precipitation less affected by GHG forcing?

**AU: We would refrain from the suggestion offered by the reviewer. Comparing CTRL simulation with a different or longer period of observations than the recent three decades would be a more appropriate solution, from the perspective of contributing forcing components. However, the quality of the observations-based precipitation datasets is much lower before the satellite era (~1979). For the analysis of atmospheric circulation we prefer to use a shorter period of NCEP/NCAR2 (DOE) data set rather than the longer Twentieth Century Reanalysis. We justify this choice with the findings in Krueger et al. 2013. This study has shown that the early part of the SLP record (first half of the 20th century) in the Twentieth Century Reanalysis suffers substantial inhomogeneities, most likely associated with the increasing number of observations and improved measurement techniques. Moreover, taking into account that both the NOAA's 20CR Reanalysis as well as ECMWF's ERA-20C datasets assimilate only surface pressure reports, sea ice, and sea surface temperature distributions, the expression of the high level atmospheric variables is highly uncertain in these datasets, as shown for example in the Figure R2 (omega at 500hPa) in the response to question #3.**

7. SNAO simulation: the analysis of SNAO impact in Figure 5 is not compared with any reanalysis product. The SNAO impact on climate in Europe is explained with variations in the stormtrack (Figure 7): why reanalysis data are shown and not model simulation?

**AU: The representation of the SNAO in observations, using the SLP in different time periods, is shown in Figure 6. However the observed impact of SNAO has been shown in detail in Folland et al. 2009 and Blade et al. 2012. We prefer to make a reference to the existing literature rather than to repeat an existing analysis, especially to maintain brevity given major comment 1 requesting that we shorten the manuscript length.**

**Yes, the SNAO teleconnection with hydroclimate in northwestern Europe can be directly explained with variations in large-scale circulation over the North Atlantic and associated variations in storm tracks (as shown in observations and simulations in Folland et al. 2009). In this manuscript we do not intend to investigate the simulated impacts of SNAO on the North-Atlantic storm tracks and the northwestern Europe hydroclimate. This topic will be investigated in a further study. We have clarified the text (pp8, lines: 37-41) and also the main focus of the study (in Introduction, pp3, lines: 20-37). To maintain brevity, given major comment 1 requesting that we shorten the manuscript length, we have removed the figure representing the variations in storm track.**

8. Figure 11: the caption of the figure and discussion at P17 should be improved. You first state that you compute EOFs for the CNTR simulation, than you project the HIST-PROJ fields onto the CTRL EOF to get the 1860-2100 time series. Than you state that you also compute EOFs separately for HIST and PROJ. Then you discuss the 1860-2100 time series in Fig. 11c, then you go to HIST and PROJ EOFs in Fig. 11ab, and finally you discuss the contribution of the 1860-2100 SNAO to the end-of century projection of precipitation (Fig. 11de). I find this discussion confusing. This is a crucial point of the paper and should be presented clearly. I recommend the authors to improve the readability of this section. Moreover, the discussion of the SNAO impact on temperature projection should be significantly expanded.

**AU: In this section we address three problems, which requires three different analysis approaches. We consider important to justify each of them. We have made a substantial effort to optimize the length of the discussion, clarified the discussion and improved the relevant caption (pp 15, 16). Following the reviewer's suggestion, we have clarified this part of the results (p11, lines: 9-39), as well as the caption of the relevant figure and the corresponding methods.**

**Section 3.2.2 investigates the impact of SNAO on the Mediterranean precipitation and temperature, with focus on the former; section 4.2. analyzes the impact of the SNAO from the perspective of climate change. There is very limited knowledge on the impacts of SNAO on temperature (basically two papers: Blade et al. 2012, Folland et al. 2009), and these results are far from unequivocal. The results shown in these two articles differ depending on chosen length of dataset and method of analysis (e.g. whether the correlations are based on interannual variations or including multidecadal signal). Our analysis seems more consistent with Folland et al. 2009, who applied rigorous statistics to avoid effects of autocorrelation caused by low-frequency components in a relatively short data set. We have now updated the relevant information on temperature in section 3.2.2 (pp8, lines 21-46) and included an extended discussion in Discussion and Summary (pp15, lines 1-23). However, we would prefer to abstain from further discussion, to avoid speculation. If the reviewer has a certain opinion on this issue, it might be helpful to receive a more elaborate suggestion.**

9. Results: the differences between CM2.5 and CMIP5 models in projecting the paper Mediterranean climate are explained with a) better representation of the SNAO teleconnection and b) improved representation of the land-atmosphere interaction by the LM3 model (see also the Abstract). However, the improvements of the LM3 model are not presented in the paper, nor how these new features actually improve the representation of the land-atmosphere interactions (*e.g. representation of soil moisture, evapotranspiration, albedo*). A brief presentation of the LM3 model as well as a discussion of how it improves climate simulation in the Mediterranean is needed.

**AU: Thank you for bringing up this gap in our writing. We have clarified the Abstract, added references to the data section (pp 3, lines: 41-46, pp4, lines: 1-9), and added text to the Discussion and Summary section (pp 16, lines 34-46) relating to land model improvements and some research highlighting LM3 performance. We agree that more analysis is required to further support this claim, which would add significant text and figures to the paper. We have included references of studies beginning to look into this problem, but have left the exploration of land-atmosphere interactions of the SNAO using LM3 for future research.**

Added references:

Berg, A., B.R. Lintner, K. Findell, S.I. Seneviratne, B. van den Hurk, A. Ducharne, F. Chéruy, S. Hagemann, D.M. Lawrence, S. Malyshev, A. Meier, and P. Gentine, 2015: Interannual Coupling between Summertime Surface Temperature and Precipitation over Land: Processes and Implications for Climate Change. J. Climate, 28, 1308–1328, https://doi.org/10.1175/JCLI-D-14-00324.1

Berg, A., Findell, K., Lintner, B., Giannini, A., Seneviratne, S. I., Van den Hurk, B., Ruth Lorenz, Andy Pitman, Stefan Hagemann, Arndt Meier, Frédérique Cheruy, Agnès Ducharne, Sergey Malyshev & P. C. D.

Milly, 2016: Land–atmosphere feedbacks amplify aridity increase over land under global warming. Nature Climate Change, 6(9), 869.

Milly, P.C., S.L. Malyshev, E. Shevliakova, K.A. Dunne, K.L. Findell, T. Gleeson, Z. Liang, P. Phillipps, R.J. Stouffer, and S. Swenson, 2014: An Enhanced Model of Land Water and Energy for Global Hydrologic and Earth-System Studies. J. Hydrometeor., 15, 1739–1761, https://doi.org/10.1175/JHM-D-13-0162.1

 10. Conclusions: most of the paper is devoted to the analysis of the SNAO teleconnection and its impact on future climate change in the Mediterranean, which show a significant (P17, Figure 11) impact on precipitation in southern Europe. And in the abstract you indicate this as one of the main results of the paper. Conversely, in the Conclusions you somehow reduce the importance of the SNAO impact (P22, L38-40), explaining the differences with the CMIP5 simulations as a consequence of the improved land model. This point needs to be clarified.

**AU: Following the reviewer's suggestion, we have clarified the main message of the manuscript. We agree that the analysis of the impacts of the SNAO teleconnection consumes a significant part of the manuscript. This part shows a significant contribution of the SNAO to precipitation over southern Europe. However, the comparison of the SNAO impacts between CM2.5 model and CMIP3/CMIP5 models suggests that the SNAO cannot explain the difference in the future projections between these models. The apparent stark contrast between the CMIP3/CMIP5 and CM2.5 regional projections could more likely originate from the enhancements in the LM3 land model incorporated to CM2.5 at high spatial resolution, rather than the impacts of the SNAO. The relevant discussion is included in Discussion and Summary: pp16, lines:34-46, pp17 lines: 1-13. Future work should focus on understanding differences in land surface responses in this region to the SNAO and projected climate change.**

Minor comments:
P2, L15: the connection between the Mediterranean and the African monsoon has been robustly described as Mediterranean → Africa (see papers by Raicich et al. 2003 and Rowell 2003 [https://doi.org/10.1175/1520- 0442(2003)0162.0.CO;2]).

The influence of the African monsoon on the Mediterranean is less clear: Ziv et al. 2004, but also Fontaine et al. 2011 [https://doi.org/10.1002/joc.2108], actually find a link between convection in Africa and subsidence in the Mediterranean, however the mechanism is still not clear (see Gaetani et al. 2011 [https://doi.org/10.1029/2011GL047150]). Indeed, the Asian monsoon could be dominant in modulating the Mediterranean-Africa connection. Please modify the sentence to account for this aspect.

**Thank you for the comment. We found and included all the missing references, and modified the sentence (Raicich et al. 2003 and Rowell 2003, Fontaine et al. 2011) (pp 2, lines: 14-17).**

P2, L26-29: please add a reference.
P3, L18-20: please add a reference.
**Thank you, we add the references.**

P3, L43: "fixed levels of radiative forcing", do you mean 'radiative forcing from fixed levels of emission/concentration'?
**Thank you, the updated information is now included in pp: 4, lines 16-21.**

P4, Methods: is the model fully coupled? How many vertical levels are in the ocean model?
**We have updated the information (pp 4, lines: 1-9).**

P6, L15-16: this sentence should be moved to the Results section. **Thank you, we have applied the suggestion.**

P6, L18-19: what do you mean with "vector time series"? The time series of the vector containing spatial data? **Correct.**

P7, L39: from Figure 4, precipitation magnitude is actually, not "apparently", larger than observations.
**We have corrected and rewritten the paragraph. (pp 7, lines 1-11)**

P7, L45: "none of the CMIP5. . ." **We have rewritten the relevant paragraph.**

P8, L1: do you mean that the CM2.5 runs in the CMIP5 archive are better than other models in the archive? Or do you refer to the runs you analyse in this paper? If this is the case, you should provide a figure to support this statement.
**Thank you. We referred to the CMIP5 analysis shown in Kelley et al. (2012). We have clarified this and add the reference where appropriate (pp3, lines: 9-14).**

P8, L10-13: when discussing the impact of NAO and SNAO on European climate, add references.
**Thank you. We have added the references throughout the paper.**
P8, L17: "and rather wet conditions".
**Thank you, we applied the correction (pp 2, line 3).**
P8, L18-19: add references on future projections of SNAO.
**This paragraph is now rewritten. We have updated the references on future projections of SNAO (pp10, lines: 8-12, pp 14, lines 27-29, pp 15, lines 42-43).**

Section 3.2: the objective is to test the capability of the model in simulating the SNAO as an independent internally-generated mode of climate variability. However, the long introduction at P8-9 does not actually help in understanding why this is necessary. Is the internal variability modulated at multidecadal time scales? Is this modulation externally forced? Please try to clarify motivations and objectives of the section.

**Thank you, we have rewritten and clarified the section (pp7, line 16 - pp8, line 46). Part of the information has been moved to Introduction. As stated in the introduction of section 3.2, the purpose is to analyze the capability of the CM2.5 model to simulate the SNAO as an independent, internally generated climate component, which would prove the physical validity of the statistically-derived component. The "internally-generated" means that that the modulation is generated by the internal climate variations and it is not externally forced. However "the origin of the multi-decadal signal of SNAO has been linked (Knight et al. 2006, Folland et al 2009, Linderholm and Folland 2017) to the Atlantic Multidecadal Oscillation, which originates from internal variations in thermohaline circulation (Knight et al. 2005, 2006, Delworth and Mann 2000; Enfield et al. 2001), but for the recent decades also from anthropogenic sources (Rotstayn and Lohman 2002; Mann and Emanuel 2006)."**

P8, L20-26: this paragraph is confusing: on the one hand, it is true that different approaches/datasets may lead to uncertainty in the observed SNAO-Mediterranean teleconnection; on the other hand, uncertainties in model simulations originate from model shortcomings. Therefore uncertainty in the real and model worlds could originate from both intrinsic non-linear nature of the phenomenon and inadequate statistical/modelling tools. Please rephrase.
**Yes, we have substantially shortened and clarified this section. Additional discussion is included pp 15, lines 1-23.**

P9, L8-9: I cannot understand why and how anthropogenic forcing should intensify SNAO contribution (to the summer atmospheric circulation over North Atlantic). Please explain.
**We have substantially rewritten this section and this sentence is not included. The relevant information is included in pp15, lines 38-46, pp10, lines: 8-12.**

P9, L34: add the figures for July and August to the Supplement.
**We won't add this figure to the Supplementary Information, which already contains 10 supplementary figures, but we have updated the information that the figure is not attached.**

P9, L35: what is the interest of comparing with the HadCM3 model?
**This section analyzes the capability of CM2.5 in simulating SNAO and compares it with the available results of other models, in this case HadCM3 and HadGEM1 in Folland et al. 2009.**
P9, L42: is it HadGEM1 or HadCM3?
**The statement is correct.**
P13, L6-7: why an East Mediterranean index is used to compute correlation in Figure 8d, instead of the first EOF for NCEP omega? **The figure (in the current version Figure 6d) shows the correlations computed based on three-decade time series of NCEP omega at the mid-atmospheric level. Taking into account the relatively short length of the data set (compared to 1000 years CTRL run) and a relatively smaller plausibility of the data at the middle and higher atmospheric levels, we refrain from applying an EOF analysis to the NCEP dataset. In our consideration, applying an EOF analysis to such a short time series could lead to degeneracy of the derived eigenvalues. In other words, the EOF mode derived from the NCEP dataset could be easily a spurious combination of several modes, rater than a realistic representation of the SNAO mode.**

P14, L25-26: what do you mean with "estimated at the original model resolution"? Do you mean "computed"? **Thank you. The section is substantially rewritten and the sentence is removed.**

P14, L32: east. **We have applied the correction (pp 10, line 10)**
P17, L1-3: I don't understand why you refer to Fig. 10a (showing end-of-century projections) to discuss changes in SNAO. You could maybe use this figure to support your analysis of future SNAO.
**The figure shows the projected future changes (please note that in the current version this is Figure 7) and indeed we are using this figure to support the analysis of future SNAO (although it supports also the interpretation of the analysis of the already observed SNAO changes). The fingerprint of the future changes, derived from the sea level pressure projections, is consistent with the observed evolution of SNAO and thus it may constitute a possible contribution of the anthropogenic component already observed in the 20th century. We have tried to clarify this issue in the manuscript (pp 11, lines: 9-28).**

P17, L14-16: it would be preferable to present the regression method to estimate the SNAO impact here rather than in the Method section.
**Yes, the relevant information is now in the Methods section.**
P17, L43: "warming is lower over . . . than . . ."
**Thank you, we have substantially rewritten and improved this section.**
P18, L7-12: it is not clear to me whether you are discussing you results (in Figure 10) or previous findings. If you discuss your results, please add more references to Figure 10, otherwise add a reference to a paper.
**Yes, we have clarified the paragraph.**
P20, L39-41 and 42-45: please add references.
**Thank you, we have updated references in the section.**
P21, L20: "preindustrial value".
**We have substantially shortened and clarified whole section, the relevant sentence is removed.**
P22, L9-10: please add a citation to CMIP5 results.
**Thank you, have updated the references where needed.**

Figures: for better comparison, figures presenting climate change in the Mediterranean should share the same geographical boundaries. Same recommendation for figures presenting SNAO and Asian monsoon teleconnections, respectively.

**Unfortunately this idea is not feasible and doesn't serve the purpose of the analysis. Each figure is presented in different context. For example, projected future changes for SLP are discussed in the context of the Euro-Atlantic climate. In contrast, regarding precipitation and temperature we want to focus on the Mediterranean region, but also discuss it form the perspective of climate in Europe. We have thoroughly reconsidered the way each figure is presented.**

Figure 6: what do contours represent? The sign looks reversed with respect to the standard SNAO pattern. Could you please fix this, not to mislead the reader?

**We have improved the figure and presented the SNAO at its positive phase. We have included the clarifying information in the caption. However, we would abstain from labeling the positive or the negative phase to a "standard pattern".**

Figure 7: does it make sense to project the SNAO index derived from 20CR onto NCEP data? Why not just analyse one dataset?
**We agree with the reviewer that it is usually easier to use just one data set. However for the sake of consistency with an earlier part of the analysis, we used the 20CR dataset instead of the NCEP dataset. In the earlier part of the analysis we used the 20CR dataset, because the alternative ones, such as NCEP-NCAR1 or NCEP –DOE, would be too short for the analysis of the evolution of SNAO during the 20th century.**

Figure 8: Do you perform EOF on omega 500 and 300 together? Or is EOF analysis performed separately on omega 500 and 300? If this is the case, which time series do you use for correlations?
**Yes, the EOF analysis is performed separately for each level. The method is described in the manuscript but we will clarify and highlight this information also in the caption (Figure 6, pp5, lines 33-35).**

Figure 10: wind is displayed at which level? Is not model resolution 0.5?
**Thank you, we have included the missing information and applied the correction (Figure 7).**
Figure 11: in panels d and e you show regressions, while in Fig. 9c you show correlations.
**Figure d and e doesn't show regressions, but the projected changes. The impact of the SNAO in figure e is removed using linear regression. We have clarified the figure the caption.**

Figure 12: is omega at 200 or 500?
**It is omega at 500 hpa. The information is now updated (Figure 9).**

See P18, L23. Supplement: please follow the logical order of the paper to number the figures. Also please write complete captions, avoiding to refer to captions in the main text.
**Thank you for all the comments and advice regarding the figures.**

Dear editor,
I have read this paper and think it fits the scope of ESD. Moreover, I think the material presented in the paper is an welcome addition to the knowledge on climatic changes in the Mediterranean. My main issues with the current manuscript are in the presentation. I think it is quite long and should be more focused, possibly with a reduction in the number of figures.
Kind regards,

**AU: We thank the reviewer for their very through revision comments and feedback regarding our manuscript. We have addressed all of the major comments when revising the manuscript. This includes either applying the reviewer's suggestion or a thorough justification for not doing that. We have also made an effort to address the minor comments. Overall, we have reworked the text of the manuscript to improve its conciseness and clarity. The manuscript is shortened by ~20%. The main focus of the study, described in Introduction, is clarified. The key messages of the study, presented in Discussion and Summary, are clarified and enhanced. The number of figures is reduced from 12 to 10, by merging Figure 1 and 2 (in the previous version) into Figure 1, and Figure 8 and 9 (in the previous version) into Figure 6. We have also reworked the structure of the manuscript, so the text discussing the results is now included in Discussion and Summary. We have clarified the abstract.**

**We have also removed the discussion of the North Atlantic storm track, and the relevant figure (Figure 7 in the previous version of the manuscript). This part of the discussion will be very important in the further study that investigates the impact of the SNAO on northern Europe, but it is of minor importance for the Mediterranean hydroclimate investigated in the current manuscript.**

Specific comments:
1. The paper is quite long and elaborate, maybe wise to focus a bit more and reduce the number of figures?

**AU: Thank you for the comment. Following the reviewer's suggestion we have worked on the manuscript to make it more comprehensible. Particularly, we have reworked the introduction, the results and the discussion of the previous studies in a more concise way (section 3.2.1 and 3.2.2). The relevant information on the Mediterranean climate moved from sections 3.2 and 3.3 to the Introduction (pp2: lines 10-22). We have also removed the figure with storm tracks.**

2. There are many long sentences throughout the paper, which make it hard to follow the reasoning sometimes. I suggest to have a good look at opportunities to shorten them.

**AU: Thank you. We have improved the readability of the Introduction, Methods, Results and Summary.**

3. In many places in the manuscript, geographic names (Levant, Asia Minor, Balkans, Sahara, North Africa, EMED, and many more) are used to describe the model results.
This assumes that the reader knows the location of all these places, which is probably not true. I suggest to indicate the relevant places in a figure and to be more specific in the geography when mentioning other places.

**AU: Thank you. We have described the necessary regions in the Methods section, and provided the specific locations (pp 5, lines: 1-6, 14-29).**

4. You compare the high resolution model output to NCEP/DOE reanalysis of 2.5

degree resolution. Why not compare it to higher resolution products, such as ERA5?

**AU: Thank you for the comment. Incorporating datasets with higher resolution would not necessarily serve the purpose of the analysis of the large-scale circulation features. We have included below an additional figure (Figure R2), depicting sea level pressure and wind vector at 850hPa both in ERA-I and ERA 20CR, showing very good agreement between the suggested observational datasets.**

[Figure]

Figure R2. Seasonal (JJA) sea level pressure (hPa) and wind vector at 850hPa (m/s) in (top) ERA-20C, (bottom) ERA-I; for the 1979-2017 period.

**However a verification of the precipitation results with higher resolution datasets would improve the robustness of the conclusions. Following the reviewer's suggestion we have incorporated EOBS dataset in the analysis of the regional precipitation (Figure 3 in the current version). EOBS provides high quality data and serves as a good reference for the regional precipitation. Given the same spatial resolution as CM2.5, EOBS allows for comparing very fine features over the complex orography of the Mediterranean.**

5. Is there a significant difference between the top and bottom panels in Fig 7? It is not clear to me, maybe use a different color scale?

**AU: Thank you. Yes, there is a significant difference between the top and bottom correlations (but not the storm track patterns), computed between the stormtrack proxy and the SNAO index. For example, negative correlations over northwest Europe in the 1970-2011 (bottom) are weaker than -0.5, while the correlations in 1950-1990 (top) are stronger than -0.65. The difference in the computed correlations questions the robustness of the SNAO (not the storm tracks), derived from the recent three-to-four decades.**
**Nevertheless, we have decided to remove this figure and the relevant discussion from the manuscript. The SNAO teleconnection with hydroclimate in northwestern Europe can be directly explained with variations in large-scale circulation over the North Atlantic and associated variations in storm tracks (as shown in observations and simulations in Folland et al. 2009). However, in this manuscript we do not intend to investigate the simulated impacts of SNAO on the North-Atlantic storm tracks and the northwestern Europe hydroclimate. This topic will be investigated in a further study. We have clarified the text (pp8, lines: 37-41) and also the main focus of the study (in Introduction, pp3, lines:**

**20-37). To maintain brevity, we have removed the figure representing the variations in storm track.**

6. Indicate different data sources (HIST, PROJ runs) in Fig 11c? Maybe by a vertical line in the plot?

**AU: Thank you, we have included the vertical line in the figure (Figure 8 in the current version).**

7. P21L35, "This may stem from...". I agree that the anthropogenic forcing is not included in the control run and may contribute to this discrepancy. But what about other explanations?

**AU: Please note that we have substantially rewritten and clarified the Summary and Discussion section (pp15, lines 1-23). We also included more information in Introduction (pp2, lines 1-12):**
**"The origin of the multi-decadal signal of SNAO has been linked (Knight et al. 2006, Folland et al 2009, Linderholm and Folland 2017) to the Atlantic Multidecadal Oscillation, which originates from internal variations in thermohaline circulation (Knight et al. 2005, 2006, Delworth and Mann 2000; Enfield et al. 2001), but for the recent decades also from anthropogenic sources (Rotstayn and Lohman 2002; Mann and Emanuel 2006)."**

**We would like to provide a more thorough explanation for the apparent better agreement between the observed and simulated SNAO before 1980s than for the later period:**

**This may stem from the fact that the SNAO component derived for the simulated and observed recent decades (1980s-2010s) could be to a large extent conditioned by a coincidence of the multidecadal scale internal (unforced) variability and multidecadal anthropogenic forcing. Taking into account the random nature of the unforced climate variations, their temporal evolution in the simulated 1980's-2010's period may look very different from the observed one. In the presence of an additional long-term component, such as anthropogenic forcing, the EOF analysis (which is set to extract a signal explaining a maximum variance) is likely to choose a combination of a random representation of the unforced variability and the anthropogenic forcing. This combination (regarded as the SNAO component) may have a very different form in the observations and simulations.**

[revised manuscript text omitted]

The warming projected in CM2.5 shows a stark gradient between the southwestern and northeastern parts of Europe, which is consistent with the CMIP5 and the EURO-CORDEX ensembles. However, for the latter, the gradient is weaker and the minimum of warming shifted northward (see Fussel et al. 2017, Map3.4; Figure SI6), i.e. located over the southeastern Baltic countries. In CM2.5 the minimum of warming is located in the northern Balkans and southeast Europe, in the vicinity of the Black Sea coast (Figure 7b), indicating values falling within 0.5-2.5°C and accompanied also with wetting tendencies. The regions such as the Iberian Peninsula, southern Balkans and Asia Minor feature warming between 3.5 and 6°C. The maximum of warming is located over North Africa and Levant, with values falling within the range of 5-8°
[revised manuscript text omitted]

Monika Barcikowska 8/9/19 8:19 AM
Formatted [22]

Monika Barcikowska 8/9/19 3:38 PM

Monika Barcikowska 8/9/19 9:08 AM

Monika Barcikowska 8/9/19 10:09 AM

Monika Barcikowska 8/9/19 10:28 AM

Monika Barcikowska 8/15/19 1:09 PM

Monika Barcikowska 8/9/19 4:38 PM

Monika Barcikowska 8/9/19 5:06 PM

Monika Barcikowska 8/9/19 6:14 PM
Deleted: We carry out a correlation analysis for these two periods, much as done in the previous section comparing recent historical and future periods. Figure SI shows a radical drop in derived correlations between the mid-level subsidence and the Mediterranean pressure, Etesian winds and their extension over the North Africa and Persian Gulf, and water vapor over the Sahel. These results are similar to those comparing present and future climate for these variables (Fig 11).

Monika Barcikowska 8/9/19 5:06 PM

Figure 10 depicts a direct response of the summer Mediterranean climate to the surface warming over EMED, estimated with composite differences between the two samples (high temperature minus low temperature), in terms of temperature, relative humidity, pressure and wind vector, geopotential height at 500 hPa and 800 hPa, omega at 500 hPa and precipitation. Figure 10c features bipolar SLP anomalies, with low-pressure anomaly over North Africa, EMED and the Middle East, and an anomalous anticyclonic circulation between northeast and northwest of this region. While the former is well collocated with the intensified heat low anomalies found as an anthropogenic signal (Figure 7a), the latter is centered over the Black Sea and spreads towards the central Mediterranean, creating a stronger zonal pressure gradient over the Mediterranean and intensified Etesian winds. The intensified heat low over the EMED and Arabian Peninsula (Figure 10c) is also consistent with the enhanced local convergence and reduced subsidence at the low- and mid-tropospheric levels at 500 hPa (Figure 10e) and 700 hPa (not shown). At the same time, the positive SLP anomalies (Figure 10c) and the increased subsidence over Asia Minor and the Black Sea are physically consistent with increased adiabatic warming and stability, manifest in the local maximum warming, reduced relative humidity and precipitation. The analysis repeated for the July-August season yields similar results, although with a reduced magnitude due to a weaker signal in June and August (Figure SI7).

The analysis repeated for the response to the warming over the domains extended towards southern parts of the central and western Mediterranean (Figure SI8a,b) yields qualitatively similar results (i.e. the bipolar SLP anomalies), but with an increased magnitude of the response over the southwestern Mediterranean. On the other hand, analysis repeated for the warming regions confined to the Levant, Arabian Peninsula and Asia Minor and Black Sea (30°-50°E, 30°-45°N, Figure SI8e), shows the pattern with the response (anticyclone anomaly) intensified towards the Middle East. The most similar results are obtained, qualitatively and quantitatively, when the region is confined to the same latitudes but slightly extended towards east and west (30°-50°E, 30°-36°N, Figure SI8c), i.e. centered over the Levant and northern parts of Arabian Peninsula.

The derived composite response to the warming over EMED is manifest in the local surface circulation. The resulting bipolar SLP anomaly leads to the intensified zonal pressure gradient and the concomitant Etesians over the central-eastern Mediterranean. The positive SLP anomalies, centered over the central and northeastern parts of the Mediterranean are consistent with the increased subsidence and drying in these regions (i.e. Italy, the Balkan coast and Asia Minor). The intensified heat low over the EMED and associated anomalous convergence weaken the low- and mid-tropospheric subsidence over the EMED and the Arabian Peninsula. All these responses are consistent with the future changes projected in JJA (Figure 7a,b,c, Figure SI4a,c,e), and particularly in the month of July (Figure SI4b,d,f), suggesting an important role of the warming arid regions of Levant and Arabian Peninsula in the future climate regime of the eastern Mediterranean.

This analysis indicates that the dynamical regime over the EMED has a nonlinear influence on local temperature. During relatively cool years the dynamical relationship between the low-level circulation and mid-level subsidence, which balances the temperature over EMED, seems to be much stronger. By contrast, warming over the EMED region can trigger a local response in the surface atmospheric circulation, which weakens the local dynamical linkages and hence their contribution in maintaining the local temperature balance. Hence it is possible that surface temperature-driven atmospheric responses will become a more prominent factor shaping future Mediterranean climate. This idea is supported by the consistency of this response (i.e. strong warming, intensifying heat low, anomalous convergence and very pronounced ascending motion at the low and mid-levels of the EMED and Arabian Peninsula, intensified zonal pressure gradient and Etesians, drying over Asia Minor and southern Balkans) with projected anthropogenic changes over the

EMED region. Similar results are obtained for June and August, although with slightly lower intensity and location of the SLP and precipitation anomalies (not shown).

The analysis, however, does not explain the processes involved in the dipole-like response in the circulation, which comprises SLP, winds and omega anomalies north from the EMED region (particularly Asia Minor and the Black Sea). One might suspect that, in response to warming over the EMED, the anomalous convergence and ascending motion over the EMED triggers a seesaw connection with northward-located regions. This link could stem from the interactions of the anomalous warming and upward velocity anomalies with the seasonally varying descending branch of the Hadley cell over EMED, in result expanding it towards Asia Minor. Testing this hypothesis needs more elaborate analysis and could be the objective of future research.

**5. Summary and Discussion**

[revised manuscript text omitted]

Consistent with the previous CMIP ensembles, CM2.5 also projects a strong gradient between warming in southwestern Europe and weaker warming in northeastern Europe. For example, the warming over the Iberian Peninsula, southern Balkans and Alps reaches locally 6°C, and the warming over the North African coast and the inland Levant region exceeds locally 7°C. The warming over northern and central parts of the region (i.e. southern France and in Italy) is slightly lower and reaches locally up to 4.5-5°C.

However, the warming projected in CM2.5 (Fig 7b,c) is much less radical, when compared to the CMIP3 (Dubrowski et al. 2014) and CMIP5 (Collins et al. 2013) ensembles, as well as the high resolution EURO-CORDEX GCM-RCM RCP8.5 multi-model ensemble (Fussel et al. 2017, Jacob et al. 2014, Figure SI5). This discrepancy is distinguishable in particular for the northern Balkans and southeastern Europe. There, CM2.5 shows a minimum warming of 0.5-2.5°C. In contrast, the 10-member RCP8.5 ensemble of the CSIRO-Mk3-6-0 model indicates warming exceeding 6°C for these regions (Figure SI5) and the multi-model ensemble average of combined GCM–RCM simulations from the EURO-CORDEX initiative (Fussel et al. 2017, Map 3.4, pp. 76) indicates warming of 3.5 - 5.5°C.

The very intense warming and drying over Europe projected in the CMIP ensembles has been linked to a temperature-dependent warm summertime bias, caused by deficient representations of moisture-temperature feedbacks in most of CMIP3 and CMIP5 models (Christensen and Boberg, 2012; Mueller and Seneviratne, 2014, Boberg and Christensen, 2012). On the other hand, Berg et al. 2016; Milly et al. 2014 demonstrated that the representation of soil moisture and land-atmospheric feedbacks between soil moisture and precipitation in the LM3 model, used in CM2.5, is significantly improved. Moreover, the atmosphere-land interactions have been shown to play an important role in the future summer climate, in particular, over central and southeastern Europe (Seneviratne et al. 2006, Diffenbaugh 2007, Hirschi et al 2011). In conclusion, the improvements in the land model incorporated in CM2.5 at its high spatial resolution are responsible for the stark contrast between the CMIP3/CMIP5 and CM2.5 regional projections (i.e. less intense warming and drying over Europe, including the minimum of warming and wetting tendencies in southeastern Europe). These feedbacks should be explored in more detail in future work using targeted experiments like the Global Land-Atmosphere Coupling Experiment (Seneviratne et al. 2013), but lie outside the scope of this paper.

Additionally, we find that the SNAO will have an important role in counterbalancing the thermodynamic effects of the projected drying over the Mediterranean, simulated in both the CM2.5 and CMIP projections. We have shown that CM2.5 projects a strengthening of the SNAO towards its positive phase, reflected in the strengthening of the meridional circulation cells over North Atlantic. The derived changes of the SNAO are manifest in the positive anomalies of precipitation (wetting) over large parts of the Mediterranean. For example, without the impact of SNAO, the drying projected in CM2.5 over the Iberian Peninsula, Italy, and the Balkan coast would be much stronger (locally up to ~30-40%).

[revised manuscript text omitted]

Monika Barcikowska 8/23/19 6:49 AM

Monika Barcikowska 8/23/19 6:50 AM

Monika Barcikowska 8/23/19 6:50 AM

Monika Barcikowska 8/23/19 6:50 AM

Monika Barcikowska 8/23/19 6:50 AM

Monika Barcikowska 8/23/19 6:50 AM

Monika Barcikowska 8/23/19 6:50 AM

Monika Barcikowska 8/23/19 6:50 AM

Monika Barcikowska 8/23/19 6:50 AM

Monika Barcikowska 8/23/19 6:50 AM

Monika Barcikowska 8/23/19 6:50 AM

Monika Barcikowska 8/23/19 6:50 AM

---

## Author Response (AR4)

We thank the editor for the comments and feedback regarding our manuscript. We have addressed all of the comments when revising the manuscript.

1) As often found, temperature fields are wrong, then also precipitation fields. You also hint sometimes on this. Can you say something about this in your discussion? Which regions do show this strong correlation and which regions not.
AU: Thank you. Following the suggestion we addressed this issue in Summary and Discussion: pg 15, l: 4-8. Accordingly, we have also updated references.

2) Minor: pg1, L46: add abbreviation NAO
AU: We added the correction. (pg 1, l: 45)

3) Minor: pg2, L2: add sea level pressure (SLP)
AU: We added the correction. (pg 2, l:1)

4) Pg2, L33: agirultural activies: which parts do you mean? Africa? As your land-atmosphere feedback is important then agricultural activities with irrigation will becoming more and more important. Can you say something how this is added in the current land models and if it is important?
AU: Thank you. Following the suggestion we addressed this issue. (pg 2, l:39 – pg 3, l:5). Accordingly, we have also updated references.

5) P2, L43: positive bias. A bias in what? Temperature or Precip?
AU: Thank you. We clarified the description. (pg 2, l: 41)

6) P2, L46: non-linear warming of hot-extremes: Non-linear to what? Is it that you find not a normal distribution? Or something else?
AU: We clarified the paragraph (pg 2, l: 44-45)

7) P3, L6: two times also: grammar
AU: Thank you, we corrected the text. (pg3, l:7)

8) P3, L21: improved land model- but especially improved land fluxes to the atmosphere ?
AU: Thank you. Yes, we added the information and the reference. (pg 3, l:23)

9) P10,L5: existing local relationships. What do you mean?
AU: We removed the sentence, because the relevant information is included in the following section. (pg 10, l:5)

10) P11,L23: the 7 degrees, is this the average over JJA during over whole day or only over daytime?
AU: The comment points to the text in pg 10, l 23  (not pg 11, l 23), in the previous version of the manuscript.
Yes, we added the information ("in JJA during the whole day"), which is found in pg 10,l 28 of the current version.

11) P12,L3-4: the changes in precip. Can you also give the % change?
AU: Yes, we added the information (pg 12, l:8-10)

12) P16, L37: 8 degrees: add over summer
AU: Yes, we added the information (pg 17, l3)

[revised manuscript text omitted]